# Learning Dynamics of Deep Matrix Factorization Beyond the Edge of Stability

**Avrajit Ghosh**[1]*, **Soo Min Kwon**[2]*, **Rongrong Wang**[1], **Saiprasad Ravishankar**[1], **Qing Qu**[2]
[1] Michigan State University,  [2] University of Michigan

## Abstract

Deep neural networks trained using gradient descent with a fixed learning rate $\eta$ often operate in the regime of "edge of stability" (EOS), where the largest eigenvalue of the Hessian equilibrates about the stability threshold $2/\eta$. In this work, we present a fine-grained analysis of the learning dynamics of (deep) linear networks (DLNs) within the deep matrix factorization loss beyond EOS. For DLNs, loss oscillations beyond EOS follow a period-doubling route to chaos. We theoretically analyze the regime of the 2-period orbit and show that the loss oscillations occur within a small subspace, with the dimension of the subspace precisely characterized by the learning rate. The crux of our analysis lies in showing that the symmetry-induced conservation law for gradient flow, defined as the balancing gap among the singular values across layers, breaks at EOS and decays monotonically to zero. Overall, our results contribute to explaining two key phenomena in deep networks: (i) shallow models and simple tasks do not always exhibit EOS (Cohen et al., 2021); and (ii) oscillations occur within top features (Zhu et al., 2023a). We present experiments to support our theory, along with examples demonstrating how these phenomena occur in nonlinear networks and how they differ from those which have benign landscape such as in DLNs.

## 1 Introduction

Understanding generalization in deep neural networks requires an understanding of the optimization process in gradient descent (GD). In the literature, it has been empirically observed that the learning rate $\eta$ plays a key role in driving generalization (Hayou et al., 2024; Lewkowycz et al., 2020). The "descent lemma" from classical optimization theory says that for a $\beta$-smooth loss $\mathcal{L}(\mathbf{\Theta})$ parameterized by $\mathbf{\Theta}$, GD iterates satisfy

$$\mathcal{L}(\mathbf{\Theta}(t+1)) \leq \mathcal{L}(\mathbf{\Theta}(t)) - \frac{\eta(2 - \eta\beta)}{2}\|\nabla\mathcal{L}(\mathbf{\Theta}(t))\|_2^2,$$

and so if the learning rate is such that $\eta < 2/\beta$, then the loss monotonically decreases. However, many recent works have shown that the training loss decreases even for $\eta > 2/\beta$, albeit non-monotonically. Surprisingly, it has been observed that learning rate beyond the stability threshold often provides better generalization over smaller ones that lie within the stability threshold. This observation has led to a series of works analyzing the behavior of GD within a regime dubbed "the edge of stability" (EOS). By letting $\mathbf{\Theta}$ parameterize a deep network, we formally define EOS as follows:

**Phenomenon 1** (Edge of Stability (Cohen et al., 2021)). *During training, the sharpness of the loss, defined as $S(\mathbf{\Theta}) := \|\nabla^2\mathcal{L}(\mathbf{\Theta})\|_2$, continues to grow until it reaches $2/\eta$ (progressive sharpening), after which it stabilizes around $2/\eta$. During this process, the training loss behaves non-monotonically over short timescales but consistently decreases over long timescales.*

Using a large learning rate to operate at the EOS is hypothesized to give better generalization performance by inducing "catapults" in the training loss (Zhu et al., 2023a). Intuitively, whenever the sharpness $S(\mathbf{\Theta})$ exceeds the local stability limit $2/\eta$, the GD iterates momentarily diverge (or catapults) out of a sharp region and self-stabilizes (Damian et al., 2023) to settle for a flatter region

---

*Equal contribution; Correspondence to ghoshavr@msu.edu; kwonsm@umich.edu

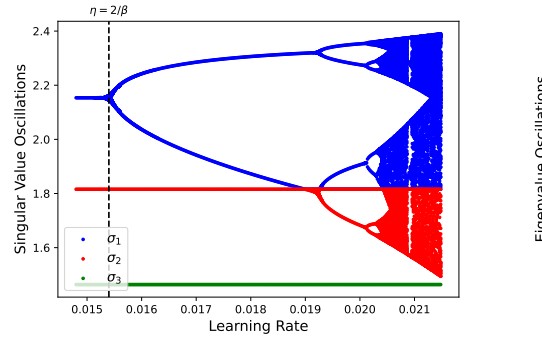 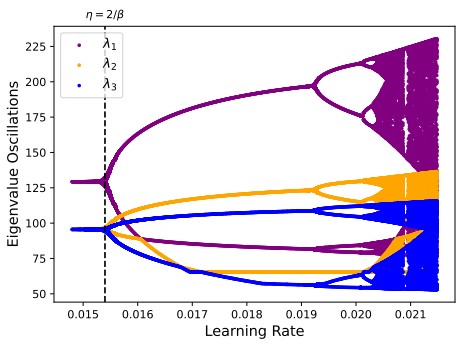

Singular Values of Weights          Eigenvalues of Hessian

Figure 1: Bifurcation plot of the oscillations in the singular values (left) and the eigenvalues of the Hessian (right) of a 3-layer end-to-end DLN. The bifurcation plots indicate the existence of a period-doubling route to chaos in DLNs, which we analyze by examining the two-period orbit. Here, $\eta > 2/\beta$ corresponds to the EOS regime, where $\beta = L\sigma_{\star,1}^{2-2/L}$ is the sharpness at the minima, $L$ is the depth of the network and $\sigma_{\star,1}$ is the first singular value of the target matrix $\mathbf{M}_\star$.

where the sharpness is below $2/\eta$. This self-stabilization mechanism enables GD to auto-regularize and find flatter solution which has shown to correlate with better generalization (Keskar et al., 2017; Izmailov et al., 2019; Petzka et al., 2021; Foret et al., 2021; Gatmiry et al., 2023). Of course, the dynamics within EOS differ based on the loss landscape. When the loss landscape is highly non-convex with many local valleys, catapults may occur, whereas sustained oscillations may exist for benign landscapes. When sustained oscillations occur, the sharpness hovers about the local stability limit $2/\eta$ rather than settling to a sharpness below $2/\eta$. We refer to this region as "beyond the EOS" following existing work (Wang et al., 2023; Zhu et al., 2023b). It is of great interest to understand these behaviors within different architectures to further our understanding of EOS.

From a theoretical perspective, there have been many recent efforts to understand EOS. These works generally focus on analyzing "simple" functions, examples including scalar losses (Zhu et al., 2023b; Wang et al., 2023; Kreisler et al., 2023), quadratic regression models (Agarwala et al., 2023), diagonal linear networks (Even et al., 2024) and two-layer matrix factorization (Chen & Bruna, 2023). However, the simplicity of these functions cannot fully capture the behaviors of deep neural networks within the EOS regime. Specifically, the following observations remain unexplained by existing analyses: (i) mild (or no) sharpening occurs when either networks are shallow or "simple" datasets are used for training (Caveat 2 from (Cohen et al., 2021)); and (ii) the oscillations and catapults in the training loss occur in the span of the top eigenvectors of the NTK (Zhu et al., 2023a).

In this work, we present a fine-grained analysis of the learning dynamics of deep linear networks (DLNs) beyond the EOS regime, demonstrating that these phenomena can be partially replicated and effectively explained using DLNs. Generally, there are two lines of work for DLNs: (i) those that analyze the effects of depth and initialization scale, and how they implicitly bias the trajectory of gradient flow towards low-rank solutions when the learning rate is chosen to be stable (Saxe et al., 2014; Arora et al., 2018; 2019; Pesme & Flammarion, 2023; Jacot et al., 2022), and (ii) those that analyze the similarities in behavior between linear and nonlinear networks (Wang et al., 2024; Zhang et al., 2024; Yaras et al., 2023). Our analysis builds upon these works to show that DLNs exhibit interesting behaviors outside the stability regime and to demonstrate how factors such as depth and initialization scale contribute to the EOS regime. Our main results can be summarized as follows:

- **Walk Towards the Flattest Global Minima Beyond EOS.** Similar to the observations made by Chen & Bruna (2023), we adopt the proof techniques of Kreisler et al. (2023) to show that the layers (or weights) of the DLN become increasingly balanced under mild assumptions at EOS. Specifically, we characterize how small the balancing gap at initialization must be for GD to reduce the imbalance over iterations. We further show that balanced minima correspond to the flattest minima in DLNs and our result captures an implicit regularization effect that drives the network toward the flattest minima at learning rates beyond the EOS regime.

- **Sharpness Scales with the Network Depth.** We identify all eigenvalues of the Hessian at the balanced minimum, demonstrating that the sharpness (i.e., the largest eigenvalue) scales with network depth. This rigorously validates the observations made by Cohen et al. (2021) and shows that the learning rate required to enter the EOS regime is depth-dependent, further highlighting its significance in deep networks.

- **Oscillations in Low-Dimensional Subspaces.** Once the network goes beyond the EOS regime, we prove that the network undergoes periodic oscillations within $r$-dimensional subspaces in DLNs, where $r$ is precisely characterized by the learning rate. For DLNs, a period-doubling route to chaos (Ott, 2002) exists in both the singular values of the DLN and the eigenvalues of the Hessian, as shown in Figure 1. We characterize the case of the two-period orbit, aiming to contribute to explaining the observations made by Zhu et al. (2023a) and Cohen et al. (2021).

## 2   NOTATION AND PROBLEM SETUP

**Notation.**   We denote vectors with bold lower-case letters (e.g., $\mathbf{x}$) and matrices with bold upper-case letters (e.g., $\mathbf{X}$). We use $\mathbf{I}_n$ to denote an identity matrix of size $n \in \mathbb{N}$. We use $[L]$ to denote the set $\{1, 2, \ldots, L\}$. We use the notation $\sigma_i(\mathbf{A})$ to denote the $i$-th singular value of the matrix $\mathbf{A}$. We also use the notation $\sigma_{\ell,i}$ to denote the $i$-th singular value of the matrix $\mathbf{W}_\ell$.

**Deep Matrix Factorization Loss.**   The objective in deep matrix factorization is to model a low-rank matrix $\mathbf{M}_\star \in \mathbb{R}^{d \times d}$ with $\mathrm{rank}(\mathbf{M}_\star) = r$ via a DLN parameterized by a set of parameters $\mathbf{\Theta} = (\mathbf{W}_1, \mathbf{W}_2, \ldots, \mathbf{W}_L)$, which can be estimated by solving

$$\underset{\mathbf{\Theta}}{\arg\min} \, f(\mathbf{\Theta}) \coloneqq \frac{1}{2} \| \underbrace{\mathbf{W}_L \cdot \ldots \cdot \mathbf{W}_1}_{=: \mathbf{W}_{L:1}} - \mathbf{M}_\star \|_\mathsf{F}^2, \tag{1}$$

where we adopt the abbreviation $\mathbf{W}_{j:i} = \mathbf{W}_j \cdot \ldots \cdot \mathbf{W}_i$ to denote the end-to-end DLN and is identity when $j < i$. We assume that each weight matrix has dimensions $\mathbf{W}_\ell \in \mathbb{R}^{d \times d}$ to observe the effects of overparameterization. We also assume that the singular values of $\mathbf{M}_\star$ are distinct.

**Optimization.**   Each weight matrix $\mathbf{W}_\ell \in \mathbb{R}^{d \times d}$ is updated using GD with iterations given by

$$\mathbf{W}_\ell(t) = \mathbf{W}_\ell(t-1) - \eta \cdot \nabla_{\mathbf{W}_\ell} f(\mathbf{\Theta}(t-1)), \quad \forall \ell \in [L], \tag{2}$$

where $\eta > 0$ is the learning rate and $\nabla_{\mathbf{W}_\ell} f(\mathbf{\Theta}(t))$ is the gradient of $f(\mathbf{\Theta})$ with respect to the $\ell$-th weight matrix at the $t$-th GD iterate.

**Initialization.**   In this work, we consider both balanced and unbalanced initializations, respectively:

$$\mathbf{W}_\ell(0) = \alpha \mathbf{I}_d, \quad \forall \ell \in [L], \tag{3}$$

$$\mathbf{W}_L(0) = \mathbf{0}, \quad \mathbf{W}_\ell(0) = \alpha \mathbf{I}_d, \quad \forall \ell \in [L-1], \tag{4}$$

where $\alpha > 0$ is a small constant. We assume $\alpha$ is chosen small enough such that $\alpha \in (0, \sigma_{\star,r})$, where $\sigma_{\star,r}$ is the $r$-th singular value of $\mathbf{M}_\star$. Generally, many existing works on both shallow and deep linear networks assume a zero-balanced initialization (i.e., $\mathbf{W}_i^\top(0)\mathbf{W}_i(0) = \mathbf{W}_j(0)\mathbf{W}_j^\top(0)$ for $i \neq j$). This introduces the invariant $\mathbf{W}_i^\top(t)\mathbf{W}_i(t) = \mathbf{W}_j(t)\mathbf{W}_j^\top(t)$ for all $t > 0$, ensuring two (degenerate) conditions throughout the training trajectory: (i) the intermediate singular vectors of each of the layers remain aligned and (ii) the singular values stay balanced. For the unbalanced initialization, the zero weight layer can be viewed as the limiting case of initializing the weights with a (very) small constant $\alpha' \ll \alpha$, and has been similarly explored by Varre et al. (2023) and Xu et al. (2024), albeit for two-layer networks. The zero weight layer relieves the balancing condition of the singular values. Rather than staying balanced, we show that the singular values become increasingly balanced (see Proposition 2). This allows us to jointly analyze the singular values of the weights.

Nevertheless, we also show that our analysis is not limited to either initialization but applies to *any initialization* that converges to a set we call the singular vector stationary set (see Proposition 1). To the best of our knowledge, it is common to assume that the singular vectors remain aligned, as many existing works make the same assumption (Varre et al., 2023; Arora et al., 2019; Saxe et al.,

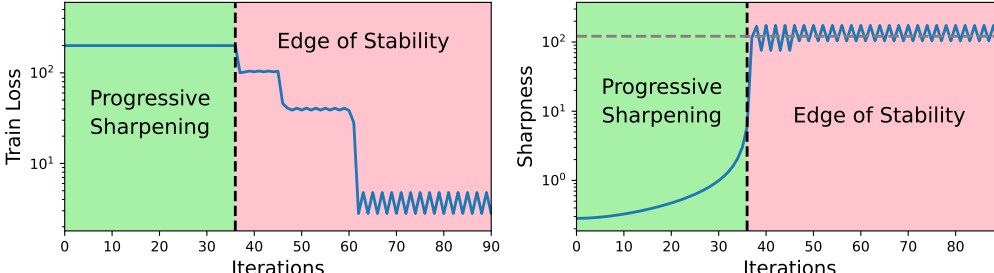

Figure 2: Depiction of the two phases of learning in the deep matrix factorization problem for a network of depth 3. Left: Plot of the training loss undergoing saddle jumps, followed by periodic oscillations. Right: Plot of the corresponding sharpness of the DLN. Upon escaping the first saddle point, the GD iterates enter the edge of the stability regime, where the sharpness hovers just about $2/\eta$.

2014; Gidel et al., 2019; Chou et al., 2024b; Min Kwon et al., 2024). Hence, throughout the rest of this paper, we refer to the balancing gap as the difference in singular values across layers and clarify where necessary.

## 3 DEEP MATRIX FACTORIZATION BEYOND THE EDGE OF STABILITY

When using a large learning rate, the learning dynamics can typically be separated into two distinct stages: (i) progressive sharpening and (ii) the edge of stability. Within the progressive sharpening stage, the sharpness lies below $2/\eta$ and tends to continually rise. Our goal is to analyze the EOS stage under the deep matrix factorization formulation. Here, we observe that the training loss fluctuates due to layerwise singular value oscillations, as illustrated in Figure 2.

### 3.1 ASSUMPTIONS AND ANALYTICAL TOOLS

Before we present our main result, we introduce two key analytical tools used in our analyses: the singular vector stationary set and singular value balancedness. First, we introduce the singular vector stationary set, which allows us to consider a wider range of initialization schemes. This set defines a broad class of weights for which singular vector alignment occurs, simplifying the dynamics of weights to those that only involve singular values.

**Proposition 1** (Singular Vector Stationary Set). *Consider the deep matrix factorization loss in Equation (1). Let $\mathbf{M}_\star = \mathbf{U}_\star \mathbf{\Sigma}_\star \mathbf{V}_\star^\top$ and $\mathbf{W}_\ell(t) = \mathbf{U}_\ell(t) \mathbf{\Sigma}_\ell(t) \mathbf{V}_\ell^\top(t)$ denote the compact SVD for the target matrix and the $\ell$-th layer weight matrix at time $t$, respectively. For any time $t \geq 0$, if $\dot{\mathbf{U}}_\ell(t) = \dot{\mathbf{V}}_\ell(t) = 0$ for all $\ell \in [L]$, then the singular vector stationary (SVS) points for each weight matrix are given by*

$$\mathrm{SVS}(f(\mathbf{\Theta})) = \begin{cases} (\mathbf{U}_L, \mathbf{V}_L) & = (\mathbf{U}_\star, \mathbf{Q}_L), \\ (\mathbf{U}_\ell, \mathbf{V}_\ell) & = (\mathbf{Q}_{\ell+1}, \mathbf{Q}_\ell), \quad \forall \ell \in [2, L-1], \\ (\mathbf{U}_1, \mathbf{V}_1) & = (\mathbf{Q}_2, \mathbf{V}_\star), \end{cases}$$

*where $\{\mathbf{Q}_\ell\}_{\ell=2}^L$ can be any orthogonal matrices.*

The singular vector stationary set states that for any set of weights where the gradients with respect to the singular vectors become zero, the singular vectors become fixed points for subsequent iterations. Once the singular vectors become stationary, running GD further isolates the dynamics on the singular values. Hence, throughout our analysis, we re-write and consider the loss

$$\frac{1}{2} \|\mathbf{W}_{L:1}(t) - \mathbf{M}^\star\|_{\mathsf{F}}^2 = \frac{1}{2} \|\mathbf{\Sigma}_{L:1} - \mathbf{\Sigma}^\star\|_{\mathsf{F}}^2 = \frac{1}{2} \sum_{i=1}^r \left( \sigma_i(\mathbf{\Sigma}_{L:1}(t)) - \sigma_{\star,i} \right)^2, \tag{5}$$

where $\mathbf{\Sigma}_{L:1}$ are the singular values of $\mathbf{W}_{L:1}$. This allows us to decouple the dynamics of the singular vectors and singular values, focusing on the periodicity that occurs in the singular values within the

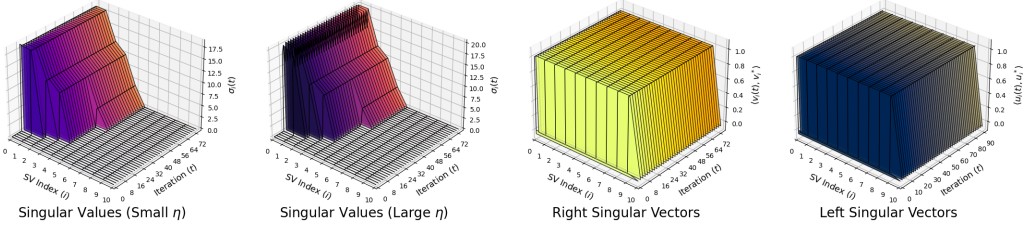

Figure 3: Illustrations of the singular vector and value evolution of the end-to-end DLN starting from the unbalanced initialization. The singular vectors of the network remain static across all iterations, as suggested by the singular vector stationary set, regardless of the learning rate. The angle between the true singular vectors and those of the network remains aligned throughout. The first singular values undergo oscillations in the large $\eta$ regime, whereas they remain constant in the small $\eta$ regime.

**EOS regime.** In Propositions 3 and 4, we prove that both the unbalanced and balanced initializations considered here belong to this set respectively, with an illustration in Figure 3. Specifically, we show that the balanced initialization in (3) belongs to the singular vector stationary set for all $t \geq 0$, while the unbalanced initialization in (4) belongs to the set for all $t \geq 1$ (far before entering the EOS regime) with $\mathbf{Q}_\ell = \mathbf{V}_\star$, allowing us to consider the loss in Equation (5).

Equipped with the loss in (5), notice that the balanced initialization in Equation (3) makes the learning dynamics such that for all $t \geq 0$,

$$\sigma_i(\mathbf{W}_\ell(t)) = \sigma_i(\mathbf{W}_k(t)), \quad \forall i \in [d], \quad \forall \ell, k \in [L],$$

where $\sigma_i(\mathbf{W}_\ell)$ denotes the $i$-th singular value of the $\ell$-th layer. This allows us to couple the dynamics and analyze the behavior of a single variable: $\sigma_i(\mathbf{\Sigma}_{L:1}(t)) = \sigma_i^L(t)$. However, this is certainly not the case for the unbalanced initialization in Equation (4). Since the singular values of the $\mathbf{W}_L(0)$ are initialized to zero, there is a non-negligible gap of $\alpha > 0$ between the singular values of the $L$-th layer and the other layers. However, in the following result, we prove that as long as $\alpha$ is small, GD will monotonically decrease the gap to balance the singular values across layers at the EOS. This will allow us to couple the dynamics in the limiting case for the unbalanced initialization as well.

**Proposition 2** (Balancing of Singular Values). *Let $\sigma_{\star,i}$ and $\sigma_{\ell,i}(t)$ denote the $i$-th singular value of $\mathbf{M}_\star \in \mathbb{R}^{d \times d}$ and $\mathbf{W}_\ell(t)$, respectively and define $S_i := L\sigma_{\star,i}^{2-\frac{2}{L}}$. Consider GD on the $i$-th index of the simplified loss in (5) with the unbalanced initialization and learning rate $\frac{2}{S_i} < \eta < \frac{2\sqrt{2}}{S_i}$. If the initialization scale $\alpha$ satisfies $0 < \alpha < \left( \ln\left( \frac{2\sqrt{2}}{\eta S_i} \right) \cdot \frac{\sigma_{\star,i}^{4/L}}{L^2 \cdot 2^{\frac{2L-3}{L}}} \right)^{1/4}$, then there exists a constant $c \in (0,1]$ such that for all $\ell \in [L-1]$, we have $\left| \sigma_{L,i}^2(t+1) - \sigma_{\ell,i}^2(t+1) \right| < c \cdot \left| \sigma_{L,i}^2(t) - \sigma_{\ell,i}^2(t) \right|$.*

The proof is available in Appendix C.2.2. This result has been shown to hold similarly for two-layer matrix factorization (Wang et al., 2022a; Ye & Du, 2021; Chen & Bruna, 2023), and our analysis extends it to the deeper case. Precisely, it considers the scalar loss for a singular value index and states that, as long as $\alpha$ is chosen below a threshold dependent on $\sigma_{\star,i}$, the $i$-th singular value across layers will become increasingly balanced. At the steady state limit of EOS, this balancing gap will decrease to zero. While this result is presented as a tool for the main result, it also has interesting implications for the dynamics of GD at EOS.

Firstly, it is well known that for gradient flow (GF), the balancing gap is conserved throughout its trajectory (see Lemma 2 and Figure 4a). While GD with small learning rates approximately conserves the gap, GD at the EOS (i.e., a learning rate close but below the stability limit) breaks this conservation. However, for a learning rate below the stability limit $2/\|\nabla^2 f(\mathbf{\Theta})\|_2$, the GD iterates may converge to an unbalanced global minimum (e.g., Figure 4b, and the balancing gap will

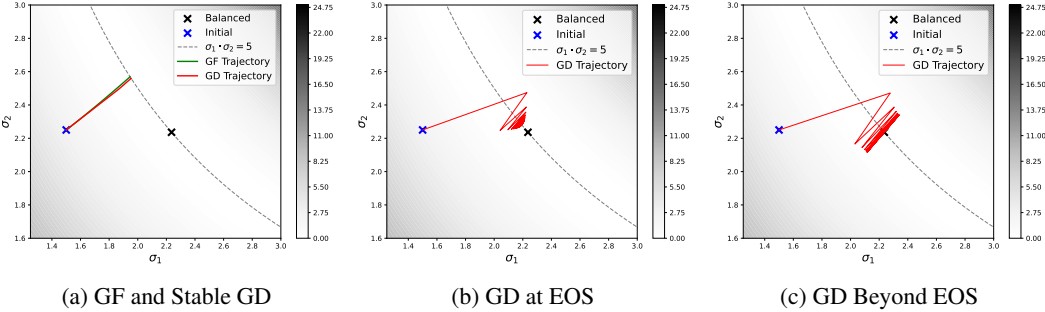

(a) GF and Stable GD  (b) GD at EOS  (c) GD Beyond EOS

Figure 4: Illustration of the GD trajectories for three different learning rates regimes for minimizing the function $f(\sigma_1, \sigma_2) = \frac{1}{2}(\sigma_2 \cdot \sigma_1 - \sigma_*)^2$, starting from an unbalanced initial point. Gradient flow conserves the balancing gap $|\sigma_1^2(t) - \sigma_2^2(t)|$ throughout its trajectory. GD at EOS decreases the gap, but stagnates once the oscillations no longer occur. GD beyond EOS decreases the gap monotonically to zero by oscillating towards and about the balanced minimum.

cease to decrease any further. In contrast, for GD beyond the stability limit (i.e., beyond the EOS), Proposition 2 states that the balancing gap decreases monotonically to zero. This highlights a key distinction between the learning dynamics of GF, GD, and GD at EOS. To illustrate these differences, we consider a toy example of minimizing a two-layer scalar function $f(\sigma_1, \sigma_2) = \frac{1}{2}(\sigma_2 \cdot \sigma_1 - \sigma_*)^2$, where $\sigma_* = 5$ in Figure 4. Once stable GD arrives at an unbalanced global minimum, it settles there and the balancing gap do not decrease further. For GD just below the stability limit (Figure 4b), the iterates oscillate, but once they cease oscillating and settle at an unbalanced global minimum, the gap also stagnates. On the other hand, GD beyond EOS (Figure 4c) drives the balancing gap strictly to zero, as the GD iterates oscillate toward and around the balanced minimum.

Secondly, for deep matrix factorization, we prove that the balanced minimum (i.e., the minimum where all of the singular values across layers are the same) corresponds to the flattest minimum (see Lemma 4). Since GD at EOS monotonically decreases the balancing gap, this also implies that GD implicitly walks from a sharper minima to the flattest minima. This also suggests an algorithmic trick: one can initially use a large learning rate to oscillate toward a flatter region and subsequently decrease the learning rate to settle at a flat minimum, as also highlighted by Chen & Bruna (2023).

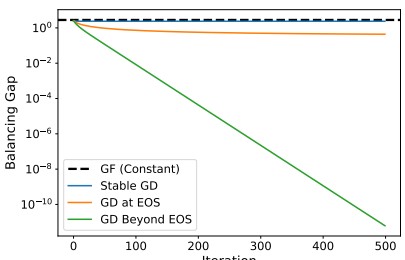

Figure 5: Plot of $|\sigma_1^2(t) - \sigma_2^2(t)|$ on a toy example, showing a decaying balancing gap beyond EOS.

Next, note that if the constant in Proposition 2 were to be strictly $c < 1$, by Lemma 10, the gap would approach zero infinitesimally. Our analysis shows the existence of $c$ in two cases: (i) $\sigma_i(\boldsymbol{\Sigma}_{L:1}) < \sigma_{\star,i}$ and (ii) $\sigma_i(\boldsymbol{\Sigma}_{L:1}) > \sigma_{\star,i}$. While we provably show that $c < 1$ for the first case, we have that $c = 1$ for the second case. This implies that when the GD iterates are below $(\sigma_i(\boldsymbol{\Sigma}_{L:1}) < \sigma_{\star,i})$ and approaching the minima, the balancing gap will monotonically decrease, but is not guaranteed to decrease to zero when we overshoot above the minima $(\sigma_i(\boldsymbol{\Sigma}_{L:1}) > \sigma_{\star,i})$. However, note that in the EOS regime, we oscillate below and above the minima as shown in Figure 4 (since for the case $\sigma_i(\boldsymbol{\Sigma}_{L:1}) < \sigma_{\star,i}$, we have $c < 1$). This indicates that we alternate between the two cases, and hence, the balancing gap will overall decrease to zero as depicted in Figure 5. Since oscillations do not occur or are not sustained in GF and stable GD, the gap does not go to zero in most cases, making this a distinct characteristic of GD beyond EOS.

Finally, we remark that Proposition 2 considers only the loss of a single singular value index, whereas Equation (5) is the sum over multiple indices. For Proposition 2 to hold for all indices, we can simply choose $\alpha$ with $\sigma_{\star,1}$ such that it is the smallest $\alpha$ satisfying the condition for all singular values $\sigma_{\star,i}$. To this end, in the following sections, we rigorously analyze the behavior of singular value oscillations around the balanced minimum. This can be viewed as the behavior of GD in the steady-state limit, as Proposition 2 implies that the singular values become balanced as $t \to \infty$.

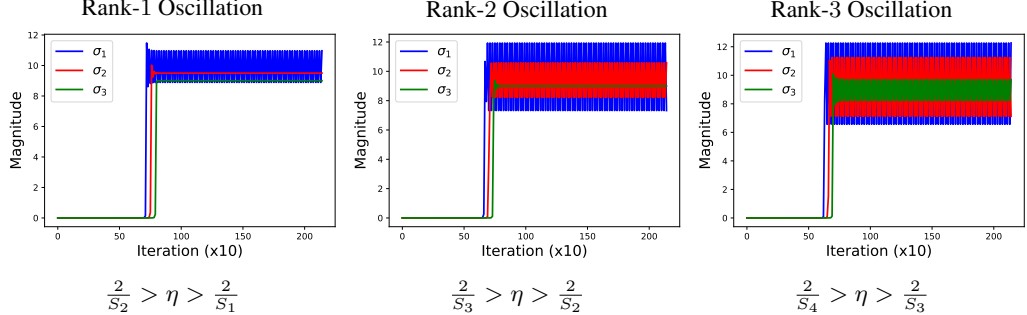

Figure 6: Evolution of the singular values of the end-to-end 3-layer network for fitting a rank-3 target matrix with singular values 10, 9.5, and 9. We use a learning rate of $\eta = 2/S_i$ with $S_i := L\sigma_{\star,i}^{2-2/L}$. The oscillations occur as a two-period orbit about the balanced minimum exactly with learning rate ranges specified in Theorem 1 for rank-$p$ oscillations ($p = 1, 2, 3$).

## 3.2 MAIN RESULTS

Using our analytical tools, we present our main results describing the learning dynamics of DLNs about the balanced solution beyond the EOS. First, we present a result characterizing the set of all eigenvalues $\lambda_{\Theta}$ of the DLN with respect to the flattened Hessian of the training loss.

**Lemma 1** (Eigenvalues of Hessian at the Balanced Minimum). *The set of all non-zero eigenvalues of the training loss Hessian of the deep matrix factorization loss $f(\Theta)$ defined in Equation (1) at the balanced minimum is given by*

$$\lambda_{\Theta} = \left\{ L\sigma_{\star,i}^{2-\frac{2}{L}}, \sigma_{\star,i}^{2-\frac{2}{L}} \right\}_{i=1}^{r} \bigcup \left\{ \sum_{\ell=0}^{L-1} \left( \sigma_{\star,i}^{1-\frac{1}{L}-\frac{1}{L}\ell} \cdot \sigma_{\star,j}^{\frac{1}{L}\ell} \right)^2 \right\}_{i \neq j}^{r} \bigcup \left\{ \sum_{\ell=0}^{L-1} \left( \sigma_{\star,k}^{1-\frac{1}{L}-\frac{1}{L}\ell} \cdot \alpha^\ell \right)^2 \right\}_{k=1}^{r}$$

*where $\sigma_{\star,i}$ is the $i$-th singular value of the target matrix $\mathbf{M}_\star \in \mathbb{R}^{d \times d}$, $\alpha \in \mathbb{R}$ is the initialization scale, $L$ is the depth of the network, and the second element of the set has a multiplicity of $d - r$.*

The proof is deferred to Appendix C.3.2. Let $\lambda_i$ denote the $i$-th largest eigenvalue of the Hessian. By Lemma 1, we observe that the sharpness is equal to $\lambda_1 = \|\nabla^2 f(\Theta)\|_2 = L\sigma_{\star,1}^{2-\frac{2}{L}}$ at the balanced minimum. In Lemma 4, we show that among all the points on the global minima, the sharpness at the balanced minimum is the smallest. Thus, if $\eta$ is set such that $\eta > 2/\lambda_1$, oscillations in the loss will occur, as the step size is large enough to induce oscillations even in the flattest region. Notice that this was alluded to in Figure 4—for GD beyond EOS (i.e., when $\eta > 2/\lambda_1$), there is stable oscillation around the minima, whereas for GD at EOS, the iterates eventually settle down after transient oscillations. Furthermore, notice that all non-zero eigenvalues are a function of network depth. For a deeper network, the sharpness will be larger, implying that a smaller learning rate can be used to drive the DLN into EOS. This provides a unique perspective on how the learning rate should be chosen as networks become deeper and explains the observation made by Cohen et al. (2021), who observed that sharpness scales with the depth of the network. With the eigenvalues, we show in the following result that oscillations occur in a two-period orbit about the balanced minimum within a rank-$p$ subspace, where the rank is dependent on the learning rate.

**Theorem 1** (Rank-$p$ Periodic Subspace Oscillations). *Let $\mathbf{M}_\star = \mathbf{U}_\star \mathbf{\Sigma}_\star \mathbf{V}_\star^\top$ denote the SVD of the target matrix and define $S_p := L\sigma_{\star,p}^{2-\frac{2}{L}}$ and $K_p' := \max\left\{ S_{p+1}, \frac{S_p}{2\sqrt{2}} \right\}$. If we run GD on the deep matrix factorization loss with learning rate $\eta = \frac{2}{K}$, where $K_p' < K < S_p$, then the top-$p$ singular values of the end-to-end DLN oscillate in a 2-period orbit ($j \in \{1, 2\}$) around the balanced minimum and admit the following decomposition:*

$$\mathbf{W}_{L:1} = \underbrace{\sum_{i=1}^{p} \rho_{i,j}^{L} \cdot \mathbf{u}_{\star,i} \mathbf{v}_{\star,i}^\top}_{\text{oscillation subspace}} + \underbrace{\sum_{k=p+1}^{d} \sigma_{\star,k} \cdot \mathbf{u}_{\star,k} \mathbf{v}_{\star,k}^\top}_{\text{stationary subspace}}, \quad j \in \{1, 2\} \tag{6}$$

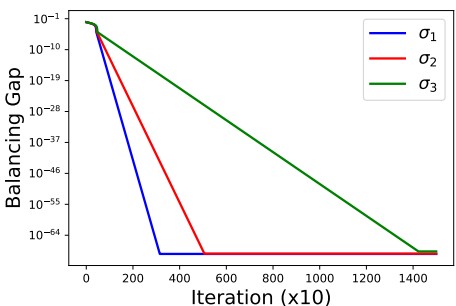 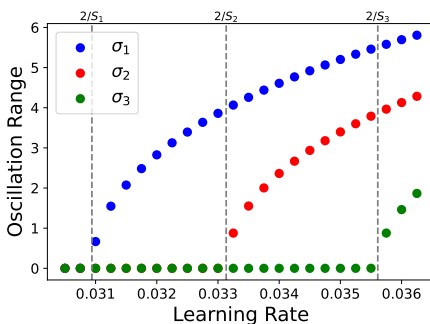

Figure 7: Experimental results on a depth-3 DLN with target singular values $10, 9.5, 9$. Left: Plot of the balancing gap decaying monotonically to zero as the learning rate is chosen $\eta > 2/S_3$. Right: Plot of the oscillation range as a function of the learning rate. As the learning rate increases, the oscillation ranges also increase.

where $\rho_{i,1} \in \left(0, \sigma_{\star,i}^{1/L}\right)$ and $\rho_{i,2} \in \left(\sigma_{\star,i}^{1/L}, (2\sigma_{\star,i})^{1/L}\right)$ are the two real roots of the polynomial $g(\rho_i) = 0$ and

$$g(\rho_i) = \rho_i^L \cdot \frac{1 + \left(1 + \eta L(\sigma_{\star,i} - \rho_i^L) \cdot \rho_i^{L-2}\right)^{2L-1}}{1 + \left(1 + \eta L(\sigma_{\star,i} - \rho_i^L) \cdot \rho_i^{L-2}\right)^{L-1}} - \sigma_{\star,i}.$$

The proof is available in Appendix C.3.3. Theorem 1 explicitly identifies the subspaces that exhibit a two-period orbit based on the range of the learning rate. It also provides a rough characterization of the oscillation amplitude, which is determined by $\rho_{i,1}$ and $\rho_{i,2}$—values below and above the balanced minimum, respectively. Since there is no closed-form solution for an arbitrary higher-order polynomial, $\rho_{i,1}$ and $\rho_{i,2}$ are defined as solutions to the polynomial $g(\rho_i)$. Overall, this aims to theoretically explain why (i) oscillations occur primarily within the top subspaces of the network, as observed by Zhu et al. (2023a), and (ii) oscillations are more pronounced in the directions of stronger features, as measured by the magnitudes of their singular values.

Notice that the range of the learning rate depends on the eigenvalues of the form $S_p = L\sigma_{\star,p}^{2-2/L}$ rather than on all eigenvalues in Lemma 1. This is because the eigenvectors associated with the other eigenvalues are orthogonal to the weights of the DLN at the balanced minimum, so oscillations will never occur in those particular eigendirections. They are only non-orthogonal in the directions of the eigenvalues of $S_p$ and, hence, oscillations occur only in those specific directions.

We also remark that our result generalizes the recent theoretical findings of Chen & Bruna (2023), where they proved the existence of a certain class of scalar functions $\tilde{f}(x)$ for which GD does not diverge even when operating beyond the stability threshold. They demonstrated that there exists a range in which the loss oscillates around the local minima with a certain periodicity. These oscillations gradually progress into higher periodic orbits (e.g., 2, 4, 8 periods), transition into chaotic behavior, and ultimately result in divergence. In our work, we prove that this oscillatory behavior beyond the stability threshold also occurs in DLNs.

## 4 EXPERIMENTAL RESULTS

### 4.1 SUBSPACE OSCILLATIONS IN DEEP NETWORKS

Firstly, we provide experimental results corroborating Theorem 1. We let the target matrix be $\mathbf{M}_\star \in \mathbb{R}^{50 \times 50}$ with rank 3, with dominant singular values $\sigma_\star = 10, 9.5, 9$. For the DLN, we consider a 3-layer network, with each layer as $\mathbf{W}_\ell \in \mathbb{R}^{50 \times 50}$ and use an initialization scale of $\alpha = 0.01$. In Figure 6, we present the behaviors of the singular values of the end-to-end network under different learning rate regimes. Recall that by Theorem 1, the $i$-th singular value undergoes periodic oscillations when $K$ is set to be $S_i < K < S_{i+1}$, where $S_i = L\sigma_{\star,i}^{2-2/L}$. Figure 6 illustrates this clearly – we only observe oscillations in the $i$-th coordinate depending on the learning rate.

Interestingly, notice that $\sigma_2$ also begins to oscillate in the rank-1 oscillation region before settling at a minimum. This occurs because, while the learning rate is large enough to catapult around an unbalanced minimum, it is not sufficiently large to induce periodic oscillations at balanced minima.

Secondly, in Figure 7, we present an experiment demonstrating the relationship between the oscillation range and the learning rate by plotting the amplitude of singular value oscillations in the end-to-end network, as well as the balancing gap, to corroborate Proposition 2 in DLNs. Clearly, in Figure 7 (left), we observe that the balancing gap decays monotonically to zero as long as the learning rate is chosen such that periodic oscillations occur in the top-3 subspaces. In Figure 7 (right), the oscillations begin to occur starting from each region $\eta = 2/S_i$, and the oscillation range (or amplitude) increases as the learning rate increases. This can also be observed in Figure 6; the amplitude of $\sigma_1$ increases as we move from the rank-1 to the rank-3 oscillation region.

## 4.2 SIMILARITIES AND DIFFERENCES BETWEEN LINEAR AND NONLINEAR NETS AT EOS

**Mild Sharpening.** "Mild" sharpening refers to the sharpness not rising to $2/\eta$ throughout learning, and generally occurs in tasks with low complexity as discussed in Caveat 2 of (Cohen et al., 2021). We illustrate mild sharpening in Figure 10, where we plot sharpness in two settings: (i) regression with simple images and (ii) classification with an MLP using a subset of the CIFAR-10 dataset.

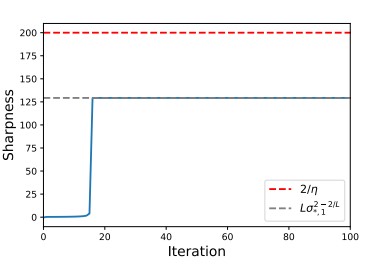

Figure 8: DLNs do not enter EOS regime if $L\sigma_1^{2-\frac{2}{L}} < 2/\eta$.

For the regression task, we minimize the loss $\mathcal{L}(\Theta) = \|G(\Theta) - \mathbf{y}_{\text{image}}\|_2^2$, where $G(\Theta)$ is a UNet parameterized by $\Theta$, and $\mathbf{y}_{\text{image}}$ denotes one of the images in Figure 10b. We observe that when $\mathbf{y}_{\text{image}}$ is a smooth, low-frequency image, the sharpness of the loss generally remains low. However, when $\mathbf{y}_{\text{image}}$ has higher frequency content, the sharpness increases and enters the EOS regime (Figure 10a). Similarly, for the classification task, we train a 2-layer fully connected neural network on $N$ labeled training images from the CIFAR-10 dataset using MSE loss and plot the sharpness in Figure 10c. The sharpness links to $N$, the number of data points used for training. For small $N$ values, such as 100 or 200, the network learns only a limited set of latent features, resulting in mild sharpening, and it does not reach the EOS threshold. However, when $N$ exceeds 1000, the sharpness increases and reaches the EOS threshold. Similar observations can also be seen in DLNs. In Figure 8, we show that the sharpness reaches $L\sigma_{\star,1}^{2-\frac{2}{L}}$, where $\sigma_{\star,1}$ is the singular value of the target matrix. Whenever $L\sigma_{\star,1}^{2-\frac{2}{L}} < 2/\eta$, the network will not enter the EOS regime. This can be viewed as low-complexity learning, as $\sigma_{\star,1}$ corresponds to the magnitude of the strongest feature of the target matrix. Hence, when $\sigma_{\star,1}$ is not large enough, the sharpness will not rise to $2/\eta$. While this cannot fully explain mild sharpening, our experiments demonstrate that interpreting sharpness as a measure of complexity, combined with our findings from DLNs, marks an important first step toward fully understanding this phenomenon.

**Difference in Oscillation Behaviors.** Here, we discuss the differences in oscillations that arise in DLNs compared to catapults that occur in practical deep nonlinear networks. The main difference lies in the loss landscape—at convergence, the Hessian for DLNs is positive semi-definite, as shown in Lemma 1, meaning there are only directions of positive curvature and flat directions (in the null space of the Hessian). Moreover, the loss landscape of DLNs are known to be benign since they do not contain any spurious local minima, but only saddle points and global minima (Kawaguchi

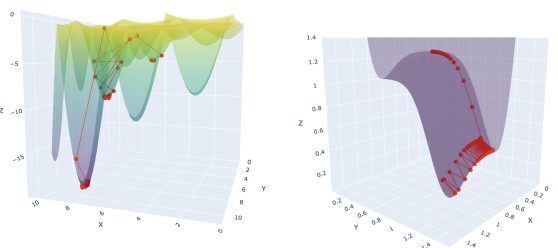

Figure 9: Loss landscape of the Holder table function and DLNs, respectively (left–right). The Holder table function is non-convex which allows catapulting to other minima, whereas DLNs do not have spurious local minima.

(2016)). In this landscape, oscillations occur because the basin walls bounce off, without the direction of escape. However, in deep nonlinear networks, it has been frequently observed that the

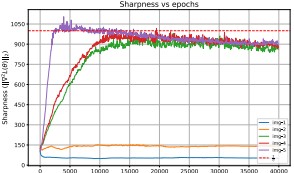
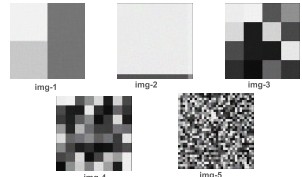
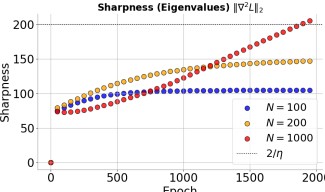

(a) Sharpness plots for training image generator networks using SGD with learning rate $\eta = 2 \times 10^{-4}$.

(b) Target images (denoted as $\mathbf{y}_{\text{image}}$) with different frequencies used for training.

(c) 2-layer FC network trained with small number $N$ of CIFAR-10 dataset with $\eta = 10^{-2}$

Figure 10: Illustration of Caveat 2 by Cohen et al. (2021) on how mild sharpening occurs on simple datasets and network. (a) Regression task showing the evolution of the sharpness when an UNet (with fixed initialization) is trained to fit a single image shown in (b). (c) Evolution of the minimal progressive sharpening on a classification task of a 2-layer MLP trained on a subset of CIFAR-10.

Hessian at the minima has negative eigenvalues (Ghorbani et al., 2019; Sagun et al., 2016). This enables an escape direction along the negative curvature, preventing sustained oscillations.

In Figure 9, we demonstrate these two differences by visualizing the loss landscapes and the iterates throughout GD marked in red. The Holder table function Figure 9 (left) exhibits numerous local minima, causing the loss to exhibit a sharp "catapult" when a large learning rate is used. In contrast, for DLNs (shown in the right) the loss oscillates in a periodic orbit around the global minima since there are no spurious local minima (Ge et al., 2016; Kawaguchi, 2016; Lu & Kawaguchi, 2017; Zhang, 2019; Yun et al., 2019).

Lastly, Damian et al. (2023) study self-stabilization, where sharpness decreases below $2/\eta$ after initially exceeding $2/\eta$. Their analysis requires assumptions such as $\nabla L(\theta) \cdot u(\theta) = 0$ and $\nabla S(\theta)$ lies in the null space of the Hessian, where $S(\theta)$ and $u(\theta)$ denotes the sharpness and its corresponding eigenvector, respectively. These assumptions do not hold exactly in DLNs. Rather, the sharpness oscillates about $2/\eta$ as shown in Figure 2 as the condition for stable oscillation holds along each eigenvector of the Hessian. The alignment of $\nabla L(\theta)$, $u(\theta)$ and $\nabla S(\theta)$ determines the nature of oscillations in deep networks or its absence thereof. This alignment usually depends on the symmetry of the parameter space and can usually vary across different architectural components. This work deals with deep linear networks which has rescaling symmetry, however several other symmetries (Kunin et al. (2021)) can be induced by softmax operator (translation symmetry) or batch-normalization (scaling symmetry), which may further affect these alignments. We leave this study for future work.

## 5 CONCLUSION AND LIMITATIONS

In this paper, we presented a fine-grained analysis of the learning dynamics of deep matrix factorization beyond the EOS, where our analysis revealed a two-period orbit within a small subspace around the balanced minimum. We showed that as long as oscillations were sustained, the balancing gap, defined as the difference in singular values across layers, decreases monotonically to zero. For DLNs, since the flattest minima correspond to the minima where all weights are balanced, this suggests an implicit walk toward flat minima without any explicit regularization, which is a distinct characteristic of GD beyond the EOS. Our results also contributed to understanding unexplained phenomena in nonlinear networks within EOS, such as mild sharpening or oscillations in a small subspace.

Since our analysis focuses on the behavior of DLNs around the balanced minima starting from an unbalanced initialization, it technically describes a steady-state limiting behavior of GD once the singular values become balanced. To fully capture the learning dynamics of DLNs, it is of great interest to derive the complete dynamics at EOS, where oscillations occur but are not sustained. Furthermore, we focused on cases where the weights belonged to the singular vector stationary set, allowing us to isolate the behavior of singular vectors from singular values. It is also of great interest to account for how singular vectors align beyond the EOS regime, as this is currently beyond the scope of our paper.

## 6 ACKNOWLEDGMENTS

AG, SR and RR acknowledge support from NSF CCF-2212065. QQ and SMK acknowledges support from NSF CAREER CCF-2143904, NSF CCF-2212066, and NSF IIS 2312842.

We thank Arthur Jacot, Nicholas Flamarrion, Sadhika Malladi, Rene Vidal and Abhishek Panigrahi for their valuable feedback. We are also grateful to Sungyoon Lee for technical discussions on balancing proof. Special thanks are extended to Molei Tao for his thoughtful contributions, which clarified the nuances between the EOS and sustained oscillations. We appreciate Eshaan Nichani for directing our attention to the work of Kreisler et al. (2023), which was instrumental in our analysis of balancing using gradient flow sharpness. Finally, we thank Jeremy Cohen for his support, from early email exchanges in 2023 to in-person discussions at NeurIPS 2023, regarding self-stabilization and mild sharpening.

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

# Appendix

CONTENTS

## A  DISCUSSION ON RELATED WORK

**Implicit Bias of Edge of Stability.**  Edge of stability was first coined by Cohen et al. (2021), where they showed that the Hessian of the training loss plateaus around $2/\eta$ when deep models were trained using GD. However, Jastrzebski et al. (2020); Jastrzebski et al. (2019) previously demonstrated that the step size influences the sharpness along the optimization trajectory. Due to the important practical implications of the edge of stability, there has been an explosion of research dedicated to understanding this phenomenon and its implicit regularization properties. Here, we survey a few of these works. Damian et al. (2023) explained edge of stability through a mechanism called "self-stabilization", where they showed that during the momentary divergence of the iterates along the sharpest eigenvector direction of the Hessian, the iterates also move along the negative direction of the gradient of the curvature, which leads to stabilizing the sharpness to $2/\eta$. Agarwala et al. (2023) proved that second-order regression models (the simplest class of models after the linearized NTK model) demonstrate progressive sharpening of the NTK eigenvalue towards a slightly different value than $2/\eta$. Arora et al. (2022) mathematically analyzed the edge of stability, where they showed that the GD updates evolve along some deterministic flow on the manifold of the minima. Lyu et al. (2022) showed that the normalization layers had an important role in the edge of stability – they showed that these layers encouraged GD to reduce the sharpness of the loss surface and enter the EOS regime. Ahn et al. (2024) established the phenomenon in two-layer networks and find phase transitions for step-sizes in which networks fail to learn "threshold" neurons. Wang et al. (2022b) also analyze a two-layer network, but provide a theoretical proof for the change in sharpness across four different phases. Even et al. (2024) analyzed the edge of stability in diagonal linear networks and found that oscillations occur on the sparse support of the vectors. Lastly, Wu et al. (2024) analyzed the convergence at the edge of stability for constant step size GD for logistic regression on linearly separable data.

**Edge of Stability in Toy Functions.**  To analyze the edge of stability in slightly simpler settings, many works have constructed scalar functions to analyze the prevalence of this phenomenon. For example, Chen & Bruna (2023) studied a certain class of scalar functions and identified conditions in which the function enters the edge of stability through a two-step convergence analysis. Wang et al. (2023) showed that the edge of stability occurs in specific scalar functions, which satisfies certain regularity conditions and developed a global convergence theory for a family of non-convex functions without globally Lipschitz continuous gradients. Zhu et al. (2023b) analyzed local oscillatory behaviors for 4-layer scalar networks with balanced initialization. Song & Yun (2023); Kalra et al. (2023) provide analyses of learning dynamics at the EOS in simplified settings such as two-layer networks. Zhu et al. (2022); Chen et al. (2023) study GD dynamics for quadratic models in large learning rate regimes. Overall, all of these works showed that the necessary condition for the edge of stability to occur is that the second derivative of the loss function is non-zero, even though they assumed simple scalar functions. Our work takes one step further to analyze the prevalence of the edge of stability in DLNs. Although our loss simplifies to a loss in terms of the singular values, they precisely characterize the dynamics of the DLNs for the deep matrix factorization problem.

**Deep Linear Networks.**  Over the past decade, many existing works have analyzed the learning dynamics of DLNs as a surrogate for deep nonlinear networks to study the effects of depth and implicit regularization (Saxe et al., 2014; Arora et al., 2018; 2019; Alkhouri et al., 2024). Generally, these works focus on unveiling the dynamics of a phenomenon called "incremental learning", where small initialization scales induce a greedy singular value learning approach (Min Kwon et al., 2024; Gissin et al., 2020; Saxe et al., 2014), analyzing the learning dynamics via gradient flow (Saxe et al., 2014; Chou et al., 2024a; Arora et al., 2019), or showing that the DLN is biased towards low-rank solution (Yaras et al., 2024; Arora et al., 2019; Min Kwon et al., 2024), amongst others. However, these works do not consider the occurence of the edge of stability in such networks. On the other hand, while works such as those by Yaras et al. (2024) and Min Kwon et al. (2024) have similar observations in that the weight updates occur within an invariant subspace as shown by Proposition 3, they do not analyze the edge of stability regime.

# B ADDITIONAL RESULTS

## B.1 EXPERIMENTAL DETAILS

**Bifurcation Plot.** In this section, we provide additional details regarding the experiments used to generate the figures in the main text. For Figure 1, we consider a rank-3 target matrix $\mathbf{M}_\star \in \mathbb{R}^{5 \times 5}$ with ordered singular values $10, 6, 3$. We use a 3-layer DLN to fit the target matrix. Since $\sigma_{\star,1} = 10$, the network enters the EOS regime at

$$\eta = \frac{2}{L\sigma_{\star,1}^{2-2/L}} = 0.0309.$$

We show that there exists a two-period orbit after $0.0309/2 = 0.0154$, as we do not have a scaling of $1/2$ in the objective function for the code used to generate the figures.

**Contour Plots.** In Figure 4, we considered the toy example

$$f(\sigma_1, \sigma_2) = \frac{1}{2}(\sigma_2 \cdot \sigma_1 - \sigma_*)^2,$$

which corresponds to a scalar two-layer network. By Lemma 1, the stability limit is computed as $\eta = 0.2$, as $L = 2$ and $\sigma_\star = 5$. To this end, for GD beyond EOS, we use a learning rate of $\eta = 0.2010$, where as we use a learning rate of $\eta = 0.1997$ for GD at EOS. For GF, we plot the conservation flow, and use a learning rate of $\eta = 0.1800$ for stable GD.

**DLN and Holder Table Function Plots.** In Figure 9 and 11, we compared the landscape of DLNs with that of a more complicated non-convex function such as the Holder table function. To mimic the DLN, we considered the loss function

$$z = L(x, y) = (x^4 - 0.8)^2 + (y^4 - 1)^2, \tag{7}$$

which corresponds to a 4-layer network. Here the eigenvector of the Hessian at the global minima coincides with the $x, y$-axis. We calculate the eigenvalues $\lambda_1$ and $\lambda_2$ at the minimum $(0.8^{0.25}, 1)$ and plot the dynamics of the iterates for step size range $\frac{2}{\lambda_2} > \eta > \frac{2}{\lambda_1}$ and $\eta > \frac{2}{\lambda_2}$. When $\frac{2}{\lambda_2} > \eta > \frac{2}{\lambda_1}$ the $x$-coordinate stays fixed at the minima $0.8^{0.25}$ and the $y$-coordinate oscillates around its minimum at $y = 1$. This is evident in the landscape figure. Similarly, when $\eta > \frac{2}{\lambda_2}$, oscillations occur in both the $x$ and $y$ direction. The loss landscape $z = L(x, y)$ does not have spurious local minima, so sustained oscillations take place in the loss basin.

For the non-convex landscape as shown in Figure 9 and 12, we consider the Holder table function:

$$f(x, y) = - \left| \sin(x)\cos(y) \exp\left(1 - \frac{\sqrt{x^2 + y^2}}{\pi}\right) \right|.$$

By observation, we initialize near a sharp minima and run GD with an increasing learning rate step size as shown in the lefthand side of Figure 12. When the learning rate is fixed, we observe that oscillations take place inside the local valley, but when learning rate is increased, it jumps out of the local valley to find a flatter basin. Similar to the observations by Cohen et al. (2021), the sharpness of the GD iterates are "regulated" by the threshold $2/\eta$, as it seems to closely follow this value as shown in Figure 12.

Overall, these examples aim to highlight the difference in linear and complex loss landscapes. The former consists of *only* saddles and global minima, and hence (stably) oscillate about the global minimum. However, in more complicated non-convex landscapes, sharpness regularization due to large learning rates enable catapulting to flatter loss basins, where sharpness is smaller than $2/\eta$.

## B.2 INITIALIZATION OUTSIDE SINGULAR VECTOR INVARIANT SET

In this section, we present an initialization example that is outside the singular vector stationary set. We consider the following initialization:

$$\mathbf{W}_L(0) = \mathbf{0}, \qquad \mathbf{W}_\ell(0) = \alpha\mathbf{P}_\ell, \quad \forall\ell \in [L-1], \tag{8}$$

Oscillation along Y-axis: $2/\lambda_2 > \eta > 2/\lambda_1$

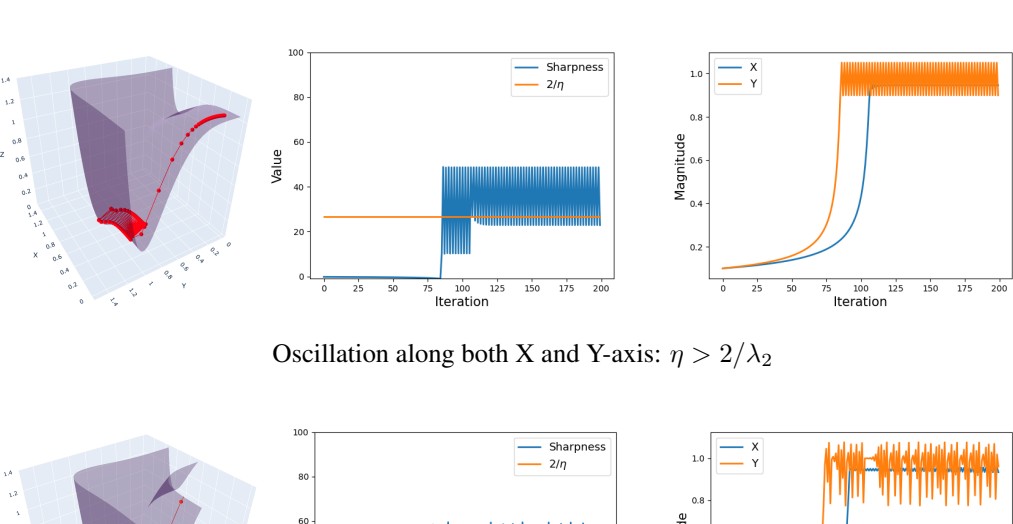

Oscillation along both X and Y-axis: $\eta > 2/\lambda_2$

Loss Landscape        Sharpness        Oscillatory Components

Figure 11: Demonstration of the EOS dynamics of a 2-dimensional depth-4 scalar network as shown in Equation (7). $X, Y$ axes are the eigenvectors of the Hessian with eigenvalues $\lambda_1$ and $\lambda_2$ respectively. Top: when $\eta > 2/\lambda_1$, the $X$ component remains fixed, while the $Y$ component oscillates with a periodicity of 2. Bottom: for $\eta > 2/\lambda_2$, the iterates oscillation in both directions.

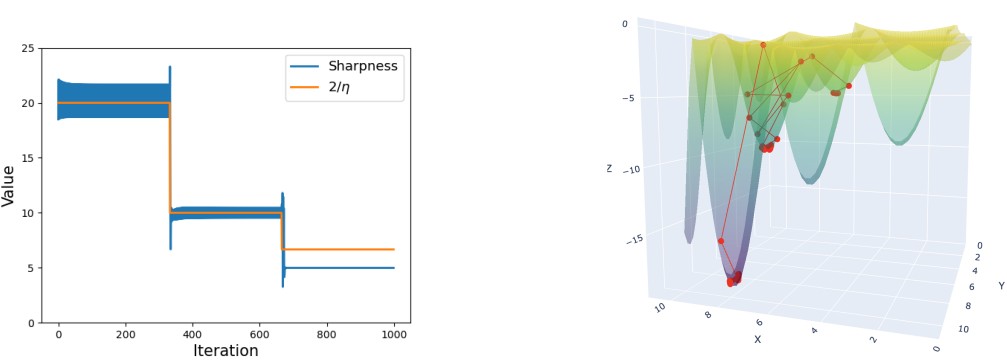

Figure 12: EOS dynamics at various step learning rates from the Holder table function. Left: plot of the learning rate steps and sharpness, showing that sharpness follows the EOS limit $2/\eta$. Right: Plot showing that the iterates catapult out of a local basin when the learning rate is increased and jumps out to a surface where the sharpness is about $2/\eta$.

where $\mathbf{P}_\ell \in \mathbb{R}^{d \times d}$ is an orthogonal matrix. Note that here for $\ell > 1$, the singular vectors do not align and lies outside the SVS set we defined in Proposition 3. We consider the deep matrix factorization problem with a target matrix $\mathbf{M}_\star \in \mathbb{R}^{d \times d}$, where $d = 100$, $r = 5$, and $\alpha = 0.01$. We empirically

obtain that the decomposition after convergence admits the form:

$$\mathbf{W}_L(t) = \mathbf{U}^\star \begin{bmatrix} \boldsymbol{\Sigma}_L(t) & \mathbf{0} \\ \mathbf{0} & \mathbf{0} \end{bmatrix} \left[ \left( \prod_{i=L-1}^{1} \mathbf{P}_i \right) \mathbf{V}^\star \right]^\top, \tag{9}$$

$$\mathbf{W}_\ell(t) = \left[ \left( \prod_{i=\ell}^{1} \mathbf{P}_i \right) \mathbf{V}^\star \right] \begin{bmatrix} \boldsymbol{\Sigma}_\ell(t) & \mathbf{0} \\ \mathbf{0} & \alpha \mathbf{I}_{d-r} \end{bmatrix} \left[ \left( \prod_{i=\ell-1}^{1} \mathbf{P}_i \right) \mathbf{V}^\star \right]^\top, \quad \forall \ell \in [2, L-1], \tag{10}$$

$$\mathbf{W}_1(t) = \mathbf{P}_1 \mathbf{V}^\star \begin{bmatrix} \boldsymbol{\Sigma}_1(t) & \mathbf{0} \\ \mathbf{0} & \alpha \mathbf{I}_{d-r} \end{bmatrix} \mathbf{V}^{\star\top}, \tag{11}$$

where $\mathbf{W}_L(0) = \mathbf{0}$ and $\mathbf{W}_\ell(0) = \alpha \mathbf{P}_l, \forall \ell \in [L-1]$. The decomposition after convergence lies in the SVS set as the singular vectors now align with each other. This demonstrates an example where even when the initialization is made outside the SVS set, GD aligns the singular vectors such that after certain iterations it lies in the SVS set.

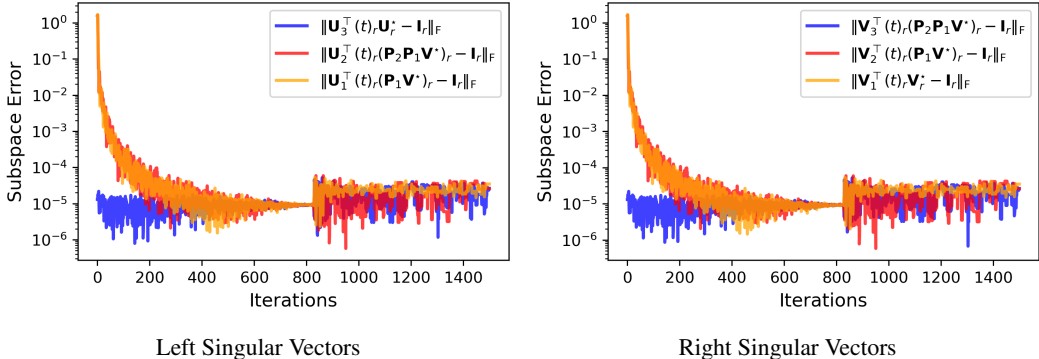

Left Singular Vectors          Right Singular Vectors

Figure 13: Empirical verification of the decomposition for initialization with orthogonal matrices (lying outside SVS set) in that after some GD iterations, the singular vectors of the intermediate matrices align to lie within SVS set, displaying singular vector invariance.

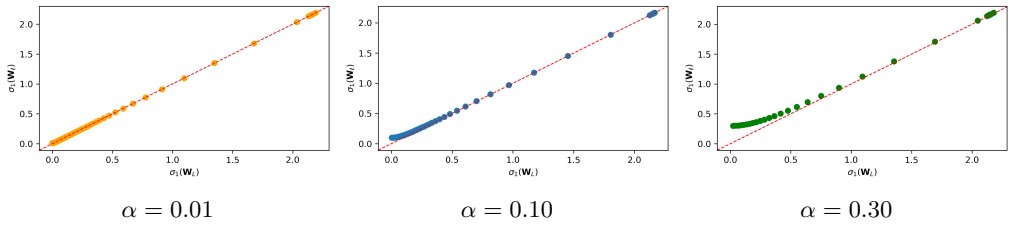

$\alpha = 0.01$        $\alpha = 0.10$        $\alpha = 0.30$

Figure 14: Observing the balancedness between the singular value initialized to $0$ and a singular value initialized to $\alpha$. The scattered points are successive GD iterations (going left to right). The initial gap between the two values is larger for a larger $\alpha$, but quickly gets closer over more GD iterations.

### B.3 ADDITIONAL EXPERIMENTS FOR BALANCING, SINGULAR VECTOR INVARIANCE, AND THEORY

Our theory relied on two tools and assumptions: balancing of singular values and stationarity of the singular vectors. In this section, we investigate how the dynamics at EOS are affected if these two assumptions do not hold.

**Balancing.** First, we present additional experimental results on Proposition 2 and how close the iterates become for different initialization scales. To this end, we consider the same setup from the previous section, where we have a target matrix $\mathbf{M}_\star \in \mathbb{R}^{d \times d}$, where $d = 100$, $r = 5$, and varying

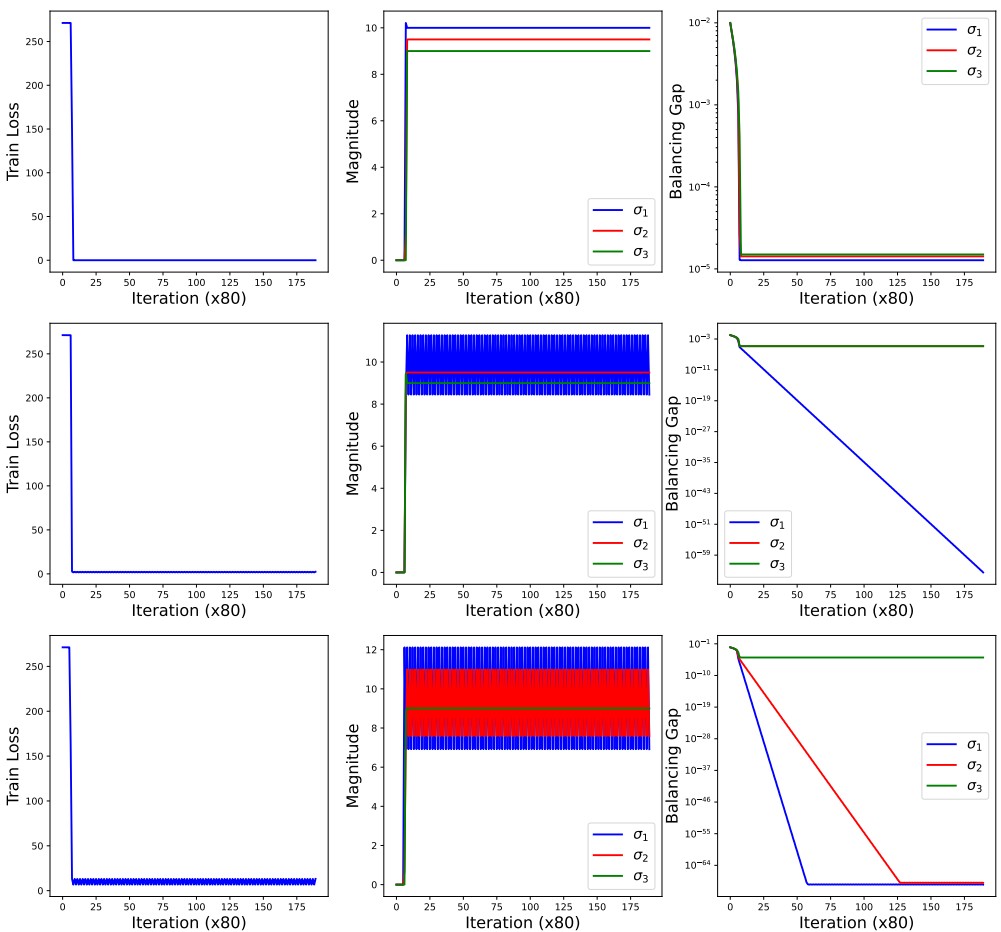

Figure 15: Plots of the training loss, singular value magnitude, and the balancing gap over iterations for different learning rates: $\eta = 0.030, 0.032, 0.034$ (top to bottom). When the learning rate is stable ($\eta < 0.031$ since the top singular value is $\sigma_{\star,1} = 10$), the balancing gap plateaus, whereas the balancing gap goes strictly to zero when the oscillations occur.

initialization $\alpha$. In Figure 14, we observe that for larger values of $\alpha$, the balancing quickly occurs, whereas for smaller values of $\alpha$, the balancing is almost immediate. This is to also highlight that our bound on $\alpha$ in Proposition 2 may be an artifact of our analysis, and can choose larger values of $\alpha$ in practice.

To this end, we also investigate how large $\alpha$ can be until Proposition 2 no longer holds. We consider the dynamics of a 3-layer DLN to fit a target matrix $\mathbf{M}_\star \in \mathbb{R}^{10 \times 10}$ of rank-3 with ordered singular values $10, 8, 6$. We use a learning rate of $\eta = 0.0166$, which corresponds to oscillations in the top-2 singular values. In Figure 16, we show the dynamics of when the initialization scale is $\alpha = 0.01$ and $\alpha = 0.5$, where balancing holds theoretically for the former but not for the latter. Clearly, we observe that balancing does not hold for $\alpha = 0.5$. However, examining the middle plots reveals that the oscillations in the singular values still have the same amplitude in both cases and for both singular values.

**Singular Vector Stationarity.** Throughout this paper, we considered two initializations in Equations (3) and (4), where balancing holds immediately and one where balancing holds for a sufficiently small initialization scale. In this section, we investigate different initializations with aim to observe (i) if they do not converge to the SVS set and (ii) how they affect the oscillations if they do

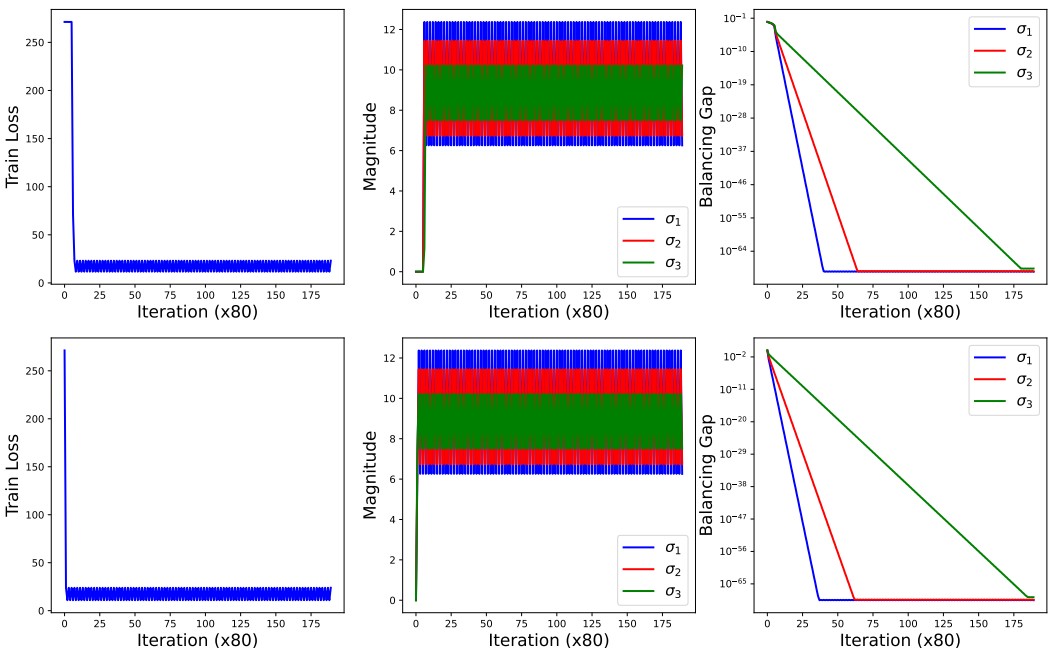

Figure 16: Top: EOS dynamics of a 3-layer DLN with initialization scale $\alpha = 0.01$, where balancing theoretically holds. Bottom: EOS dynamics of the DLN with initialization scale $\alpha = 0.5$. While the balancing does not hold for $\alpha = 0.5$, the oscillations in the singular values are still prevalent, with the same amplitude.

not belong to the SVS set. To this end, we consider the following:

$$\mathbf{W}_L(0) = \mathbf{0}, \quad \mathbf{W}_\ell(0) = \alpha\mathbf{I}_d, \quad \forall\ell \in [L-1], \qquad \text{(Original)}$$

$$\mathbf{W}_L(0) = \mathbf{0}, \quad \mathbf{W}_\ell(0) = \alpha\mathbf{P}_\ell, \quad \forall\ell \in [L-1], \qquad \text{(Orthogonal)}$$

$$\mathbf{W}_L(0) = \mathbf{0}, \quad \mathbf{W}_\ell(0) = \alpha\mathbf{H}_\ell, \quad \forall\ell \in [L-1], \qquad \text{(Random)}$$

where $\mathbf{P}_\ell$ is an orthogonal matrix and $\mathbf{H}_\ell$ is a random matrix with Gaussian entries. For all of these initialization schemes, we consider the same setup as in the balancing case, with an initialization scale of $\alpha = 0.01$. To observe if singular vector stationarity holds, we consider the subspace distance as follows:

$$\text{Subspace Distance} = \|\mathbf{U}_{\ell-1,r}^\top\mathbf{V}_{\ell,r} - \mathbf{I}_r\|_\mathsf{F}, \qquad (12)$$

where $\mathbf{U}_{\ell,r}$ and $\mathbf{V}_{\ell,r}$ are the top-$r$ left and right singular vectors of layer $\mathbf{W}_\ell$, respectively. Since Proposition 1 implies that the intermediate singular vectors cancel, the initialization converges to the SVS set if the subspace distance goes to zero. In Figure 17, we plot the dynamics for all of the initializations. Generally, we observe that the subspace distance for all cases go to zero, validating the use of the SVS set for analysis purposes.

**Additional Results.** In this section, we provide more experimental results to corroborate our theory. Recall that in Lemma 1, we proved that the learning rate needed to enter the EOS is a function of the depth, and that deeper networks can enter EOS using a smaller learning rate. To verify this claim, we provide an additional experiment where the target matrix is $\mathbf{M}_\star \in \mathbb{R}^{5\times 5}$ with the top singular value set to $\sigma_{\star,1} = 0.5$. We use an initialization scale of $\alpha = 0.01$. In Figure 18, we can clearly see that shallower networks need a larger learning rate, and vice versa to enter EOS. Here, black refers to stable learning and white refers to regions in which oscillations occur (EOS regime).

### B.4 PERIODIC AND FREE OSCILLATIONS

In this section, we present additional experiments on oscillation and catapults in both deep linear and nonlinear networks to supplement the results in the main paper. First, we consider a 3-layer MLP

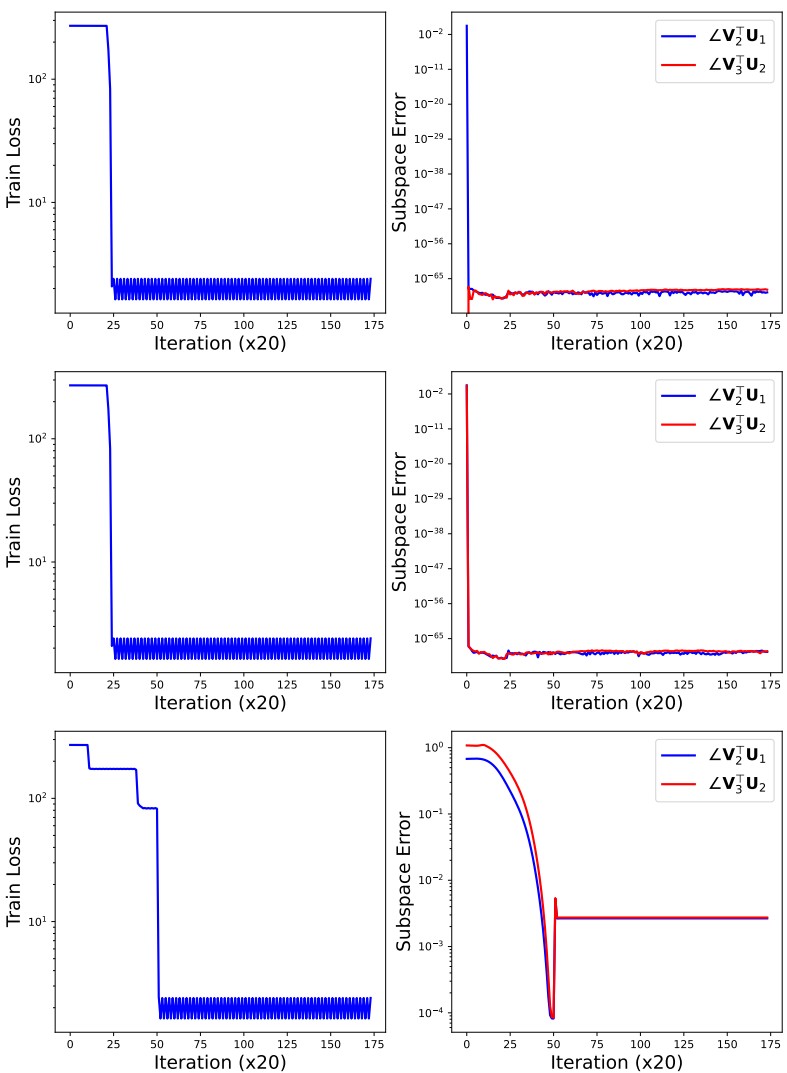

Figure 17: EOS dynamics of a 3-layer DLN for different initializations where it all converges to the SVS set. The subspace distance is defined in Equation (12). Top: Dynamics with the original identity initialization. Middle: Dynamics with orthogonal initialization. Bottom: Dynamics with random initialization.

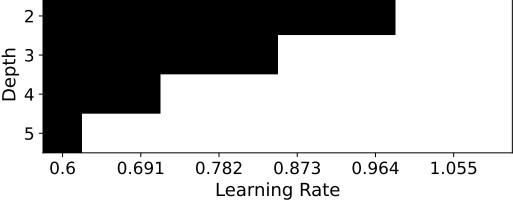

Figure 18: Demonstrating that deeper networks requires a smaller learning rate to enter the EOS regime for DLNs, as implied by Proposition 2, for a target matrix with top singular value $\sigma_{\star,1} = 0.5$ and initialization $\alpha = 0.01$. Black refers to stable learning and white refers to regions in which oscillations in the loss and singular values occur. The EOS limit exactly matches $\eta = 2/L\sigma_{\star,i}^{2-\frac{2}{L}}$.

without bias terms for the weights, with each hidden layer consisting of 1000 units. The network

is trained using MSE loss with a learning rate of $\eta = 4$, along with random weights scaled by $\alpha = 0.01$ and full-batch gradient descent on a 5K subset of the MNIST dataset, following Cohen et al. (2021). The motivation for omitting bias terms comes from the findings of Zhang et al. (2024), where they provably show that a ReLU network without bias terms behaves similarly to a linear network. With this in mind, we aimed to investigate how oscillations manifest in comparison to deep linear networks (DLNs). In Figure 19, we plot the training loss, top-5 singular values, and sharpness throughout training. Interestingly, despite the non-convexity of the loss landscape, the oscillations appear to be almost periodic across all three plots. It would be of great interest to theoretically study the behavior of EOS for this network architecture and determine whether our analyses extend to this case as well.

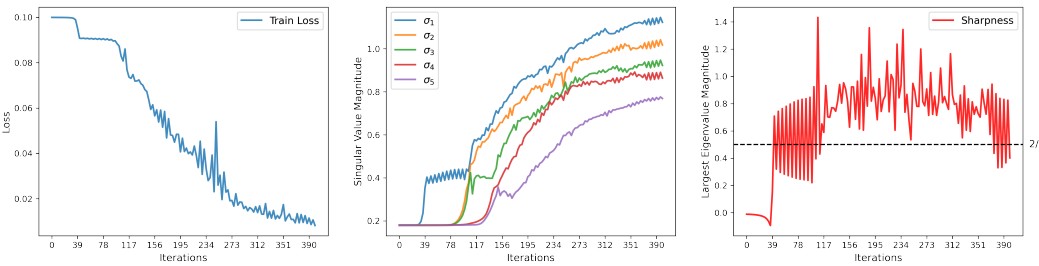

Figure 19: Plot of the training loss, singular values, and sharpness for an MLP network with no bias. Similar to the DLN case, there are oscillations in each of the plots throughout iterations.

Next, we consider the DLN setting to corroborate our result from Theorem 1. We consider modeling rank-3 target matrix with singular values $\sigma_{\star,i} = \{10, 9, 8\}$ with a 3-layer DLN with initialization scale $\alpha = 0.1$. By computing the sharpness under these settings, notice that $2/\lambda_1 = L\sigma_{\star,1}^{2-\frac{2}{L}} \approx 0.01547$ and $2/\lambda_2 \approx 0.01657$. In Figure 21, we use learning rates near these values, and plot the oscillations in the singular values. Here, we can see that the oscillations follow exactly our theory.

Lastly, we provide additional experiments demonstrating stronger oscillation in feature directions as measured by the singular values. To this end, we consider a 4-layer MLP with ReLU activations with hidden layer size in each unit of 200 for classification on a subsampled 20K set on MNIST and CIFAR-10. In Figure 20, we show that the oscillations in the training loss are artifacts of jumps only in the top singular values, which is also what we observe in the DLN setting.

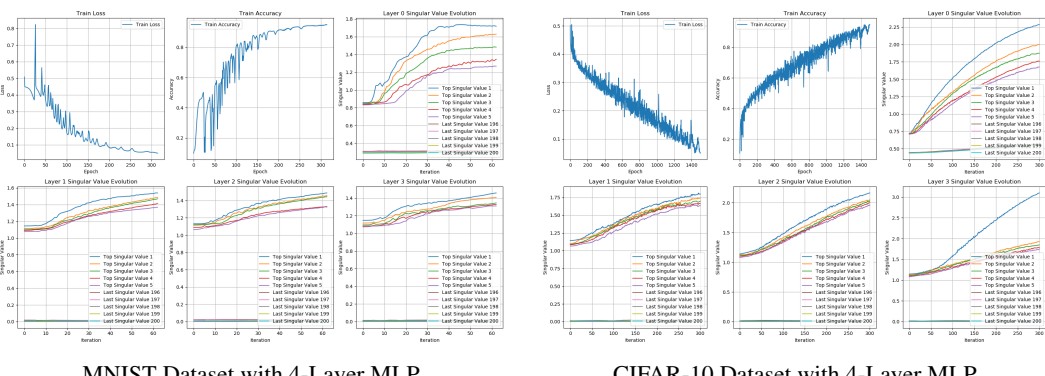

MNIST Dataset with 4-Layer MLP          CIFAR-10 Dataset with 4-Layer MLP

Figure 20: Prevalence of oscillatory behaviors in top subspaces in 4-layer networks with ReLU activations on two different datasets.

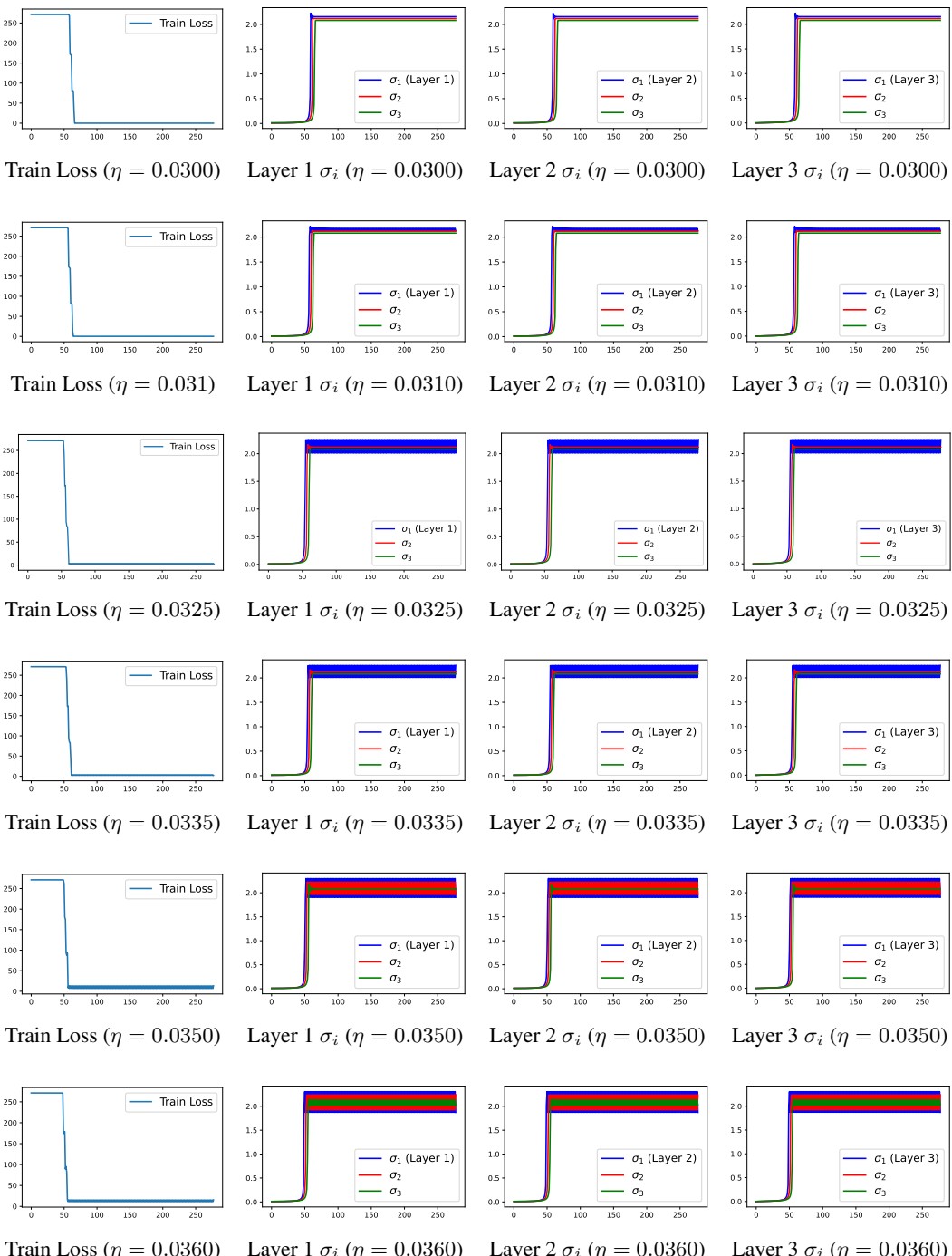

Figure 21: Depiction of the training loss and the singular values of each weight matrix for fitting a rank-3 matrix with singular values $10, 9.5, 9$. The weights enter the EOS regime based on the learning rate $\eta > 2/K$, where $K = L\sigma_{\star,i}^{2-2/L}$ and $L = 3$. For a sufficiently large learning rate (e.g., $\eta = 0.04$), the singular values start to enter a period-4 orbit.

## C  DEFERRED PROOFS

In this section, we present the deferred proofs from the main manuscript.

### C.1  PROOFS FOR SINGULAR VECTOR STATIONARITY

#### C.1.1  PROOF OF PROPOSITION 1

*Proof.* Let us consider the dynamics of $\mathbf{W}_\ell(t)$ in terms of its SVD with respect to time:

$$\dot{\mathbf{W}}_\ell(t) = \dot{\mathbf{U}}_\ell(t)\boldsymbol{\Sigma}_\ell(t)\mathbf{V}_\ell^\top(t) + \mathbf{U}_\ell(t)\dot{\boldsymbol{\Sigma}}_\ell(t)\mathbf{V}_\ell^\top(t) + \mathbf{U}_\ell(t)\boldsymbol{\Sigma}_\ell(t)\dot{\mathbf{V}}_\ell^\top(t). \tag{13}$$

By left multiplying by $\mathbf{U}_\ell^\top(t)$ and right multiplying by $\mathbf{V}_\ell(t)$, we have

$$\mathbf{U}_\ell^\top(t)\dot{\mathbf{W}}_\ell(t)\mathbf{V}_\ell(t) = \mathbf{U}_\ell^\top(t)\dot{\mathbf{U}}_\ell(t)\boldsymbol{\Sigma}_\ell(t) + \dot{\boldsymbol{\Sigma}}_\ell(t) + \boldsymbol{\Sigma}_\ell(t)\dot{\mathbf{V}}_\ell^\top(t)\mathbf{V}_\ell(t), \tag{14}$$

where we used the fact that $\mathbf{U}_\ell(t)$ and $\mathbf{V}_\ell(t)$ have orthonormal columns. Now, note that we also have

$$\mathbf{U}_\ell^\top(t)\mathbf{U}_\ell(t) = \mathbf{I}_r \implies \dot{\mathbf{U}}_\ell^\top(t)\mathbf{U}_\ell(t) + \mathbf{U}_\ell^\top(t)\dot{\mathbf{U}}_\ell(t) = \mathbf{0},$$

which also holds for $\mathbf{V}_\ell(t)$. This implies that $\dot{\mathbf{U}}_\ell^\top(t)\mathbf{U}_\ell(t)$ is a skew-symmetric matrix, and hence have zero diagonals. Since $\boldsymbol{\Sigma}_\ell(t)$ is diagonal, $\mathbf{U}_\ell^\top(t)\dot{\mathbf{U}}_\ell(t)\boldsymbol{\Sigma}_\ell(t)$ and $\boldsymbol{\Sigma}_\ell(t)\dot{\mathbf{V}}_\ell^\top(t)\mathbf{V}_\ell(t)$ have zero diagonals as well. On the other hand, since $\dot{\boldsymbol{\Sigma}}_\ell(t)$ is a diagonal matrix, we can write

$$\hat{\mathbf{I}}_r \odot \left( \mathbf{U}_\ell^\top(t)\dot{\mathbf{W}}_\ell(t)\mathbf{V}_\ell(t) \right) = \mathbf{U}_\ell^\top(t)\dot{\mathbf{U}}_\ell(t)\boldsymbol{\Sigma}_\ell(t) + \boldsymbol{\Sigma}_\ell(t)\dot{\mathbf{V}}_\ell^\top(t)\mathbf{V}_\ell(t), \tag{15}$$

where $\odot$ stands for the Hadamard product and $\hat{\mathbf{I}}_r$ is a square matrix holding zeros on its diagonal and ones elsewhere. Taking transpose of Equation (15), while recalling that $\mathbf{U}_\ell^\top(t)\dot{\mathbf{U}}_\ell(t)$ and $\mathbf{V}_\ell^\top(t)\dot{\mathbf{V}}_\ell(t)$ are skew-symmetric, we have

$$\hat{\mathbf{I}}_r \odot \left( \mathbf{V}_\ell^\top(t)\dot{\mathbf{W}}_\ell^\top(t)\mathbf{U}_\ell(t) \right) = -\boldsymbol{\Sigma}_\ell(t)\mathbf{U}_\ell^\top(t)\dot{\mathbf{U}}_\ell(t) - \dot{\mathbf{V}}_\ell^\top(t)\mathbf{V}_\ell(t)\boldsymbol{\Sigma}_\ell(t). \tag{16}$$

Then, by right multiplying Equation (15) by $\boldsymbol{\Sigma}_\ell(t)$, left-multiply Equation (16) by $\boldsymbol{\Sigma}_\ell(t)$, and by adding the two terms, we get

$$\hat{\mathbf{I}}_r \odot \left( \mathbf{U}_\ell^\top(t)\dot{\mathbf{W}}_\ell(t)\mathbf{V}_\ell(t)\boldsymbol{\Sigma}_\ell(t) + \boldsymbol{\Sigma}_\ell(t)\mathbf{V}_\ell^\top(t)\dot{\mathbf{W}}_\ell^\top(t)\mathbf{U}_\ell(t) \right)$$
$$= \mathbf{U}_\ell^\top(t)\dot{\mathbf{U}}_\ell(t)\boldsymbol{\Sigma}_\ell^2(t) - \boldsymbol{\Sigma}_\ell^2(t)\dot{\mathbf{V}}_\ell^\top(t)\mathbf{V}_\ell(t).$$

Since we assume that the singular values of $\mathbf{M}_\star$ are distinct, the top-$r$ diagonal elements of $\boldsymbol{\Sigma}_\ell^2(t)$ are also distinct (i.e., $\Sigma_r^2(t) \neq \Sigma_{r'}^2(t)$ for $r \neq r'$). This implies that

$$\mathbf{U}_\ell^\top(t)\dot{\mathbf{U}}_\ell(t) = \mathbf{H}(t) \odot \left[ \mathbf{U}_\ell^\top(t)\dot{\mathbf{W}}_\ell(t)\mathbf{V}_\ell(t)\boldsymbol{\Sigma}_\ell(t) + \boldsymbol{\Sigma}_\ell(t)\mathbf{V}_\ell^\top(t)\dot{\mathbf{W}}_\ell^\top(t)\mathbf{U}_\ell(t) \right],$$

where the matrix $\mathbf{H}(t) \in \mathbb{R}^{d \times d}$ is defined by:

$$H_{r,r'}(t) := \begin{cases} \left( \Sigma_{r'}^2(t) - \Sigma_r^2(t) \right)^{-1}, & r \neq r', \\ 0, & r = r'. \end{cases} \tag{17}$$

Then, multiplying from the left by $\mathbf{U}_\ell(t)$ yields

$$\mathbf{P}_{\mathbf{U}_\ell(t)}\dot{\mathbf{U}}_\ell(t) = \mathbf{U}_\ell(t) \left( \mathbf{H}(t) \odot \left[ \mathbf{U}_\ell^\top(t)\dot{\mathbf{W}}_\ell(t)\mathbf{V}_\ell(t)\boldsymbol{\Sigma}_\ell(t) + \boldsymbol{\Sigma}_\ell(t)\mathbf{V}_\ell^\top(t)\dot{\mathbf{W}}_\ell^\top(t)\mathbf{U}_\ell(t) \right] \right), \tag{18}$$

with $\mathbf{P}_{\mathbf{U}_\ell(t)} := \mathbf{U}_\ell(t)\mathbf{U}_\ell^\top(t)$ being the projection onto the subspace spanned by the (orthonormal) columns of $\mathbf{U}_\ell(t)$. Denote by $\mathbf{P}_{\mathbf{U}_{\ell\perp}(t)}$ the projection onto the orthogonal complement ( i.e., $\mathbf{P}_{\mathbf{U}_{\ell\perp}(t)} := \mathbf{I}_r - \mathbf{U}_\ell(t)\mathbf{U}_\ell^\top(t)$). Apply $\mathbf{P}_{\mathbf{U}_{\ell\perp}(t)}$ to both sides of Equation (13):

$$\mathbf{P}_{\mathbf{U}_{\ell\perp}(t)}\dot{\mathbf{U}}_\ell(t) = \mathbf{P}_{\mathbf{U}_{\ell\perp}(t)}\dot{\mathbf{U}}_\ell(t)\boldsymbol{\Sigma}_\ell(t)\mathbf{V}_\ell^\top(t) + \mathbf{P}_{\mathbf{U}_{\ell\perp}(t)}\mathbf{U}_\ell(t)\dot{\boldsymbol{\Sigma}}_\ell(t)\mathbf{V}_\ell^\top(t) \tag{19}$$

$$+ \mathbf{P}_{\mathbf{U}_{\ell\perp}(t)}\mathbf{U}_\ell(t)\boldsymbol{\Sigma}_\ell(t)\dot{\mathbf{V}}_\ell^\top(t). \tag{20}$$

Note that $\mathbf{P}_{\mathbf{U}_{\ell\perp}(t)}\mathbf{U}_\ell(t) = 0$, and multiply from the right by $\mathbf{V}_\ell(t)\mathbf{\Sigma}_\ell^{-1}(t)$ (the latter is well-defined since we have the compact SVD and the top-$r$ elements are non-zero):

$$\mathbf{P}_{\mathbf{U}_{\ell\perp}(t)}\dot{\mathbf{U}}_\ell(t) = \mathbf{P}_{\mathbf{U}_{\ell\perp}(t)}\dot{\mathbf{W}}_\ell(t)\mathbf{V}_\ell(t)\mathbf{\Sigma}_\ell^{-1}(t) = (\mathbf{I}_r - \mathbf{U}_\ell(t)\mathbf{U}^\top(t))\dot{\mathbf{W}}(t)\mathbf{V}_\ell(t)\mathbf{\Sigma}_\ell^{-1}(t). \quad (21)$$

Then by adding the two equations above, we obtain an expression for $\dot{\mathbf{U}}(t)$:

$$\begin{aligned}
\dot{\mathbf{U}}_\ell(t) &= \mathbf{P}_{\mathbf{U}_\ell(t)}\dot{\mathbf{U}}_\ell(t) + \mathbf{P}_{\mathbf{U}_{\ell\perp}(t)}\dot{\mathbf{U}}_\ell(t) \\
&= \mathbf{U}_\ell(t)\left(\mathbf{H}(t) \odot \left[\mathbf{U}_\ell^\top(t)\dot{\mathbf{W}}_\ell(t)\mathbf{V}_\ell(t)\mathbf{\Sigma}_\ell(t) + \mathbf{\Sigma}_\ell(t)\mathbf{V}_\ell^\top(t)\dot{\mathbf{W}}_\ell^\top(t)\mathbf{U}_\ell(t)\right]\right) \\
&\quad + (\mathbf{I}_r - \mathbf{U}_\ell(t)\mathbf{U}_\ell^\top(t))\dot{\mathbf{W}}(t)\mathbf{V}_\ell(t)\mathbf{\Sigma}_\ell^{-1}(t).
\end{aligned} \quad (22)$$

We can similarly derive the dynamics for $\dot{\mathbf{V}}_\ell(t)$ and $\dot{\mathbf{\Sigma}}_\ell(t)$:

$$\dot{\mathbf{V}}_\ell(t) = \mathbf{V}_\ell(t)\left(\mathbf{H}(t) \odot \left[\mathbf{\Sigma}_\ell(t)\mathbf{U}_\ell^\top(t)\dot{\mathbf{W}}_\ell(t)\mathbf{V}_\ell(t) + \mathbf{V}_\ell^\top(t)\dot{\mathbf{W}}_\ell^\top(t)\mathbf{U}_\ell(t)\mathbf{\Sigma}_\ell(t)\right]\right) \quad (23)$$

$$+ \left(\mathbf{I}_r - \mathbf{V}_\ell(t)\mathbf{V}_\ell^\top(t)\right)\dot{\mathbf{W}}_\ell^\top(t)\mathbf{U}_\ell(t)\mathbf{\Sigma}_\ell^{-1}(t), \quad (24)$$

$$\dot{\mathbf{\Sigma}}_\ell(t) = \mathbf{I}_r \odot \left[\mathbf{U}_\ell^\top(t)\dot{\mathbf{W}}_\ell(t)\mathbf{V}_\ell(t)\right].$$

Now, we will left multiply $\dot{\mathbf{U}}_\ell(t)$ and $\dot{\mathbf{V}}_\ell(t)$ with $\mathbf{U}_\ell^\top(t)$ and $\mathbf{V}_\ell^\top(t)$, respectively, to obtain

$$\mathbf{U}_\ell^\top(t)\dot{\mathbf{U}}_\ell(t) = -\mathbf{H}(t) \odot \left[\mathbf{U}_\ell^\top(t)\nabla_{\mathbf{W}_\ell}f(\mathbf{\Theta})\mathbf{V}_\ell(t)\mathbf{\Sigma}_\ell(t) + \mathbf{\Sigma}_\ell(t)\mathbf{V}_\ell^\top(t)\nabla_{\mathbf{W}_\ell}f(\mathbf{\Theta})\mathbf{U}_\ell(t)\right],$$

$$\mathbf{V}_\ell^\top(t)\dot{\mathbf{V}}_\ell(t) = -\mathbf{H}(t) \odot \left[\mathbf{\Sigma}_\ell(t)\mathbf{U}_\ell^\top(t)\nabla_{\mathbf{W}_\ell}f(\mathbf{\Theta})\mathbf{V}_\ell(t) + \mathbf{V}_\ell^\top(t)\nabla_{\mathbf{W}_\ell}f(\mathbf{\Theta})\mathbf{U}_\ell(t)\mathbf{\Sigma}_\ell(t)\right],$$

where we replaced $\dot{\mathbf{W}}_\ell(t) := -\nabla_{\mathbf{W}_\ell}f(\mathbf{\Theta})$, as $\dot{\mathbf{W}}_\ell(t)$ is the gradient of $f(\mathbf{\Theta})$ with respect to $\mathbf{W}_\ell$ by definition. By rearranging and multiplying by $\mathbf{\Sigma}_\ell(t)$, we have

$$\mathbf{U}_\ell^\top(t)\dot{\mathbf{U}}_\ell(t)\mathbf{\Sigma}_\ell(t) - \mathbf{\Sigma}_\ell(t)\mathbf{V}^T(t)\dot{\mathbf{V}}_\ell(t) = -\hat{\mathbf{I}}_r \odot [\mathbf{U}_\ell^\top(t)\nabla_{\mathbf{W}_\ell}f(\mathbf{\Theta})\mathbf{V}_\ell(t)]. \quad (25)$$

Hence, when $\dot{\mathbf{U}}_\ell(t) = 0$ and $\dot{\mathbf{V}}_\ell(t) = 0$, it must be that the left-hand side is zero and so $\mathbf{U}_\ell^\top(t)\nabla_{\mathbf{W}_\ell}f(\mathbf{\Theta})\mathbf{V}_\ell(t)$ is a diagonal matrix.

Now, notice that for the given loss function $f(\mathbf{\Theta})$, we have

$$-\dot{\mathbf{W}}_\ell(t) = \nabla_{\mathbf{W}_\ell}f(\mathbf{\Theta}(t)) = \mathbf{W}_{L:\ell+1}^\top(t) \cdot (\mathbf{W}_{L:1}(t) - \mathbf{M}_\star) \cdot \mathbf{W}_{\ell-1:1}^\top(t).$$

Then, from Equation (25), when the singular vectors are stationary, we have

$$\mathbf{U}_\ell^\top(t)\mathbf{W}_{L:\ell+1}^\top(t) \cdot (\mathbf{W}_{L:1}(t) - \mathbf{M}_\star) \cdot \mathbf{W}_{\ell-1:1}^\top(t)\mathbf{V}_\ell(t)$$

must be a diagonal matrix for all $\ell \in [L]$. The only solution to the above should be (since the intermediate singular vectors need to cancel to satisfy the diagonal condition), is the set

$$\text{SVS}(f(\mathbf{\Theta})) = \begin{cases} (\mathbf{U}_L, \mathbf{V}_L) &= (\mathbf{U}_\star, \mathbf{Q}_L), \\ (\mathbf{U}_\ell, \mathbf{V}_\ell) &= (\mathbf{Q}_{\ell+1}, \mathbf{Q}_\ell), \quad \forall \ell \in [2, L-1], \\ (\mathbf{U}_1, \mathbf{V}_1) &= (\mathbf{Q}_2, \mathbf{V}_\star), \end{cases}$$

where $\{\mathbf{Q}_\ell\}_{\ell=2}^L$ are any set of orthogonal matrices. Then, notice that when the singular vectors are stationary, the dynamics become isolated on the singular values:

$$\dot{\mathbf{\Sigma}}_\ell(t) = \mathbf{I}_r \odot \left[\mathbf{U}_\ell^\top(t)\dot{\mathbf{W}}_\ell(t)\mathbf{V}_\ell(t)\right],$$

since $\left[\mathbf{U}_\ell^\top(t)\dot{\mathbf{W}}_\ell(t)\mathbf{V}_\ell(t)\right]$ is diagonal. This completes the proof.

$\square$

C.1.2  SUPPORTING RESULTS

**Proposition 3.** *Let* $\mathbf{M}_\star = \mathbf{U}_\star \mathbf{\Sigma}_\star \mathbf{V}_\star^\top$ *denote the SVD of the target matrix. The initialization in Equation (4) is a member of the singular vector stationary set in Proposition 1, where* $\mathbf{Q}_L = \ldots = \mathbf{Q}_2 = \mathbf{V}_\star.$

*Proof.* Recall that the initialization is given by

$$\mathbf{W}_L(0) = 0 \quad \text{and} \quad \mathbf{W}_\ell(0) = \alpha \mathbf{I}_d \quad \forall \ell \in [L-1].$$

We will show that under this initialization, each weight matrix admits the following decomposition for all $t \geq 1$:

$$\mathbf{W}_L(t) = \mathbf{U}_\star \begin{bmatrix} \widetilde{\mathbf{\Sigma}}_L(t) & \mathbf{0} \\ \mathbf{0} & \mathbf{0} \end{bmatrix} \mathbf{V}_\star^\top, \qquad \mathbf{W}_\ell(t) = \mathbf{V}_\star \begin{bmatrix} \widetilde{\mathbf{\Sigma}}(t) & \mathbf{0} \\ \mathbf{0} & \alpha \mathbf{I}_{d-r} \end{bmatrix} \mathbf{V}_\star^\top, \quad \forall \ell \in [L-1], \quad (26)$$

where

$$\widetilde{\mathbf{\Sigma}}_L(t) = \widetilde{\mathbf{\Sigma}}_L(t-1) - \eta \cdot \left( \widetilde{\mathbf{\Sigma}}_L(t-1) \cdot \widetilde{\mathbf{\Sigma}}^{L-1}(t-1) - \mathbf{\Sigma}_{\star,r} \right) \cdot \widetilde{\mathbf{\Sigma}}^{L-1}(t-1)$$

$$\widetilde{\mathbf{\Sigma}}(t) = \widetilde{\mathbf{\Sigma}}(t-1) \cdot \left( \mathbf{I}_r - \eta \cdot \widetilde{\mathbf{\Sigma}}_L(t-1) \cdot \left( \widetilde{\mathbf{\Sigma}}_L(t-1) \cdot \widetilde{\mathbf{\Sigma}}^{L-1}(t-1) - \mathbf{\Sigma}_{\star,r} \right) \cdot \widetilde{\mathbf{\Sigma}}^{L-3}(t-1) \right),$$

where $\widetilde{\mathbf{\Sigma}}_L(t), \widetilde{\mathbf{\Sigma}}(t) \in \mathbb{R}^{r \times r}$ is a diagonal matrix with $\widetilde{\mathbf{\Sigma}}_L(1) = \eta \alpha^{L-1} \cdot \mathbf{\Sigma}_{r,\star}$ and $\widetilde{\mathbf{\Sigma}}(1) = \alpha \mathbf{I}_r.$

This will prove that the singular vectors are stationary with $\mathbf{\Sigma}_L = \ldots = \mathbf{\Sigma}_2 = \mathbf{V}_\star.$ We proceed with mathematical induction.

**Base Case.** For the base case, we will show that the decomposition holds for each weight matrix at $t = 1$. The gradient of $f(\mathbf{\Theta})$ with respect to $\mathbf{W}_\ell$ is

$$\nabla_{\mathbf{W}_\ell} f(\mathbf{\Theta}) = \mathbf{W}_{L:\ell+1}^\top \cdot (\mathbf{W}_{L:1} - \mathbf{M}_\star) \cdot \mathbf{W}_{\ell-1:1}^\top.$$

For $\mathbf{W}_L(1)$, we have

$$\begin{aligned}
\mathbf{W}_L(1) &= \mathbf{W}_L(0) - \eta \cdot \nabla_{\mathbf{W}_L} f(\mathbf{\Theta}(0)) \\
&= \mathbf{W}_L(0) - \eta \cdot (\mathbf{W}_{L:1}(0) - \mathbf{M}_\star) \cdot \mathbf{W}_{L-1:1}^\top(0) \\
&= \eta \alpha^{L-1} \mathbf{\Sigma}_\star \\
&= \mathbf{U}_\star \cdot \left( \eta \alpha^{L-1} \cdot \mathbf{\Sigma}_\star \right) \cdot \mathbf{V}_\star^\top \\
&= \mathbf{U}_\star \begin{bmatrix} \widetilde{\mathbf{\Sigma}}_L(1) & \mathbf{0} \\ \mathbf{0} & \mathbf{0} \end{bmatrix} \mathbf{V}_\star^\top.
\end{aligned}$$

Then, for each $\mathbf{W}_\ell(1)$ in $\ell \in [L-1]$, we have

$$\begin{aligned}
\mathbf{W}_\ell(1) &= \mathbf{W}_\ell(0) - \eta \cdot \nabla_{\mathbf{W}_\ell} f(\mathbf{\Theta}(0)) \\
&= \alpha \mathbf{I}_d,
\end{aligned}$$

where the last equality follows from the fact that $\mathbf{W}_L(0) = \mathbf{0}$. Finally, we have

$$\mathbf{W}_\ell(1) = \alpha \mathbf{V}_\star \mathbf{V}_\star^\top = \mathbf{V}_\star \begin{bmatrix} \widetilde{\mathbf{\Sigma}}(1) & \mathbf{0} \\ \mathbf{0} & \alpha \mathbf{I}_{d-r} \end{bmatrix} \mathbf{V}_\star^\top, \quad \forall \ell \in [L-1].$$

**Inductive Step.** By the inductive hypothesis, suppose that the decomposition holds. Then, notice that we can simplify the end-to-end weight matrix to

$$\mathbf{W}_{L:1}(t) = \mathbf{U}_\star \begin{bmatrix} \widetilde{\mathbf{\Sigma}}_L(t) \cdot \widetilde{\mathbf{\Sigma}}^{L-1}(t) & \mathbf{0} \\ \mathbf{0} & \mathbf{0} \end{bmatrix} \mathbf{V}_\star^\top,$$

for which we can simplify the gradients to

$$\begin{aligned}
\nabla_{\mathbf{W}_L} f(\mathbf{\Theta}(t)) &= \left( \mathbf{U}_\star \begin{bmatrix} \widetilde{\mathbf{\Sigma}}_L(t) \cdot \widetilde{\mathbf{\Sigma}}^{L-1}(t) - \mathbf{\Sigma}_{\star,r} & \mathbf{0} \\ \mathbf{0} & \mathbf{0} \end{bmatrix} \mathbf{V}_\star^\top \right) \cdot \mathbf{V}_\star \begin{bmatrix} \widetilde{\mathbf{\Sigma}}^{L-1}(t) & \mathbf{0} \\ \mathbf{0} & \mathbf{0} \end{bmatrix} \mathbf{V}_\star^\top \\
&= \mathbf{U}_\star \begin{bmatrix} \left( \widetilde{\mathbf{\Sigma}}_L(t) \cdot \widetilde{\mathbf{\Sigma}}^{L-1}(t) - \mathbf{\Sigma}_{\star,r} \right) \cdot \widetilde{\mathbf{\Sigma}}^{L-1}(t) & \mathbf{0} \\ \mathbf{0} & \mathbf{0} \end{bmatrix} \mathbf{V}_\star^\top,
\end{aligned}$$

for the last layer matrix, and similarly,

$$\nabla_{\mathbf{W}_\ell} f(\boldsymbol{\Theta}(t)) = \mathbf{V}_\star \begin{bmatrix} \widetilde{\boldsymbol{\Sigma}}_L(t) \cdot \left( \widetilde{\boldsymbol{\Sigma}}_L(t) \cdot \widetilde{\boldsymbol{\Sigma}}^{L-1}(t) - \boldsymbol{\Sigma}_{\star,r} \right) \cdot \widetilde{\boldsymbol{\Sigma}}^{L-2}(t) & \mathbf{0} \\ \mathbf{0} & \mathbf{0} \end{bmatrix} \mathbf{V}_\star^\top, \quad \ell \in [L-1],$$

for all other layer matrices. Thus, for the next GD iteration, we have

$$\begin{aligned} \mathbf{W}_L(t+1) &= \mathbf{W}_L(t) - \eta \cdot \nabla_{\mathbf{W}_L}(\boldsymbol{\Theta}(t)) \\ &= \mathbf{U}_\star \begin{bmatrix} \widetilde{\boldsymbol{\Sigma}}_L(t) - \eta \cdot \left( \widetilde{\boldsymbol{\Sigma}}_L(t) \cdot \widetilde{\boldsymbol{\Sigma}}^{L-1}(t) - \boldsymbol{\Sigma}_{\star,r} \right) \cdot \widetilde{\boldsymbol{\Sigma}}^{L-1}(t) & \mathbf{0} \\ \mathbf{0} & \mathbf{0} \end{bmatrix} \mathbf{V}_\star^\top \\ &= \mathbf{U}_\star \begin{bmatrix} \widetilde{\boldsymbol{\Sigma}}_L(t+1) & \mathbf{0} \\ \mathbf{0} & \mathbf{0} \end{bmatrix} \mathbf{V}_\star^\top. \end{aligned}$$

Similarly, we have

$$\begin{aligned} \mathbf{W}_\ell(t+1) &= \mathbf{W}_\ell(t) - \eta \cdot \nabla_{\mathbf{W}_\ell}(\boldsymbol{\Theta}(t)) \\ &= \mathbf{V}_\star \begin{bmatrix} \widetilde{\boldsymbol{\Sigma}}(t) - \eta \cdot \widetilde{\boldsymbol{\Sigma}}_L(t) \cdot \left( \widetilde{\boldsymbol{\Sigma}}_L(t) \cdot \widetilde{\boldsymbol{\Sigma}}^{L-1}(t) - \boldsymbol{\Sigma}_{\star,r} \right) \cdot \widetilde{\boldsymbol{\Sigma}}^{L-2}(t) & \mathbf{0} \\ \mathbf{0} & \alpha \mathbf{I}_{d-r} \end{bmatrix} \mathbf{V}_\star^\top \\ &= \mathbf{V}_\star \begin{bmatrix} \widetilde{\boldsymbol{\Sigma}}(t) \cdot \left( \mathbf{I}_r - \eta \cdot \widetilde{\boldsymbol{\Sigma}}_L(t) \cdot \left( \widetilde{\boldsymbol{\Sigma}}_L(t) \cdot \widetilde{\boldsymbol{\Sigma}}^{L-1}(t) - \boldsymbol{\Sigma}_{\star,r} \right) \cdot \widetilde{\boldsymbol{\Sigma}}^{L-3}(t) \right) & \mathbf{0} \\ \mathbf{0} & \alpha \mathbf{I}_{d-r} \end{bmatrix} \mathbf{V}_\star^\top \\ &= \mathbf{V}_\star \begin{bmatrix} \widetilde{\boldsymbol{\Sigma}}(t+1) & \mathbf{0} \\ \mathbf{0} & \alpha \mathbf{I}_{d-r} \end{bmatrix} \mathbf{V}_\star^\top, \end{aligned}$$

for all $\ell \in [L-1]$. This completes the proof. $\qquad\square$

**Proposition 4.** *Let $\mathbf{M}_\star = \mathbf{V}_\star \boldsymbol{\Sigma}_\star \mathbf{V}_\star^\top \in \mathbb{R}^{d \times d}$ denote the SVD of the target matrix. The balanced initialization in Equation (3) is a member of the singular vector stationary set in Proposition 1, where $\mathbf{U}_L = \mathbf{Q}_L = \ldots = \mathbf{Q}_2 = \mathbf{V}_1 = \mathbf{V}_\star$.*

*Proof.* Using mathematical induction, we will show that with the balanced initialization in Equation (3), each weight matrix admits a decomposition of the form

$$\mathbf{W}_\ell(t) = \mathbf{V}_\star \boldsymbol{\Sigma}_\ell(t) \mathbf{V}_\star^\top, \tag{27}$$

which implies that the singular vectors are stationary for all $t$ such that $\mathbf{U}_L = \mathbf{Q}_L = \ldots = \mathbf{Q}_2 = \mathbf{V}_1 = \mathbf{V}_\star$.

**Base Case.** Consider the weights at iteration $t = 0$. By the initialization scheme, we can write each weight matrix as

$$\mathbf{W}_\ell(0) = \alpha \mathbf{I}_d \implies \mathbf{W}_\ell(0) = \alpha \mathbf{V}_\star \mathbf{V}_\star^\top,$$

which implies that $\mathbf{W}_\ell(0) = \mathbf{V}_\star \boldsymbol{\Sigma}_\ell(0) \mathbf{V}_\star^\top$ with $\boldsymbol{\Sigma}_\ell(0) = \alpha \mathbf{I}_d$.

**Inductive Step.** By the inductive hypothesis, assume that the decomposition holds for all $t \geq 0$. We will show that it holds for all iterations $t + 1$. Recall that the gradient of $f(\boldsymbol{\Theta})$ with respect to $\mathbf{W}_\ell$ is

$$\nabla_{\mathbf{W}_\ell} f(\boldsymbol{\Theta}) = \mathbf{W}_{L:\ell+1}^\top \cdot (\mathbf{W}_{L:1} - \mathbf{M}_\star) \cdot \mathbf{W}_{\ell-1:1}^\top.$$

Then, for $\mathbf{W}_\ell(t+1)$, we have

$$\begin{aligned} \mathbf{W}_\ell(t+1) &= \mathbf{W}_\ell(t) - \eta \cdot \nabla_{\mathbf{W}_L} f(\boldsymbol{\Theta}(t)) \\ &= \mathbf{V}_\star \boldsymbol{\Sigma}_\ell(t) \mathbf{V}_\star^\top - \eta \mathbf{W}_{L:\ell+1}^\top(t) \cdot (\mathbf{W}_{L:1}(t) - \mathbf{M}_\star) \cdot \mathbf{W}_{\ell-1:1}^\top(t) \\ &= \mathbf{V}_\star \boldsymbol{\Sigma}_\ell(t) \mathbf{V}_\star^\top - \eta \mathbf{V}_\star \cdot \left( \boldsymbol{\Sigma}_\ell^{L-\ell}(t) \cdot \left( \boldsymbol{\Sigma}_\ell^L(t) - \boldsymbol{\Sigma}_\star \right) \cdot \boldsymbol{\Sigma}_\ell^{\ell-1}(t) \right) \cdot \mathbf{V}_\star^\top \\ &= \mathbf{V}_\star \cdot \left( \boldsymbol{\Sigma}_\ell(t) - \eta \cdot \boldsymbol{\Sigma}_\ell^{L-\ell}(t) \cdot \left( \boldsymbol{\Sigma}_\ell^L(t) - \boldsymbol{\Sigma}_\star \right) \cdot \boldsymbol{\Sigma}_\ell^{\ell-1}(t) \right) \cdot \mathbf{V}_\star^\top \\ &= \mathbf{V}_\star \boldsymbol{\Sigma}(t) \mathbf{V}_\star^\top, \end{aligned}$$

where $\boldsymbol{\Sigma}(t) = \boldsymbol{\Sigma}_\ell(t) - \eta \cdot \boldsymbol{\Sigma}_\ell^{L-\ell}(t) \cdot \left( \boldsymbol{\Sigma}_\ell^L(t) - \boldsymbol{\Sigma}_\star \right) \cdot \boldsymbol{\Sigma}_\ell^{\ell-1}(t)$. This completes the proof. $\qquad\square$

## C.2 PROOFS FOR BALANCING

In this section, we present our proof of Proposition 2 along with supporting results. Throughout these results, we use the notion of the gradient flow solution (GFS) and the GFS sharpness as presented by Kreisler et al. (2023), which we briefly recap.

Consider minimizing a smooth loss function $\mathcal{L} : \mathbb{R}^d \to \mathbb{R}$ using gradient flow (GF):

$$\dot{\mathbf{w}}(t) = -\nabla\mathcal{L}(\mathbf{w}(t)).$$

The GFS denoted by $S_{\mathrm{GF}}(\mathbf{w})$ is the limit of the gradient flow trajectory when initialized at $\mathbf{w}$. Furthermore, the GFS sharpness denoted by $\psi(\mathbf{w})$ is defined to be the sharpness of $S_{\mathrm{GF}}(\mathbf{w})$, i.e., the largest eigenvalue of $\nabla^2\mathcal{L}(S_{\mathrm{GF}}(\mathbf{w}))$.

### C.2.1 SUPPORTING LEMMAS

**Lemma 2** (Conservation of Balancedness in GF). *Consider the singular value scalar loss*

$$\mathcal{L}\left(\{\sigma_\ell\}_{\ell=1}^L\right) = \frac{1}{2}\left(\prod_{\ell=1}^L \sigma_\ell - \sigma_\star\right)^2.$$

*Under gradient flow, the balancedness between two singular values defined by $\sigma_\ell^2(t) - \sigma_m^2(t)$ for all $m, \ell \in [L]$ is constant for all $t \geq 0$.*

*Proof.* Notice that the result holds specifically for gradient flow and not descent. The dynamics of each scalar factor for gradient flow can be written as

$$\dot{\sigma}_\ell(t) = -\left(\prod_{\ell=1}^L \sigma_\ell(t) - \sigma_\star\right) \cdot \prod_{i\neq\ell}^L \sigma_i(t)$$

Then, the time derivative of balancing is given as

$$\frac{\partial}{\partial t}(\sigma_\ell^2(t) - \sigma_m^2(t)) = \sigma_\ell(t)\dot{\sigma}_\ell(t) - \sigma_m(t)\dot{\sigma}_m(t)$$

$$= -\sigma_\ell(t)\left(\prod_{\ell=1}^L \sigma_\ell(t) - \sigma_\star\right) \cdot \prod_{i\neq\ell}^L \sigma_i(t) + \sigma_m(t)\left(\prod_{m=1}^L \sigma_\ell(t) - \sigma_\star\right) \cdot \prod_{j\neq m}^L \sigma_j(t).$$

$$= 0.$$

Hence, the quantity $\sigma_\ell^2(t) - \sigma_m^2(t)$ remains constant for all time $t \geq 0$, hence preserving balancedness. □

**Lemma 3** (Sharpness at Minima). *Consider the singular value scalar loss*

$$\mathcal{L}(\{\sigma_i\}_{i=1}^d) = \frac{1}{2}\left(\prod_{i=1}^L \sigma_i - \sigma_\star\right)^2,$$

*The sharpness at the global minima is given as $\|\nabla^2\mathcal{L}\|_2 = \sum_{i=1}^L \frac{\sigma_\star^2}{\sigma_i^2}$.*

*Proof.* The gradient is given by

$$\nabla_{\sigma_i}\mathcal{L} = \left(\prod_{\ell=1}^L \sigma_\ell(t) - \sigma_\star\right)\prod_{j\neq i}^L \sigma_j(t).$$

Then,

$$\nabla_{\sigma_j}\nabla_{\sigma_i}\mathcal{L} = \prod_{\ell\neq i}^L \sigma_\ell(t)\prod_{\ell\neq j}^L \sigma_\ell(t) + \left(\prod_{\ell=1}^L \sigma_\ell(t) - \sigma_\star\right)\prod_{\ell\neq j, \ell\neq i}^L \sigma_\ell(t)$$

Let $\pi(t) = \prod_{i=1}^{L} \sigma_i(t)$. Then, at the global minima, we have

$$\nabla_{\sigma_j} \nabla_{\sigma_i} \mathcal{L} = \frac{\pi^2}{\sigma_i \sigma_j} = \frac{\sigma_\star^2}{\sigma_i \sigma_j}$$

Thus, the sharpness of the largest eigenvalue is given as $\|\nabla^2 \mathcal{L}\|_2 = \sum_{i=1}^{L} \frac{\sigma_\star^2}{\sigma_i^2}$. $\qquad \square$

**Lemma 4** (Balanced Minima is the Flattest). *Consider the singular value scalar loss*

$$\mathcal{L}\left(\{\sigma_i\}_{i=1}^{L}\right) = \frac{1}{2}\left(\prod_{i=1}^{L} \sigma_i - \sigma_\star\right)^2.$$

*The balanced minimum (i.e., $\sigma_i = \sigma_\star^{1/L}$ for all $i \in [L]$) has the smallest sharpness amongst all global minima with a value of $\|\nabla^2 \mathcal{L}\|_2 = L\sigma_\star^{2-2/L}$.*

*Proof.* By Lemma 3, recall that the sharpness at the global minima is given in the form

$$\|\nabla^2 \mathcal{L}\|_2 = \sum_{i=1}^{L} \frac{\sigma_\star^2}{\sigma_i^2}.$$

To show that the balanced minimum is the flattest (i.e., it has the smallest sharpness amongst all global minima), we will show that KKT stationarity condition of the constrained objective

$$\min_{\{\sigma_i\}_{i=1}^{L}} \sum_{i=1}^{L} \frac{\sigma_\star^2}{\sigma_i^2} \quad \text{s.t.} \quad \prod_{i=1}^{L} \sigma_i = \sigma_\star,$$

are only met at the balanced minimum, which gives us the sharpness value $\|\nabla^2 \mathcal{L}\|_2 = L\sigma_\star^{2-2/L}$. The Lagrangian is given by

$$L(\sigma_1, \ldots, \sigma_L, \mu) = \sum_{i=1}^{L} \frac{\sigma_\star^2}{\sigma_i^2} + \mu\left(\prod_{i=1}^{L} \sigma_i - \sigma_\star\right).$$

Then, the stationary point conditions of the Langrangian is given by

$$\frac{\partial L}{\partial \sigma_i} = -\frac{2\sigma_\star^2}{\sigma_i^3} + \mu \prod_{j \neq i} \sigma_j = 0, \tag{28}$$

$$\frac{\partial L}{\partial \mu} = \prod_{i=1}^{L} \sigma_i - \sigma_\star = 0. \tag{29}$$

From Equation (28), the solution of the stationary point gives

$$\frac{2\sigma_\star^2}{\sigma_i^3} = \mu \prod_{j \neq i} \sigma_j \implies \mu = \frac{2\sigma_\star^2}{\sigma_i^3 \prod_{j \neq i} \sigma_j} = \frac{2\sigma_\star^2}{\sigma_i^2 \sigma_\star} = \frac{2\sigma_\star}{\sigma_i^2}.$$

This also indicates that at the stationary point, $\sigma_i = \sqrt{\frac{2\sigma_\star}{\mu}}$ for all $i \in [L]$, which means that the condition is *only* satisfied at the balanced minimum, i.e, $\sigma_i = \sigma_\star^{1/L}$. Furthermore, notice that

$$\nabla^2 f(\sigma_i) = 6\sigma_\star^2 \cdot \text{Diag}\left(\frac{1}{\sigma_i^4}\right) \succ \mathbf{0},$$

where $f(\sigma_i) = \sum_{i=1}^{L} \frac{\sigma_\star^2}{\sigma_i^2}$, indicating that $f$ only has a minimum. Notice that Equation (29) holds immediately. Thus, the balanced minimum has the smallest shaprness (flattest), which plugging into $f$ gives a sharpness of $\|\nabla^2 \mathcal{L}\|_2 = L\sigma_\star^{2-2/L}$.

$\qquad \square$

**Lemma 5.** *Let* $\mathbf{s} := [\sigma_1 \quad \sigma_2 \quad \ldots \quad \sigma_L] \in \mathbb{R}^L$ *and define the singular value scalar loss as*

$$\mathcal{L}(\mathbf{s}) = \frac{1}{2} \left( \prod_{i=1}^{L} \sigma_i - \sigma_\star \right)^2,$$

*for some* $\sigma_\star > 0$. *If* $\sigma \in \mathbb{R}^L$ *are initialized such that*

$$\sigma_L(0) = 0 \quad and \quad \sigma_\ell(0) = \alpha, \quad \forall \ell \in [L-1],$$

*where* $0 < \alpha < \left( \ln \left( \frac{2\sqrt{2}}{\eta L \sigma_\star^{2-\frac{2}{L}}} \right) \cdot \frac{\sigma_\star^{\frac{4}{L}}}{L^2 \cdot 2^{\frac{2L-3}{L}}} \right)^{\frac{1}{4}}$ *and* $\eta > 0$, *then the GFS sharpness satisfies* $\psi(\mathbf{s}) \leq \frac{2\sqrt{1+c}}{\eta}$ *for some* $0 < c < 1$.

*Proof.* We will show that the necessary condition for the GFS sharpness to satisfy $\psi(\mathbf{s}) \leq \frac{2\sqrt{1+c}}{\eta}$ for some $\eta > 0$ and $0 < c < 1$ to hold is that the initialization scale $\alpha$ must satisfy $0 < \alpha < \left( \ln \left( \frac{2\sqrt{2}}{\eta L \sigma_\star^{2-\frac{2}{L}}} \right) \cdot \frac{\sigma_\star^{\frac{4}{L}}}{L^2 \cdot 2^{\frac{2L-3}{L}}} \right)^{\frac{1}{4}}$.

Since the singular values $\sigma_\ell$ for all $\ell \in [L-1]$ are initialized to $\alpha$, note that they all follow the same dynamics. Then, let us define the following for simplicity in exposition:

$$y := \sigma_1 = \ldots = \sigma_{L-1} \quad and \quad x := \sigma_L,$$

and so $\prod_{\ell=1}^{L} \sigma_\ell = xy^{L-1}$. Then, note that the gradient flow (GF) solution is the intersection between

$$xy^{L-1} = \sigma_\star \quad and \quad x^2 - y^2 = -\alpha^2,$$

where the first condition comes from convergence and the second comes from the conservation flow law of GF from in Lemma 2. Then, if we can find a solution at the intersection such that

$$(\hat{x}(\alpha), \hat{y}(\alpha)) = \begin{cases} xy^{L-1} = \sigma_\star \\ x^2 - y^2 = -\alpha^2, \end{cases} \tag{30}$$

solely in terms of $\alpha$, we can plug in $(\hat{x}(\alpha), \hat{y}(\alpha))$ into the GFS[1]:

$$\psi(\hat{x}(\alpha), \hat{y}(\alpha)) = \psi(\mathbf{s}) \overset{(i)}{=} \sum_{i=1}^{L} \frac{\sigma_\star^2}{\sigma_i^2} = \sigma_\star^2 \left( \frac{1}{\hat{x}(\alpha)^2} + \frac{L-1}{\hat{y}(\alpha)^2} \right) < \frac{2\sqrt{2}}{\eta}, \tag{31}$$

and solve to find an upper bound in terms of $\alpha$, where (i) comes from Lemma 3. The strict inequality ensures that we can find a $c$ in $c \in [0, 1)$ such that $\psi(\mathbf{s}) \leq \frac{2\sqrt{1+c}}{\eta}$. However, the intersection $(\hat{x}(\alpha), \hat{y}(\alpha))$ is a $2L$-th order polynomial in $\hat{y}(\alpha)$ which does not have a straightforward closed-form solution solely in terms of $\alpha$. To this end, we aim to find a more tractable upper bound on $\psi(\hat{x}(\alpha), \hat{y}(\alpha))$ by using variational calculus, and use that to find a bound on $\alpha$ instead. Specifically, we will compute the differential $d\psi$, upper bound $d\psi$ with a tractable function, and then integrate to obtain our new function $\psi'$ for which we use to set $\psi' < \frac{2\sqrt{2}}{\eta}$.

**Computing the Differentials** $d\hat{x}$ **and** $d\hat{y}$. Before computing the differential $d\psi$, we need to derive the differentials of $\hat{x}(\alpha)$ and $\hat{y}(\alpha)$. We drop the $\alpha$ notation and use $\hat{x}$ and $\hat{y}$ where applicable. By plugging in $\hat{x}$ into Equation (30), the solution $\hat{y}$ satisfies

$$\hat{y}^{2L} - \alpha^2 \hat{y}^{2L-2} = \sigma_\star^2.$$

Then, by differentiating the relation with respect to $\alpha$, we obtain the following variational relation:

$$2L\hat{y}^{2L-1}d\hat{y} - \alpha^2 2(L-1)\hat{y}^{2L-3}d\hat{y} - 2\alpha\hat{y}^{2L-2}d\alpha = 0$$

$$\implies \hat{y}^{2L-3}(\hat{y}^2 L - \alpha^2(L-1))d\hat{y} = \alpha\hat{y}^{2(L-1)}d\alpha$$

$$\implies d\hat{y} = \frac{\hat{y}\alpha}{(\hat{y}^2 L - \alpha^2(L-1))}d\alpha, \tag{32}$$

---

[1]Note that throughout the proof $(\hat{x}(\alpha), \hat{y}(\alpha))$ denotes the gradient flow solution as function of $\alpha$. It does not refer to the GF trajectory.

where we used Lemma 3.10 of Kreisler et al. (2023) to deduce that $\hat{y} > 0$ and so $\hat{y}^{2L-2} > 0$. Then, notice that we have $\hat{y} > \sqrt{\frac{L-1}{L}}\alpha$ from initialization, and so we $\frac{d\hat{y}}{d\alpha} > 0$, (i.e., $\hat{y}(\alpha)$ is an increasing function of $\alpha$). Then, we also have

$$\lim_{\alpha \to 0} \hat{y}(\alpha) = \sigma_\star^{1/L} \quad \text{and} \quad \lim_{\alpha \to 0} \hat{x}(\alpha) = \sigma_\star^{1/L},$$

as it corresponds to exact balancing. Hence, as $\alpha$ increases from 0, $\hat{y}(\alpha)$ increases from $\sigma_\star^{1/L}$.

Similarly, the intersection at the global minima satisfies the following relation for $\hat{x}$:

$$\hat{x}^{\left(2 + \frac{2}{L-1}\right)} + \hat{x}^{\frac{2}{L-1}}\alpha^2 = \sigma_\star^{\frac{2}{L-1}}$$

$$\implies \left(2 + \frac{2}{L-1}\right)\hat{x}^{\left(\frac{2}{L-1}+1\right)}d\hat{x} + \left(\frac{2}{L-1}\right)\alpha^2\hat{x}^{\left(\frac{2}{L-1}-1\right)}d\hat{x} + 2\alpha\hat{x}^{\frac{2}{L-1}}d\alpha = 0$$

$$\implies d\hat{x} = \frac{-\alpha}{\left(\frac{L\hat{x}}{L-1} + \frac{\alpha^2}{(L-1)\hat{x}}\right)}d\alpha. \tag{33}$$

Note that since $\hat{x} > 0$, we have $\frac{dx}{d\alpha} < 0$. This implies that as $\alpha$ increases from 0, $\hat{x}(\alpha)$ decreases from $\sigma_\star^{1/L}$.

**Computing the Differential $d\psi$.** Now we are position to derive the differential $d\psi$. Let us define $\Psi(\alpha) := \psi(\hat{x}(\alpha), \hat{y}(\alpha))$ as we ultimately want the behavior in terms of $\alpha$. Let us simplify $\Psi(\alpha)$ first:

$$\Psi(\alpha) := \psi(\hat{x}(\alpha), \hat{y}(\alpha)) = \sigma_\star^2 \left(\frac{1}{\hat{x}(\alpha)^2} + \frac{L-1}{\hat{y}(\alpha)^2}\right) \qquad \text{(From Equation (31))}$$

$$= \sigma_\star^2 \left(\frac{\hat{y}(\alpha)^2 + (L-1)\hat{x}(\alpha)^2}{\hat{x}(\alpha)^2\hat{y}(\alpha)^2}\right) \tag{34}$$

$$\implies \frac{\hat{y}^2}{L} + \left(1 - \frac{1}{L}\right)\hat{x}^2 = \frac{\Psi(\alpha)\hat{x}^2\hat{y}^2}{L\sigma_\star^2}. \tag{35}$$

Then, computing the differential, we have the following:

$$d\Psi = \sigma_\star^2 \left(-\frac{2}{\hat{x}^3}d\hat{x} - \frac{2(L-1)}{\hat{y}^3}d\hat{y}\right) \tag{36}$$

$$= \frac{1}{\hat{x}^3}\left[\frac{2\alpha\sigma_\star^2}{\frac{L\hat{x}}{L-1} + \frac{\alpha^2}{(L-1)\hat{x}}}\right]d\alpha - \left[\frac{(L-1)}{\hat{y}^3}\frac{2\alpha\hat{y}\sigma_\star^2}{(\hat{y}^2 L - \alpha^2(L-1))}\right]d\alpha \qquad \text{(Substitute } d\hat{x}, d\hat{y}\text{)}$$

$$= \left[\frac{1}{\hat{x}^4 + \left(\frac{\alpha^2}{L}\right)\hat{x}^2} - \frac{1}{\hat{y}^4 - \alpha^2\hat{y}^2\left(\frac{L-1}{L}\right)}\right] \cdot \frac{2\alpha(L-1)\sigma_\star^2}{L}d\alpha \tag{37}$$

$$= \left[\frac{\hat{y}^4 - \hat{x}^4 - \alpha^2\left(\frac{\hat{x}^2}{L} + \left(1 - \frac{1}{L}\right)\hat{y}^2\right)}{\left(\hat{x}^4 + \frac{\alpha^2}{L}\hat{x}^2\right) \cdot \left(\hat{y}^4 - \alpha^2\hat{y}\left(\frac{L-1}{L}\right)\right)}\right] \cdot \frac{2\alpha(L-1)\sigma_\star^2}{L}d\alpha. \tag{38}$$

Then, recall the intersection constraint:

$$\hat{y}^2 - \hat{x}^2 = \alpha^2 \implies (\hat{y}^2 - \hat{x}^2)(\hat{y}^2 + \hat{x}^2) = \alpha^2(\hat{y}^2 + \hat{x}^2) \tag{39}$$

$$\implies \hat{x}^4 - \hat{y}^4 = \alpha^2 \cdot (\hat{x}^2 + \hat{y}^2). \tag{40}$$

By substituting in Equation (40), we can simplify further:

$$d\Psi = \left[\frac{\alpha^2\left(\frac{\hat{y}^2}{L} + \left(1 - \frac{1}{L}\right)\hat{x}^2\right)}{(\hat{x}^4 + \frac{\alpha^2}{L}\hat{x}^2)(\hat{y}^4 - \alpha^2\hat{y}^2\left(\frac{L-1}{L}\right))}\right] \cdot \frac{2\alpha(L-1)\sigma_\star^2}{L}d\alpha.$$

Now, we can plug in Equation (35) into the numerator:

$$
d\Psi = \left[ \frac{\alpha^2 \left( \frac{\hat{y}^2}{L} + \left(1 - \frac{1}{L}\right)\hat{x}^2 \right)}{(\hat{x}^4 + \frac{\alpha^2}{L}\hat{x}^2)(\hat{y}^4 - \alpha^2 \hat{y}^2 \left(\frac{L-1}{L}\right))} \right] \cdot \frac{2\alpha(L-1)\sigma_\star^2}{L} d\alpha
$$

$$
= \left[ \frac{\alpha^2 \left( \frac{\Psi(\alpha)\hat{x}^2\hat{y}^2}{L\sigma_\star^2} \right)}{(\hat{x}^4 + \frac{\alpha^2}{L}\hat{x}^2)(\hat{y}^4 - \alpha^2 \hat{y}^2 \left(\frac{L-1}{L}\right))} \right] \cdot \frac{2\alpha(L-1)\sigma_\star^2}{L} d\alpha
$$

$$
= \left[ \frac{\alpha^2 \Psi(\alpha)\hat{x}^2\hat{y}^2}{L\sigma_\star^2(\hat{x}^4 + \frac{\alpha^2}{L}\hat{x}^2)(\hat{y}^4 - \alpha^2 \hat{y}^2 \left(\frac{L-1}{L}\right))} \right] \cdot \frac{2\alpha(L-1)\sigma_\star^2}{L} d\alpha
$$

$$
= \left[ \frac{2\Psi(\alpha)}{(\hat{x}^2 + \frac{\alpha^2}{L})(\hat{y}^2 - \alpha^2 \left(\frac{L-1}{L}\right))} \right] \cdot \left( \frac{1}{L} - \frac{1}{L^2} \right) \alpha^3 \, d\alpha.
$$

Finally, notice that from the conservation flow, we also have

$$
\hat{y}^2 - \hat{x}^2 = \alpha^2 \implies \hat{x}^2 + \frac{\alpha^2}{L} = \hat{y}^2 - \alpha^2 \left( \frac{L-1}{L} \right),
$$

and so

$$
d\Psi = \left[ \frac{2\Psi(\alpha)}{(\hat{x}^2 + \frac{\alpha^2}{L})^2} \right] \cdot \left( \frac{1}{L} - \frac{1}{L^2} \right) \alpha^3 \, d\alpha \implies \frac{d\Psi}{\Psi(\alpha)} = \underbrace{\left[ \frac{2}{(\hat{x}^2 + \frac{\alpha^2}{L})^2} \right] \cdot \left( \frac{1}{L} - \frac{1}{L^2} \right) \alpha^3 \, d\alpha}_{=:P(\alpha)}
$$

$$
= P(\alpha)\alpha^3 \, d\alpha.
$$

**Upper Bounding the Differential.** Note that it is difficult to directly solve for $\alpha$ from $P(\alpha)$, as $\hat{x}$ is also a function of $\alpha$. Hence, we can upper bound $P(\alpha)$ by a function $F(\alpha)$ such that $F(\alpha) \geq P(\alpha)$ for all $\alpha > 0$, and use this to solve for $\alpha$. We proceed by looking at the derivative of $P(\alpha)$:

$$
P'(\alpha) = \left[ \frac{-4}{(\hat{x}^2 + \frac{\alpha^2}{L})^3} \right] \left( \frac{1}{L} - \frac{1}{L^2} \right) \left( 2\hat{x}\frac{d\hat{x}}{d\alpha} + \frac{2\alpha}{L} \right)
$$

$$
= \left[ \frac{-4}{(\hat{x}^2 + \frac{\alpha^2}{L})^3} \right] \left( \frac{1}{L} - \frac{1}{L^2} \right) \left( \frac{2\alpha}{L} - \frac{2\hat{x}\alpha}{\frac{L\hat{x}}{L-1} + \frac{\alpha^2}{(L-1)\hat{x}}} \right)
$$

$$
= \frac{8\alpha}{(\hat{x}^2 + \frac{\alpha^2}{L})^3} \left( \frac{1}{L} - \frac{1}{L^2} \right) \left( \frac{L-1}{L + \frac{\alpha^2}{\hat{x}^2}} - \frac{1}{L} \right)
$$

**Case 1: $L = 2$.** Consider the case when $L = 2$. Then, notice that for all $\alpha > 0$, $P'(\alpha) < 0$. Thus, we can choose $F(\alpha)$ as such:

$$
F = \lim_{\alpha \to 0} P(\alpha) = \frac{2}{\sigma_\star^{4/L}} \left( \frac{1}{L} - \frac{1}{L^2} \right),
$$

which is constant in $\alpha$ that upper bounds $P(\alpha)$.

**Case 2: $L > 2$.** Now consider the general case. Notice that

$$
\hat{x}(\alpha) = \frac{\alpha}{\sqrt{L(L-2)}}
$$

is the only critical point of $P(\alpha)$ (since $\hat{x} > 0$). Furthermore, we have

$$
\hat{x}(\alpha) < \frac{\alpha}{\sqrt{L(L-2)}} \implies P'(\alpha) < 0,
$$

implying that $P(\alpha)$ is decreasing. Then, since $\hat{x}(\alpha)$ is also a decreasing function in $\alpha$, this means that there exists an $\alpha_{\text{crit}}$ such that for all $\alpha > \alpha_{\text{crit}}$, $P(\alpha)$ is always decreasing. We can find $\alpha_{\text{crit}}$ as such:

$$\hat{x}(\alpha_{\text{crit}}) = \frac{\alpha_{\text{crit}}}{\sqrt{L(L-2)}} \implies \hat{y}(\alpha_{\text{crit}}) = \alpha_{\text{crit}}\sqrt{1 + \frac{1}{L(L-2)}}.$$

By plugging these into our constraint set, we obtain

$$\left(\frac{\alpha_{\text{crit}}}{\sqrt{L(L-2)}}\right)\left(\alpha_{\text{crit}}\sqrt{1 + \frac{1}{L(L-2)}}\right)^{L-1} = \sigma_\star$$

$$\implies \alpha_{\text{crit}}^L \left(\sqrt{1 + \frac{1}{L(L-2)}}\right)^{L-1} = \sigma_\star\sqrt{L(L-2)}$$

$$\implies \alpha_{\text{crit}}^L = \frac{\sigma_\star\sqrt{L(L-2)}}{\left(\sqrt{1 + \frac{1}{L(L-2)}}\right)^{L-1}}$$

$$\implies \alpha_{\text{crit}} = \frac{\sigma_\star^{1/L}}{\left(\frac{1}{\sqrt{L(L-2)}}\left(1 + \frac{1}{L(L-2)}\right)^{\frac{L-1}{2}}\right)^{1/L}}.$$

Next, also note that for any $\alpha < \alpha_{\text{crit}}$, $P'(\alpha) > 0$, and so $P(\alpha)$ is increasing. Hence, $P(\alpha_{\text{crit}})$ corresponds to the maximum value of $P$. Therefore, we can choose $F = P(\alpha_{\text{crit}})$ as a constant function that upper bounds $P(\alpha)$. This leads to

$$F = P(\alpha_{\text{crit}}) = \left[\frac{2}{(\hat{x}(\alpha_{\text{crit}})^2 + \frac{\alpha_{\text{crit}}^2}{L})^2}\right] \cdot \left(\frac{1}{L} - \frac{1}{L^2}\right)$$

$$= \left[\frac{2}{\left(\frac{\alpha_{\text{crit}}^2}{L(L-2)} + \frac{\alpha_{\text{crit}}^2}{L}\right)^2}\right] \cdot \left(\frac{1}{L} - \frac{1}{L^2}\right)$$

$$= \left[\frac{2}{\left(\frac{(L-1)\alpha_{\text{crit}}^2}{L(L-2)}\right)^2}\right] \cdot \left(\frac{1}{L} - \frac{1}{L^2}\right)$$

$$= \frac{2}{\sigma_\star^{4/L}} \cdot \underbrace{\left(\frac{1}{L} - \frac{1}{L^2}\right)\left(\frac{L(L-2)}{L-1}\right)^2\left(\frac{1}{\sqrt{L(L-2)}}\left(1 + \frac{1}{L(L-2)}\right)^{\frac{L-1}{2}}\right)^{4/L}}_{=:h(L)}$$

$$= \frac{2h(L)}{\sigma_\star^{4/L}}.$$

**Combining Both Cases.** To avoid using two separate functions $F$ for different values of $L$, we can upper bound the function $h(L)$ to encompass both cases. This yields the following upper bound:

$$h(L) \leq L^2 \cdot \left(\left(1 + \frac{1}{L(L-2)}\right)^{\frac{L-1}{2}}\right)^{\frac{4}{L}} \leq L^2 \cdot 2^{\frac{2(L-1)}{L}} =: g(L).$$

Finally, we are left with the new differential

$$\frac{d\Psi}{\Psi(\alpha)} = \frac{2g(L)}{\sigma_\star^{4/L}}\alpha^3 \, d\alpha.$$

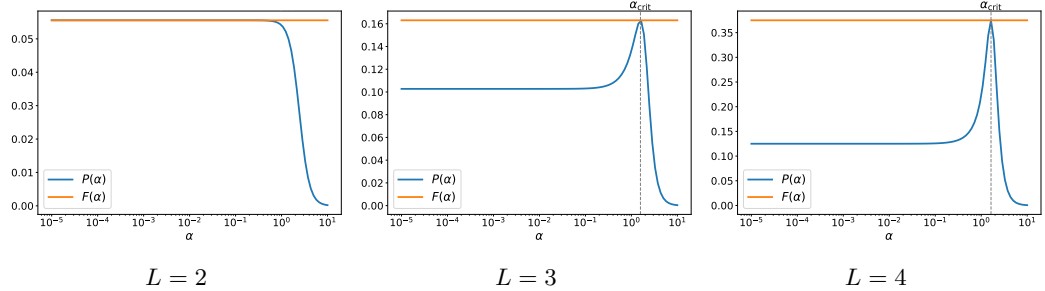

Figure 22: Plot of $P(\alpha)$ along with its upper bound evaluated at $F = P(\alpha_{\text{crit}})$ for different depths. The critical point occurs exactly at the computed value of $\alpha_{\text{crit}}$ and the function $F \geq P(\alpha)$ for all $\alpha > 0$.

**Finding Upper Bound on $\alpha$.** Firstly, we integrate the new differential:

$$\int \frac{d\Psi}{\Psi(\alpha)} = \frac{2g(L)}{\sigma_\star^{4/L}} \int \alpha^3 \, d\alpha \implies \ln\left(\frac{\Psi}{\Psi_0}\right) = \frac{g(L)\alpha^4}{2\sigma_\star^{4/L}}$$

$$\implies \Psi = \Psi_0 \exp\left(\frac{g(L)\alpha^4}{2\sigma_\star^{4/L}}\right),$$

where $\Psi_0 = \lim_{\alpha \to 0} \Psi(\alpha) = L\sigma_\star^{2-\frac{2}{L}}$. Now, we can solve for $\alpha$:

$$L\sigma_\star^{2-\frac{2}{L}} \exp\left(\frac{g(L)\alpha^4}{2\sigma_\star^{4/L}}\right) < \frac{2\sqrt{2}}{\eta} \implies \exp\left(\frac{g(L)\alpha^4}{2\sigma_\star^{4/L}}\right) < \frac{2\sqrt{2}}{\eta L\sigma_\star^{2-\frac{2}{L}}}$$

$$\implies \alpha < \left(\ln\left(\frac{\frac{2\sqrt{2}}{\eta}}{L\sigma_\star^{2-\frac{2}{L}}}\right) \cdot \frac{2\sigma_\star^{4/L}}{g(L)}\right)^{1/4}$$

$$\implies \alpha < \left(\ln\left(\frac{2\sqrt{2}}{\eta L\sigma_\star^{2-\frac{2}{L}}}\right) \cdot \frac{2\sigma_\star^{4/L}}{L^2 \cdot 2^{\frac{2(L-1)}{L}}}\right)^{1/4}$$

Simplifying further, we obtain

$$\alpha < \left(\ln\left(\frac{2\sqrt{2}}{\eta L\sigma_\star^{2-\frac{2}{L}}}\right) \cdot \frac{\sigma_\star^{4/L}}{L^2 \cdot 2^{\frac{2L-3}{L}}}\right)^{1/4},$$

which gives us the desired bound. This completes the proof. $\qquad\square$

**Lemma 6.** *Let $\pi(\mathbf{s}) := \prod_{\ell=1}^{L} \sigma_\ell$ denote the end-to-end product of $\mathbf{s} \in \mathbb{R}^L$ and suppose that each $\sigma_\ell > 0$. If the GFS sharpness $\psi(\mathbf{s}) \leq \frac{2\sqrt{1+c}}{\eta}$ for some $c \in (0, 1]$, then*

$$\sum_{i=1}^{\min\{2, L-1\}} \frac{\eta^2(\pi(\mathbf{s}) - \sigma_\star)^2\pi^2(\mathbf{s})}{\sigma_{[L-i]}^2\sigma_{[D]}^2} \leq 1 + c.$$

*Proof.* We consider two cases: (i) $\pi(\mathbf{s}) \in [0, \sigma_\star)$ and (ii) $\pi(\mathbf{s}) > \sigma_\star$. Note that we ignore the case of $\pi(\mathbf{s}) = \sigma_\star$ as this occurs with probability zero at EoS.

**Case 1** $(\pi(\mathbf{s}) \in [0, \sigma_\star))$. For this case, notice that we have

$$\sum_{i=1}^{\min\{2, L-1\}} \frac{\eta^2(\pi(\mathbf{s}) - \sigma_\star)^2\pi^2(\mathbf{s})}{\sigma_{L-i}^2\sigma_L^2} \leq \frac{\eta^2\pi^2(\mathbf{s})}{\sigma_{L-i}^2\sigma_L^2}. \qquad\qquad (\pi(\mathbf{s}) < \sigma_\star)$$

Then, note that the GFS sharpness is constant for all weights on the GF trajectory, as it is defined to be the sharpness at the limit of the GF trajectory (i.e., the GFS). Hence, we can focus on the weights at the solution, or global minima.

Define the GFS as $\mathbf{z} := S_{\mathrm{GF}}(\mathbf{s})$. By Lemma 2, each coordinate in $\mathbf{z} \in \mathbb{R}^L$ (and hence $\mathbf{s} \in \mathbb{R}^L$) is balanced across layers under GF, and so we have that

$$\sigma_\ell^2 - \sigma_m^2 = z_\ell^2 - z_m^2 \qquad \forall \ell, m \in [L].$$

Hence, it is suffices to show that

$$\sum_{i=1}^{\min\{2,L-1\}} \frac{\eta^2 \pi(\mathbf{z})^2}{z_{L-i}^2 z_L^2} \leq 1 + c \implies \sum_{i=1}^{\min\{2,L-1\}} \frac{\eta^2 \pi^2(\mathbf{s})}{\sigma_{L-i}^2 \sigma_L^2} \leq 1 + c.$$

Then, note that $\pi(\mathbf{z}) = \sigma_\star$, since it lies on the global minima, and so we have

$$\sum_{i=1}^{\min\{2,L-1\}} \frac{\eta^2 \pi^2(\mathbf{z})}{z_{L-i}^2 z_L^2} = \sum_{i=1}^{\min\{2,L-1\}} \frac{\eta^2 \sigma_\star^2}{z_{L-i}^2 z_L^2}. \tag{41}$$

From Lemma 3, the sharpness at the global minima is given as

$$\psi(\mathbf{s}) = \left\| \nabla^2 \mathcal{L}(\mathbf{z}) \right\| = \sum_{i=1}^{L} \frac{\sigma_\star^2}{z_i^2}. \tag{42}$$

This immediately implies that $\frac{\sigma_\star^2}{z_L^2} \leq \psi(\mathbf{s})$ and equivalently, $\exists \beta \in [0,1]$ such that $\frac{\sigma_\star^2}{z_L^2} = \beta \psi(\mathbf{s})$. Therefore, we have

$$\sum_{i=1}^{\min\{2,L-1\}} \frac{\sigma_\star^2}{z_{L-i}^2} \leq (1-\beta)\psi(\mathbf{s}). \tag{43}$$

Substituting Equations (42) and (43) into the expression we aim to bound, we obtain

$$\sum_{i=1}^{\min\{2,L-1\}} \frac{\eta^2(\pi(\mathbf{s}) - \sigma_\star^2)^2 \pi^2(\mathbf{s})}{\sigma_{L-i}^2 \sigma_L^2} = \sum_{i=1}^{\min\{2,L-1\}} \frac{\eta^2 \sigma_\star^2}{z_{L-i}^2 z_L^2} \leq \eta^2 \beta(1-\beta)\psi^2(\mathbf{s}) \leq \frac{\eta^2}{4}\psi^2(\mathbf{s}) \leq 1 + c,$$

where we used the fact that the maximum of $\beta(1-\beta)$ is $\frac{1}{4}$ when $\beta = \frac{1}{2}$ and $\psi(\mathbf{s}) \leq \frac{2\sqrt{1+c}}{\eta}$. Thus, if $\psi(\mathbf{s}) \leq \frac{2\sqrt{1+c}}{\eta}$, then for every weight $\mathbf{s} \in \mathbb{R}^L$ lying on its GF trajectory, we have

$$\sum_{i=1}^{\min\{2,L-1\}} \frac{\eta^2(\pi(\mathbf{s}) - \sigma_\star)^2 \pi^2(\mathbf{s})}{\sigma_{L-i}^2 \sigma_L^2} \leq 1 + c.$$

**Case 2 ($\pi(\mathbf{s}) > \sigma_\star$).** Consider the case in which $\pi(\mathbf{s}) > \sigma_\star$. By assumption, note that we have $\sigma_i > 0$, which implies that each GD update will also remain positive:

$$\sigma_i - \eta(\pi(\mathbf{s}) - \sigma_\star)\pi(\mathbf{s})\frac{1}{\sigma_i} > 0.$$

From this, we get

$$2 > \frac{\eta(\pi(\mathbf{s}) - \sigma_\star)\pi(\mathbf{s})}{\sigma_i^2} > 0,$$

This implies that

$$\sum_{i=1}^{\min\{2,L-1\}} \frac{\eta^2(\pi(\mathbf{s}) - \sigma_\star)^2 \pi^2(\mathbf{s})}{\sigma_{L-i}^2 \sigma_L^2} \leq (1+c),$$

with $c = 1$.

Putting both cases together, we have that

$$\sum_{i=1}^{\min\{2,L-1\}} \frac{\eta^2(\pi(\mathbf{s}) - \sigma_\star)^2 \pi^2(\mathbf{s})}{\sigma_{L-i}^2 \sigma_L^2} \leq (1+c),$$

for $c \in (0,1]$, which completes the proof.

$\square$

### C.2.2 PROOF OF PROPOSITION 2

*Proof.* Consider the $i$-th index of the simplified loss in (5):

$$\frac{1}{2}\left(\prod_{\ell=1}^{L}\sigma_{\ell,i}-\sigma_{\star,i}\right)^2 =: \frac{1}{2}\left(\prod_{\ell=1}^{L}\sigma_{\ell}-\sigma_{\star}\right)^2,$$

and omit the dependency on $i$ for ease of exposition. Our goal is to show that the $L$-th singular value $\sigma_L$ initialized to zero become increasingly balanced to $\sigma_\ell$ which are initialized to $\alpha > 0$. To that end, let us define the balancing dynamics between $\sigma_i$ and $\sigma_j$ as $b_{i,j}(t+1) := \left(\sigma_i^{(t+1)}\right)^2 - \left(\sigma_j^{(t+1)}\right)^2$ and $\pi(\mathbf{s}(t)) := \prod_{\ell=1}^{L}\sigma_\ell(t)$ for the product of singular values at iteration $t$. Then, we can simplify the balancing dynamics as such:

$$b_{i,j}(t+1) = (\sigma_i(t+1))^2 - (\sigma_j(t+1))^2 \tag{44}$$

$$= \left(\sigma_i(t) - \eta\left(\pi(\mathbf{s}(t)) - \sigma_\star\right)\frac{\pi(\mathbf{s}(t))}{\sigma_i(t)}\right)^2 - \left(\sigma_j(t) - \eta\left(\pi(\mathbf{s}(t)) - \sigma_\star\right)\frac{\pi(\mathbf{s}(t))}{\sigma_j(t)}\right)^2 \tag{45}$$

$$= (\sigma_i(t))^2 - (\sigma_j(t))^2 + \eta^2\left(\pi(\mathbf{s}(t)) - \sigma_\star\right)^2\left(\frac{\pi^2(\mathbf{s}(t))}{(\sigma_i(t))^2} - \frac{\pi^2(\mathbf{s}(t))}{(\sigma_j(t))^2}\right) \tag{46}$$

$$= \left((\sigma_i(t))^2 - (\sigma_j(t))^2\right)\left(1 - \eta^2(\pi(\mathbf{s}(t)) - \sigma_\star)^2\frac{\pi^2(\mathbf{s}(t))}{(\sigma_i(t))^2(\sigma_j(t))^2}\right) \tag{47}$$

$$= b_{i,j}(t)\left(1 - \eta^2(\pi(\mathbf{s}(t)) - \sigma_\star)^2\frac{\pi^2(\mathbf{s}(t))}{(\sigma_i(t))^2(\sigma_j(t))^2}\right). \tag{48}$$

Then, in order to show that $|b_{i,j}(t+1)| < c\,|b_{i,j}(t)|$ for some $0 < c \le 1$, we need to prove that

$$\left|1 - \eta^2(\pi(\mathbf{s}(t)) - \sigma_\star)^2\frac{\pi^2(\mathbf{s}(t))}{(\sigma_i(t))^2(\sigma_j(t))^2}\right| < c,$$

for all iterations $t$. Note that for our case, it is sufficient to show the result for $i = L$ and $j = \ell$ for any $\ell \ne L$. WLOG, suppose that the $\sigma$ are sorted such that $\sigma_1 \ge \sigma_2 \ge \ldots \ge \sigma_L$. By assumption, since our initialization scale satisfies

$$0 < \alpha < \left(\ln\left(\frac{2\sqrt{2}}{\eta L\sigma_\star^{2-\frac{2}{L}}}\right)\cdot\frac{\sigma_\star^{4/L}}{L^2\cdot 2^{\frac{2L-3}{L}}}\right)^{1/4},$$

by Lemma 5, we have that the GFS sharpness $\psi(\cdot)$ for positive $\mathbf{s} = [\sigma_1 \quad \cdots \quad \sigma_L] \in \mathbb{R}^L$ (i.e., each element $\sigma_\ell > 0$) satisfies $\psi(\mathbf{s}) < \frac{2\sqrt{2}}{\eta}$. Then, by Lemma 6, we have

$$\sum_{i=1}^{\min\{2,L-1\}}\frac{\eta^2(\pi(\mathbf{s}) - \sigma_\star)^2\pi^2(\mathbf{s})}{\sigma_{[L-i]}^2\sigma_{[D]}^2} \le 1 + c, \tag{49}$$

for some $c \in [0, 1)$. Then, notice that Equation (49) implies that

$$\frac{\eta^2(\pi(\mathbf{s}) - \sigma_\star)^2\pi^2(\mathbf{s})}{\sigma_{L-1}^2\sigma_L^2} < 1 + c \quad \text{and} \quad \frac{\eta^2(\pi(\mathbf{s}) - \sigma_\star)^2\pi^2(\mathbf{s})}{\sigma_i^2\sigma_j^2} < \frac{1+c}{2}, \tag{50}$$

for all $i \in [L]$, $j \in [L-2]$ and $i < j$. Notice that the latter inequality comes from the fact that

$$\frac{\eta^2(\pi(\mathbf{s}) - \sigma_\star)^2\pi^2(\mathbf{s})}{\sigma_{L-2}^2\sigma_L^2} + \frac{\eta^2(\pi(\mathbf{s}) - \sigma_\star)^2\pi^2(\mathbf{s})}{\sigma_{L-2}^2\sigma_L^2} < \frac{\eta^2(\pi(\mathbf{s}) - \sigma_\star)^2\pi^2(\mathbf{s})}{\sigma_{L-1}^2\sigma_L^2} + \frac{\eta^2(\pi(\mathbf{s}) - \sigma_\star)^2\pi^2(\mathbf{s})}{\sigma_{L-2}^2\sigma_L^2}$$

$$< 1 + c,$$

which implies that

$$2\frac{\eta^2(\pi(\mathbf{s})-\sigma_\star)^2\pi^2(\mathbf{s})}{\sigma_{L-2}^2\sigma_L^2} < 1+c \implies \frac{\eta^2(\pi(\mathbf{s})-\sigma_\star)^2\pi^2(\mathbf{s})}{\sigma_{L-2}^2\sigma_L^2} < \frac{1+c}{2},$$

and since $\sigma$ are sorted, it holds for all other $\sigma$. Therefore from Equation (48), we have for all $i \in [L-2]$,

$$b_{i,i+1}(t+1) < c \cdot b_{i,i+1}(t) \quad \text{and} \quad b(t+1)_{L-2,L} < c \cdot b_{L-2,L}(t),$$

as well as

$$-c \cdot b_{L-1,L}(t) < b_{L-1,L}(t+1) < c \cdot b_{L-1,L}(t).$$

Then, notice that since we initialized all of the singular values $\sigma_\ell$ for $\ell \in [L-1]$ to be the same, they follow the same dynamics. Since we already showed that $|b_{L-1,L}(t+1)| < c \cdot |b_{L-1,L}(t)|$, it must follow that

$$|b_{i,j}(t+1)| < c \cdot |b_{i,j}(t)|, \quad \forall i,j \in [L].$$

This completes the proof.

$\square$

## C.3    Proofs for Periodic Orbits

Before presenting our proof for Theorem 1, we first show that the required condition from Chen & Bruna (2023) for stable oscillations to occur (see Lemma 11) is also satisfied for DLNs beyond the EOS, as shown in Appendix C.3.1.

### C.3.1    Supporting Lemmas

**Lemma 7** (Stable Subspace Oscillations). *Define $S_p := L\sigma_{\star,p}^{2-\frac{2}{L}}$ and $K_p' := \max\left\{S_{p+1}, \frac{S_p}{2\sqrt{2}}\right\}$. If we run GD on the deep matrix factorization loss in (1) with learning rate $\eta = \frac{2}{K}$, where $K_p' < K < S_p$, then 2-period orbit oscillation occurs in the direction of $\Delta_{S_p}$, where $\Delta_{S_p}$ denotes the eigenvector associated with the eigenvalue $S_p$ of the Hessian at the balanced minimum.*

*Proof.* Define $f_{\Delta_i}$ as the 1-D function at the cross section of the loss landscape and the line following the direction of $\Delta_i$ passing the (balanced) minima, where $\Delta_i$ is the $i$-th eigenvector of the training loss Hessian at the balanced minimum. To prove the result, we will invoke Lemma 11, which states that two-period orbit oscillation occurs in the direction of $\Delta_i$ if the minima of $f_{\Delta_i}$ satisfies $f_{\Delta_i}^{(3)} > 0$ and $3[f_{\Delta_i}^{(3)}]^2 - f_{\Delta_i}^{(2)}f_{\Delta_i}^{(4)} > 0$, for $\eta > \frac{2}{\lambda_i}$. We show that while the condition holds for all of the eigenvector directions, the oscillations can only occur specifically in the directions of $\Delta_{S_i}$.

First, we will derive the eigenvectors of the Hessian of the training loss at convergence (i.e., $\mathbf{M}_\star = \mathbf{W}_{L:1}$). To obtain the eigenvectors of the Hessian of parameters $(\mathbf{W}_L, \ldots, \mathbf{W}_2, \mathbf{W}_1)$, consider a small perturbation of the parameters:

$$\boldsymbol{\Theta} := (\Delta\mathbf{W}_\ell + \mathbf{W}_\ell)_{\ell=1}^L = (\mathbf{W}_L + \Delta\mathbf{W}_L, \ldots, \mathbf{W}_2 + \Delta\mathbf{W}_2, \mathbf{W}_1 + \Delta\mathbf{W}_1).$$

Given that $\mathbf{W}_{L:1} = \mathbf{M}_\star$, consider and evaluate the loss function at this minima:

$$\mathcal{L}(\boldsymbol{\Theta}) = \frac{1}{2}\left\| \sum_\ell \mathbf{W}_{L:\ell+1}\Delta\mathbf{W}_\ell\mathbf{W}_{\ell-1:1} \right. \tag{51}$$

$$\left. + \sum_{\ell<m} \mathbf{W}_{L:\ell+1}\Delta\mathbf{W}_\ell\mathbf{W}_{\ell-1:m+1}\Delta\mathbf{W}_m\mathbf{W}_{m-1:1} + \ldots + \Delta\mathbf{W}_{L:1} \right\|_F^2. \tag{52}$$

By expanding each of the terms and splitting by the orders of $\Delta\mathbf{W}_\ell$ (perturbation), we get that the second-order term is equivalent to

$$\Theta\left(\sum_{\ell=1}^{L}\|\Delta\mathbf{W}_\ell\|^2\right) : \quad \frac{1}{2}\left\|\sum_\ell \mathbf{W}_{L:\ell+1}\Delta\mathbf{W}_\ell\mathbf{W}_{\ell-1:1}\right\|_{\mathsf{F}}^2$$

$$\Theta\left(\sum_{\ell=1}^{L}\|\Delta\mathbf{W}_\ell\|^3\right) : \quad \mathrm{tr}\left[\left(\sum_\ell \mathbf{W}_{L:\ell+1}\Delta\mathbf{W}_\ell\mathbf{W}_{\ell-1:1}\right)^\top \left(\sum_{\ell<m} \mathbf{W}_{L:\ell+1}\Delta\mathbf{W}_\ell\mathbf{W}_{\ell-1:m+1}\Delta\mathbf{W}_m\mathbf{W}_{m-1:1}\right)\right]$$

$$\Theta\left(\sum_{\ell=1}^{L}\|\Delta\mathbf{W}_\ell\|^4\right) : \quad \frac{1}{2}\|\sum_{\ell<m} \mathbf{W}_{L:\ell+1}\Delta\mathbf{W}_\ell\mathbf{W}_{\ell-1:m+1}\Delta\mathbf{W}_m\mathbf{W}_{m-1:1}\|_{\mathsf{F}}^2$$

$$+\,\mathrm{tr}\left[\sum_l \left(\mathbf{W}_{L:\ell+1}\Delta\mathbf{W}_\ell\mathbf{W}_{\ell-1:1}\right)^\top \left(\sum_{l<m<p} \mathbf{W}_{L:\ell+1}\Delta\mathbf{W}_\ell\mathbf{W}_{\ell-1:m+1}\Delta\mathbf{W}_m\mathbf{W}_{m-1:p+1}\Delta\mathbf{W}_p\mathbf{W}_{p-1:1}\right)\right]$$

The direction of the steepest change in the loss at the minima correspond to the largest eigenvector direction of the Hessian. Since higher order terms such as $\Theta\left(\sum_{\ell=1}^{L}\|\Delta\mathbf{W}_\ell\|^3\right)$ are insignifcant compared to the second order terms $\Theta\left(\sum_{\ell=1}^{L}\|\Delta\mathbf{W}_\ell\|^2\right)$, finding the direction that maximizes the second order term leads to finding the eigenvector of the Hessian. Then, the eigenvector corresponding to the maximum eigenvalue of $\nabla^2\mathcal{L}$ is the solution of

$$\Delta_1 := \mathrm{vec}(\Delta\mathbf{W}_L,\ldots\Delta\mathbf{W}_1) = \underset{\|\Delta\mathbf{W}_L\|_{\mathsf{F}}^2+\ldots+\|\Delta\mathbf{W}_1\|_{\mathsf{F}}^2=1}{\mathrm{argmax}} f\left(\Delta\mathbf{W}_L,\ldots,\Delta\mathbf{W}_1\right), \qquad (53)$$

where

$$f(\Delta\mathbf{W}_L,\ldots,\Delta\mathbf{W}_1) := \frac{1}{2}\|\Delta\mathbf{W}_L\mathbf{W}_{L-1:1} + \ldots + \mathbf{W}_{L:3}\Delta\mathbf{W}_2\mathbf{W}_1 + \mathbf{W}_{L:2}\Delta\mathbf{W}_1\|_{\mathsf{F}}^2. \qquad (54)$$

While the solution of Equation (53) gives the maximum eigenvector direction of the Hessian, $\Delta_1$, the other eigenvectors can be found by solving

$$\Delta_r := \underset{\substack{\|\Delta\mathbf{W}_L\|_{\mathsf{F}}^2+\ldots+\|\Delta\mathbf{W}_1\|_{\mathsf{F}}^2=1,\\ \Delta_r\perp\Delta_{r-1},..,\Delta_r\perp\Delta_1}}{\mathrm{argmax}} f\left(\Delta\mathbf{W}_L,\ldots,\Delta\mathbf{W}_1\right). \qquad (55)$$

By expanding $f(\cdot)$, we have that

$$f(\Delta\mathbf{W}_L,\ldots,\Delta\mathbf{W}_1) = \|\Delta\mathbf{W}_L\mathbf{W}_{L-1:1}\|_{\mathsf{F}}^2 + \ldots + \|\mathbf{W}_{L:3}\Delta\mathbf{W}_2\mathbf{W}_1\|_{\mathsf{F}}^2 + \|\mathbf{W}_{L:2}\Delta\mathbf{W}_1\|_{\mathsf{F}}^2$$
$$+ \mathrm{tr}\left[(\Delta\mathbf{W}_L\mathbf{W}_{L-1:1})^\top (\mathbf{W}_{L:3}\Delta\mathbf{W}_2\mathbf{W}_1 + \ldots + \mathbf{W}_{L:2}\Delta\mathbf{W}_1)\right] + \ldots +$$
$$\mathrm{tr}\left[(\mathbf{W}_{L:2}\Delta\mathbf{W}_1)^\top (\mathbf{W}_{L:3}\Delta\mathbf{W}_2\mathbf{W}_1 + \ldots + \mathbf{W}_{L:3}\Delta\mathbf{W}_2\mathbf{W}_1)\right]. \qquad (56)$$

We can solve Equation (53) by maximizing each of the terms, which can be done in two steps:

(i) Each Frobenius term in the expansion is maximized when the left singular vector of $\Delta\mathbf{W}_\ell$ aligns with $\mathbf{W}_{L:\ell+1}$ and the right singular vector aligns with $\mathbf{W}_{\ell-1:1}$. This is a result of Von Neumann's trace inequality (Mirsky, 1975). Similarly, each term in the trace is maximized when the singular vector of the perturbations align with the products.

(ii) Due to the alignment, Equation (53) can be written in just the singular values. Let $\Delta s_{\ell,i}$ denote the $i$-th singular value of the perturbation matrix $\Delta\mathbf{W}_\ell$. Recall that all of the singular values of $\mathbf{M}_\star$ are distinct (i.e., $\sigma_{\star,1} > \ldots > \sigma_{\star,r}$). Hence, it is easy to see that Equation (53) is maximized when $\Delta s_{\ell,i} = 0$ (i.e, all the weight goes to $\Delta s_{\ell,1}$). Thus, each perturbation matrix must be rank-1.

Now since each perturbation is rank-1, we can write each perturbation as

$$\Delta \mathbf{W}_\ell = \Delta s_\ell \Delta \mathbf{u}_\ell \Delta \mathbf{v}_\ell^\top, \quad \forall \ell \in [L], \tag{57}$$

for $\Delta s_\ell > 0$ and orthonormal vectors $\Delta \mathbf{u}_\ell \in \mathbb{R}^d$ and $\Delta \mathbf{v}_\ell \in \mathbb{R}^d$ with $\sum_{\ell=1}^L \Delta s_\ell^2 = 1$. Plugging this in each term, we obtain:

$$\|\mathbf{W}_{L:\ell+1}\Delta_1 \mathbf{W}_\ell \mathbf{W}_{\ell-1:1}\|_2^2 = \Delta_1 s_\ell^2 \cdot \left\| \underbrace{\mathbf{V}_\star \sigma_\star^{\frac{L-\ell}{L}} \mathbf{V}_\star^\top \Delta \mathbf{u}_\ell}_{=:\mathbf{a}} \underbrace{\Delta \mathbf{v}_\ell^\top \mathbf{V}_\star \sigma_\star^{\frac{\ell-1}{L}} \mathbf{V}_\star^\top}_{=:\mathbf{b}^\top} \right\|_2^2.$$

Since alignment maximizes this expression as discussed in first point, we have:

$\mathbf{u}_\ell = \mathbf{v}_\ell = \mathbf{v}_{\star,1}$ for all $\ell \in [2, L-1]$, then

$$\mathbf{a} = \sigma_{\star,1}^{\frac{L-\ell}{L}} \mathbf{v}_{\star,1} \quad \text{and} \quad \mathbf{b}^\top = \sigma_{\star,1}^{\frac{\ell-1}{L}} \mathbf{v}_{\star,1}^\top \implies \mathbf{a}\mathbf{b}^\top = \sigma_{\star,1}^{1-\frac{1}{L}} \cdot \mathbf{v}_{\star,1}\mathbf{v}_{\star,1}^\top.$$

The very same argument can be made for the trace terms. Hence, in order to maximize $f(\cdot)$, we must have

$$\mathbf{v}_L = \mathbf{v}_{\star,1}, \quad \text{and} \quad \mathbf{u}_1 = \mathbf{v}_{\star,1},$$
$$\mathbf{u}_\ell = \mathbf{v}_\ell = \mathbf{v}_{\star,1}, \quad \forall \ell \in [2, L-1].$$

To determine $\mathbf{u}_L$ and $\mathbf{v}_1$, we can look at one of the trace terms:

$$\text{tr}\left[ (\Delta_1 \mathbf{W}_L \mathbf{W}_{L-1:1})^\top (\mathbf{W}_{L:3}\Delta_1 \mathbf{W}_2 \mathbf{W}_1 + \ldots + \mathbf{W}_{L:2}\Delta_1 \mathbf{W}_1) \right] \leq \left( \frac{L-1}{L} \right) \cdot \sigma_{\star,1}^{2-\frac{2}{L}}.$$

To reach the upper bound, we require $\mathbf{u}_L = \mathbf{u}_{\star,1}$ and $\mathbf{v}_1 = \mathbf{v}_{\star,1}$. Finally, as the for each index, the singular values are balanced, we will have $\Delta_1 s_\ell = \frac{1}{\sqrt{L}}$ for all $\ell \in [L]$ to satisfy the constraint. Finally, we get that the leading eigenvector is

$$\Delta_1 := \text{vec}\left( \frac{1}{\sqrt{L}}\mathbf{u}_1\mathbf{v}_1^\top, \frac{1}{\sqrt{L}}\mathbf{v}_1\mathbf{v}_1^\top, \ldots, \frac{1}{\sqrt{L}}\mathbf{v}_1\mathbf{v}_1^\top \right).$$

Notice that we can also verify that $f(\Delta_1) = L\sigma_{\star,1}^{2-\frac{2}{L}}$, which is the leading eigenvalue (or sharpness) derived in Lemma 1.

To derive the remaining eigenvectors, we need to find all of the vectors in which $\Delta_i^\top \Delta_j = 0$ for $i \neq j$, where

$$\Delta_i = \text{vec}(\Delta_i \mathbf{W}_L, \ldots \Delta_i \mathbf{W}_1),$$

and $f(\Delta_i) = \lambda_i$, where $\lambda_i$ is the $i$-th largest eigenvalue. By repeating the same process as above, we find that the eigenvector-eigenvalue pair as follows:

$$\Delta_1 = \text{vec}\left( \frac{1}{\sqrt{L}}\mathbf{u}_1\mathbf{v}_1^\top, \frac{1}{\sqrt{L}}\mathbf{v}_1\mathbf{v}_1^\top, \ldots, \frac{1}{\sqrt{L}}\mathbf{v}_1\mathbf{v}_1^\top \right), \quad \lambda_1 = L\sigma_{\star,1}^{2-\frac{2}{L}}$$

$$\Delta_2 = \text{vec}\left( \frac{1}{\sqrt{L}}\mathbf{u}_1\mathbf{v}_2^\top, \frac{1}{\sqrt{L}}\mathbf{v}_1\mathbf{v}_2^\top, \ldots, \frac{1}{\sqrt{L}}\mathbf{v}_1\mathbf{v}_2^\top \right), \quad \lambda_2 = \left( \sum_{i=0}^{L-1} \sigma_{\star,1}^{1-\frac{1}{L}-\frac{1}{L}i} \cdot \sigma_{\star,2}^{\frac{1}{L}i} \right)$$

$$\Delta_3 = \text{vec}\left( \frac{1}{\sqrt{L}}\mathbf{u}_2\mathbf{v}_1^\top, \frac{1}{\sqrt{L}}\mathbf{v}_2\mathbf{v}_1^\top, \ldots, \frac{1}{\sqrt{L}}\mathbf{v}_2\mathbf{v}_1^\top \right), \quad \lambda_3 = \left( \sum_{i=0}^{L-1} \sigma_{\star,1}^{1-\frac{1}{L}-\frac{1}{L}i} \cdot \sigma_{\star,2}^{\frac{1}{L}i} \right)$$

$$\vdots$$

$$\Delta_d = \text{vec}\left( \frac{1}{\sqrt{L}}\mathbf{u}_2\mathbf{v}_2^\top, \frac{1}{\sqrt{L}}\mathbf{v}_2\mathbf{v}_2^\top, \ldots, \frac{1}{\sqrt{L}}\mathbf{v}_2\mathbf{v}_2^\top \right), \quad \lambda_d = L\sigma_{\star,2}^{2-\frac{2}{L}}$$

$$\vdots$$

$$\Delta_{dr+r} = \text{vec}\left( \frac{1}{\sqrt{L}}\mathbf{u}_d\mathbf{v}_r^\top, \frac{1}{\sqrt{L}}\mathbf{v}_d\mathbf{v}_r^\top, \ldots, \frac{1}{\sqrt{L}}\mathbf{v}_d\mathbf{v}_r^\top \right),$$

which gives a total of $dr + r$ eigenvectors.

Second, equipped with the eigenvectors, let us consider the 1-D function $f_{\Delta_i}$ generated by the cross-section of the loss landscape and each eigenvector $\Delta_i$ passing the minima:

$$
\begin{aligned}
f_{\Delta_i}(\mu) &= \mathcal{L}(\mathbf{W}_L + \mu\Delta\mathbf{W}_L, \ldots, \mathbf{W}_2 + \mu\Delta\mathbf{W}_2, \mathbf{W}_1 + \mu\Delta\mathbf{W}_1), \\
&= \mu^2 \cdot \frac{1}{2}\|\Delta\mathbf{W}_L\mathbf{W}_{L-1:1} + \ldots + \mathbf{W}_{L:3}\Delta\mathbf{W}_2\mathbf{W}_1 + \mathbf{W}_{L:2}\Delta\mathbf{W}_1\|_{\mathsf{F}}^2 \\
&\quad + \mu^3 \cdot \sum_{\ell=1, \ell<m}^L \mathrm{tr}\left[(\mathbf{W}_{L:\ell+1}\Delta\mathbf{W}_\ell\mathbf{W}_{\ell-1:1})^\top (\mathbf{W}_{L:\ell+1}\Delta\mathbf{W}_\ell\mathbf{W}_{\ell-1:m+1}\Delta\mathbf{W}_m\mathbf{W}_{m-1:1})\right] \\
&\quad + \mu^4 \cdot \frac{1}{2}\left\|\left(\sum_{\ell<m}\mathbf{W}_{L:\ell+1}\Delta\mathbf{W}_\ell\mathbf{W}_{\ell-1:m+1}\Delta\mathbf{W}_m\mathbf{W}_{m-1:1}\right)\right\|_{\mathsf{F}}^2 \\
&\quad + \mu^4 \cdot \sum_{\ell<m<p}^L \mathrm{tr}\left[(\mathbf{W}_{L:\ell+1}\Delta\mathbf{W}_\ell\mathbf{W}_{\ell-1:1})^\top (\mathbf{W}_{L:\ell+1}\Delta\mathbf{W}_\ell\mathbf{W}_{\ell-1:m+1}\Delta\mathbf{W}_m\mathbf{W}_{m-1:p+1}\Delta\mathbf{W}_p\mathbf{W}_{p-1:1})\right].
\end{aligned}
$$

Then, the several order derivatives of $f_{\Delta_i}(\mu)$ at $\mu = 0$ can be obtained from Taylor expansion as

$$
\begin{aligned}
f_{\Delta_i}^{(2)}(0) &= \|\Delta_i\mathbf{W}_L\mathbf{W}_{L-1:1} + \ldots + \mathbf{W}_{L:3}\Delta_i\mathbf{W}_2\mathbf{W}_1 + \mathbf{W}_{L:2}\Delta_i\mathbf{W}_1\|_{\mathsf{F}}^2 = \lambda_i^2 \\
f_{\Delta_i}^{(3)}(0) &= 6\sum_{\ell=1}^L \mathrm{tr}\left[(\mathbf{W}_{L:\ell+1}\Delta_i\mathbf{W}_\ell\mathbf{W}_{\ell-1:1})^\top (\mathbf{W}_{L:\ell+2}\Delta_i\mathbf{W}_{\ell+1}\mathbf{W}_\ell\Delta_i\mathbf{W}_{\ell-1}\mathbf{W}_{\ell-2:1})\right] \\
&= 6\left\|\sum_\ell \mathbf{W}_{L:\ell+1}\Delta_i\mathbf{W}_\ell\mathbf{W}_{\ell-1:1}\right\|_{\mathsf{F}} \cdot \left\|\left(\sum_{\ell<m}\mathbf{W}_{L:\ell+1}\Delta\mathbf{W}_\ell\mathbf{W}_{\ell-1:m+1}\Delta\mathbf{W}_m\mathbf{W}_{m-1:1}\right)\right\|_{\mathsf{F}} \\
&\coloneqq 6\lambda_i \cdot \beta_i \\
f_{\Delta_i}^{(4)}(0) &= 12\|\Delta_i\mathbf{W}_L\Delta_i\mathbf{W}_{L-1}\mathbf{W}_{L-2:1} + \ldots + \mathbf{W}_{L:4}\Delta_i\mathbf{W}_3\mathbf{W}_2\Delta_i\mathbf{W}_1 + \mathbf{W}_{L:3}\Delta_i\mathbf{W}_2\Delta_i\mathbf{W}_1\|_{\mathsf{F}}^2 \\
&\quad + 24\sum_{\ell=1}^L \mathrm{tr}\left[(\mathbf{W}_{L:\ell+1}\Delta_i\mathbf{W}_\ell\mathbf{W}_{\ell-1:1})^\top \left(\sum_{l<m<p}\mathbf{W}_{L:\ell+1}\Delta\mathbf{W}_\ell\mathbf{W}_{\ell-1:m+1}\Delta\mathbf{W}_m\mathbf{W}_{m-1:p+1}\Delta\mathbf{W}_p\mathbf{W}_{p-1:1}\right)\right] \\
&\coloneqq 12\beta_i^2 + 24\lambda_i \cdot \delta_i,
\end{aligned}
$$

where we defined

$$
\lambda_i = \left\|\sum_\ell \mathbf{W}_{L:\ell+1}\Delta_i\mathbf{W}_\ell\mathbf{W}_{\ell-1:1}\right\|_{\mathsf{F}} \qquad\qquad \text{(Total } \binom{L}{1} \text{ terms)}
$$

$$
\beta_i = \left\|\left(\sum_{\ell<m}\mathbf{W}_{L:\ell+1}\Delta\mathbf{W}_\ell\mathbf{W}_{\ell-1:m+1}\Delta\mathbf{W}_m\mathbf{W}_{m-1:1}\right)\right\|_{\mathsf{F}} \qquad\qquad \text{(Total } \binom{L}{2} \text{ terms)}
$$

$$
\delta_i = \left\|\left(\sum_{l<m<p}\mathbf{W}_{L:\ell+1}\Delta\mathbf{W}_\ell\mathbf{W}_{\ell-1:m+1}\Delta\mathbf{W}_m\mathbf{W}_{m-1:p+1}\Delta\mathbf{W}_p\mathbf{W}_{p-1:1}\right)\right\|_{\mathsf{F}},
$$

$$
\text{(Total } \binom{L}{3} \text{ terms)}
$$

and used the fact that $\mathrm{tr}(\mathbf{A}^\top\mathbf{B}) = \|\mathbf{A}\|_{\mathsf{F}} \cdot \|\mathbf{B}\|_{\mathsf{F}}$ under singular vector alignment.

Then, since $\beta_i$ has $\binom{L}{2}$ terms inside the sum, when the Frobenium term is expanded, it will have $\frac{\binom{L}{2}(\binom{L}{2}+1)}{2}$ number of terms. Under alignment and balancedness, $\beta_i^2 = \Delta s_\ell^2 \sigma_i^{2-\frac{4}{L}} \times \frac{\binom{L}{2}(\binom{L}{2}+1)}{2}$

and $\lambda_i \delta_i = \Delta s_\ell^2 \sigma_i^{2-\frac{4}{L}} \times \binom{L}{3} L$. Thus, we have the expression

$$2\beta_i^2 - \lambda_i \delta_i = \Delta s_\ell^2 \sigma_i^{2-\frac{4}{L}} \left( 2 \frac{\binom{L}{2}\left(\binom{L}{2}+1\right)}{2} - \binom{L}{3} L \right)$$

$$= \Delta s_\ell^2 \sigma_i^{2-\frac{4}{L}} \binom{L}{3} L \times \left( \frac{3\left(\frac{L(L-1)}{2}+1\right)}{L(L-2)} - 1 \right)$$

$$= \Delta s_\ell^2 \sigma_i^{2-\frac{4}{L}} \frac{2\binom{L}{3} L}{L(L-2)} \times \left( (L-1)^2 + 5 \right) > 0,$$

for any depth $L > 2$. Finally, the condition of stable oscillation of 1-D function is

$$3[f_{\Delta_i}^{(3)}]^2 - f_{\Delta_i}^{(2)} f_{\Delta_i}^{(4)} = 108\lambda_i^2 \beta_i^2 - (\lambda_i^2)(12\beta_i^2 + 24(2\lambda_i)(\delta_i)) = 48\lambda_i^2 (2\beta_i^2 - \lambda_i \delta_i) > 0,$$

which we have proven to be positive for any depth $L > 2$, for all the eigenvector directions corresponding to the non-zero eigenvalues. Lastly, by Proposition 3, notice that we can write the vectorized weights in the form

$$\widetilde{\Delta} := \text{vec}\left( \mathbf{W}_L, \mathbf{W}_{L-1}, \ldots, \mathbf{W}_1 \right)$$

$$= \text{vec}\left( \mathbf{U}_\star \boldsymbol{\Sigma}_L \mathbf{V}_\star^\top, \mathbf{V}_\star \boldsymbol{\Sigma}_{L-1} \mathbf{V}_\star^\top, \ldots, \mathbf{V}_\star \boldsymbol{\Sigma}_1 \mathbf{V}_\star^\top \right)$$

$$= \sum_{i=1}^d \text{vec}\left( \sigma_{L,i} \cdot \mathbf{u}_{\star,i} \mathbf{v}_{\star,i}^\top, \sigma_{L-1,i} \cdot \mathbf{v}_{\star,i} \mathbf{v}_{\star,i}^\top, \ldots, \sigma_{1,i} \cdot \mathbf{v}_{\star,i} \mathbf{v}_{\star,i}^\top \right).$$

Then, $\Delta_i^\top \widetilde{\Delta} \neq 0$ only in the eigenvector directions that correspond to the eigenvalues of the form $S_i = L\sigma_{\star,i}^{2-2/L}$. Hence, the oscillations can only occur in the direction of $\Delta_{S_i}$, where $\Delta_{S_i}$ are the eigenvectors corresponding to the eigenvalues $S_i$. This completes the proof.

$\square$

### C.3.2 PROOF OF LEMMA 1

*Proof.* By Proposition 3, notice that we can re-write the loss in Equation (1) as

$$\frac{1}{2} \|\mathbf{W}_{L:1} - \mathbf{M}_\star\|_\mathsf{F}^2 = \frac{1}{2} \|\boldsymbol{\Sigma}_{L:1} - \boldsymbol{\Sigma}_\star\|_\mathsf{F}^2,$$

where $\boldsymbol{\Sigma}_{L:1}$ are the singular values of $\mathbf{W}_{L:1}$. We will first show that the eigenvalues of the Hessian with respect to the weight matrices $\mathbf{W}_\ell$ are equivalent to those of the Hessian taken with respect to its singular values $\boldsymbol{\Sigma}_\ell$. To this end, consider the vectorized form of the loss:

$$f(\boldsymbol{\Theta}) := \frac{1}{2} \|\mathbf{W}_{L:1} - \mathbf{M}_\star\|_\mathsf{F}^2 = \frac{1}{2} \|\text{vec}(\mathbf{W}_{L:1}) - \text{vec}(\mathbf{M}_\star)\|_2^2.$$

Then, each block of the Hessian $\nabla_{\boldsymbol{\Theta}}^2 f(\boldsymbol{\Theta}) \in \mathbb{R}^{d^2 L \times d^2 L}$ with respect to the vectorized parameters is given as

$$\left[ \nabla_{\boldsymbol{\Theta}}^2 f(\boldsymbol{\Theta}) \right]_{m,\ell} = \nabla_{\text{vec}(\mathbf{W}_m)} \nabla_{\text{vec}(\mathbf{W}_\ell)}^\top f(\boldsymbol{\Theta}) \in \mathbb{R}^{d^2 \times d^2}.$$

By the vectorization trick, each vectorized layer matrix has an SVD of the form $\text{vec}(\mathbf{W}_\ell) = \text{vec}(\mathbf{U}_\ell \boldsymbol{\Sigma}_\ell \mathbf{V}_\ell^\top) = (\mathbf{V}_\ell \otimes \mathbf{U}_\ell) \cdot \text{vec}(\boldsymbol{\Sigma}_\ell)$. Then, notice that we have

$$\nabla_{\text{vec}(\mathbf{W}_\ell)} f(\boldsymbol{\Theta}(t)) = (\mathbf{V}_\ell \otimes \mathbf{U}_\ell) \cdot \nabla_{\text{vec}(\boldsymbol{\Sigma}_\ell)} f(\boldsymbol{\Theta}(t)),$$

which gives us that each block of the Hessian is given by

$$\left[ \nabla_{\boldsymbol{\Theta}}^2 f(\boldsymbol{\Theta}) \right]_{m,\ell} = \nabla_{\text{vec}(\mathbf{W}_m)} \nabla_{\text{vec}(\mathbf{W}_\ell)}^\top f(\boldsymbol{\Theta})$$

$$= (\mathbf{V}_m \otimes \mathbf{U}_m) \cdot \underbrace{\nabla_{\text{vec}(\boldsymbol{\Sigma}_m)} \nabla_{\text{vec}(\boldsymbol{\Sigma}_\ell)}^\top f(\boldsymbol{\Theta})}_{=: \mathbf{H}_{m,\ell}} \cdot (\mathbf{V}_\ell \otimes \mathbf{U}_\ell)^\top.$$

Then, since the Kronecker product of two orthogonal matrices is also an orthogonal matrix by Lemma 9, we can write the overall Hessian matrix as

$$
\widetilde{\mathbf{H}} = \begin{bmatrix}
\mathbf{R}_1 \mathbf{H}_{1,1} \mathbf{R}_1 & \mathbf{R}_1 \mathbf{H}_{1,2} \mathbf{R}_2 & \dots & \mathbf{R}_1 \mathbf{H}_{1,L} \mathbf{R}_L \\
\mathbf{R}_2 \mathbf{H}_{2,1} \mathbf{R}_1 & \mathbf{R}_2 \mathbf{H}_{2,2} \mathbf{R}_2 & \dots & \mathbf{R}_2 \mathbf{H}_{2,L} \mathbf{R}_L \\
\vdots & \vdots & \ddots & \vdots \\
\mathbf{R}_L \mathbf{H}_{L,1} \mathbf{R}_1 & \mathbf{R}_L \mathbf{H}_{L,2} \mathbf{R}_2 & \dots & \mathbf{R}_L \mathbf{H}_{L,L} \mathbf{R}_L
\end{bmatrix},
$$

for orthogonal matrices $\{\mathbf{R}_\ell\}_{\ell=1}^L$. Then, by Lemma 8, the eigenvalues of $\widetilde{\mathbf{H}}$ are the same as those of $\mathbf{H}$, where $\mathbf{H} \in \mathbb{R}^{d^2 L \times d^2 L}$ is the Hessian matrix with respect to the vectorized $\mathbf{\Sigma}_\ell$:

$$
\mathbf{H} = \begin{bmatrix}
\mathbf{H}_{1,1} & \mathbf{H}_{1,2} & \dots & \mathbf{H}_{L,1} \\
\mathbf{H}_{2,1} & \mathbf{H}_{2,2} & \dots & \mathbf{H}_{L,2} \\
\vdots & \vdots & \ddots & \vdots \\
\mathbf{H}_{1,L} & \mathbf{H}_{2,L} & \dots & \mathbf{H}_{L,L}
\end{bmatrix}.
$$

Now, we can consider the following vectorized loss:

$$
\begin{aligned}
f(\mathbf{\Theta}) = \frac{1}{2}\|\mathbf{\Sigma}_{L:1} - \mathbf{\Sigma}_\star\|_{\mathsf{F}}^2 &= \frac{1}{2}\left\|\operatorname{vec}\left(\mathbf{\Sigma}_{L:1} - \mathbf{\Sigma}_\star\right)\right\|_2^2 \\
&= \frac{1}{2}\|\underbrace{\left(\mathbf{\Sigma}_{\ell-1:1}^\top \otimes \mathbf{\Sigma}_{L:\ell+1}\right)}_{=:\mathbf{A}_\ell} \cdot \operatorname{vec}(\mathbf{\Sigma}_\ell) - \operatorname{vec}(\mathbf{\Sigma}_\star)\|_2^2.
\end{aligned}
$$

Then, the gradient with respect to $\operatorname{vec}(\mathbf{\Sigma}_\ell)$ is given by

$$
\nabla_{\operatorname{vec}(\mathbf{\Sigma}_\ell)} f(\mathbf{\Theta}) = \mathbf{A}_\ell^\top \left(\mathbf{A}_\ell \cdot \operatorname{vec}(\mathbf{\Sigma}_\ell) - \operatorname{vec}(\mathbf{\Sigma}_\star)\right).
$$

Then, for $m = \ell$, we have

$$
\mathbf{H}_{\ell,\ell} = \nabla_{\operatorname{vec}(\mathbf{\Sigma}_\ell)}^2 f(\mathbf{\Theta}) = \mathbf{A}_\ell^\top \mathbf{A}_\ell.
$$

For $m \neq \ell$, we have

$$
\begin{aligned}
\mathbf{H}_{m,\ell} = \nabla_{\operatorname{vec}(\mathbf{\Sigma}_m)} \nabla_{\operatorname{vec}(\mathbf{\Sigma}_\ell)} f(\mathbf{\Theta}) &= \nabla_{\operatorname{vec}(\mathbf{\Sigma}_m)} \left[\mathbf{A}_\ell^\top \left(\mathbf{A}_\ell \operatorname{vec}(\mathbf{\Sigma}_\ell) - \operatorname{vec}(\mathbf{M}^\star)\right)\right] \\
&= \nabla_{\operatorname{vec}(\mathbf{\Sigma}_m)} \mathbf{A}_\ell^\top \cdot \underbrace{\left(\mathbf{A}_\ell \operatorname{vec}(\mathbf{\Sigma}_\ell) - \operatorname{vec}(\mathbf{M}^\star)\right)}_{=0 \text{ at convergence}} + \mathbf{A}_\ell^\top \cdot \nabla_{\operatorname{vec}(\mathbf{\Sigma}_m)}(\mathbf{A}_\ell \operatorname{vec}(\mathbf{\Sigma}_\ell) - \operatorname{vec}(\mathbf{M}^\star)) \\
&= \mathbf{A}_\ell^\top \mathbf{A}_m,
\end{aligned}
$$

where we have used the product rule along with the fact that $\mathbf{A}_\ell \operatorname{vec}(\mathbf{\Sigma}_\ell) = \mathbf{A}_m \operatorname{vec}(\mathbf{\Sigma}_m)$.

Overall, the Hessian at convergence for GD is given by

$$
\mathbf{H} = \begin{bmatrix}
\mathbf{A}_1^\top \mathbf{A}_1 & \mathbf{A}_1^\top \mathbf{A}_2 & \dots & \mathbf{A}_1^\top \mathbf{A}_L \\
\mathbf{A}_2^\top \mathbf{A}_1 & \mathbf{A}_2^\top \mathbf{A}_2 & \dots & \mathbf{A}_2^\top \mathbf{A}_L \\
\vdots & \vdots & \ddots & \vdots \\
\mathbf{A}_L^\top \mathbf{A}_1 & \mathbf{A}_L^\top \mathbf{A}_2 & \dots & \mathbf{A}_L^\top \mathbf{A}_L
\end{bmatrix}
$$

Now, we can derive an explicit expression for each $\mathbf{A}_{m,\ell}$ by considering the implicit balancing effect of GD in Proposition 2. Under balancing and Proposition 3, we have that at convergence,

$$
\mathbf{\Sigma}_{L:1} = \mathbf{\Sigma}_\star \implies \mathbf{\Sigma}_\ell = \begin{bmatrix} \mathbf{\Sigma}_{\star,r}^{1/L} & \mathbf{0} \\ \mathbf{0} & \alpha \cdot \mathbf{I}_{d-r} \end{bmatrix}, \quad \forall \ell \in [L-1], \quad \text{and} \quad \mathbf{\Sigma}_L = \mathbf{\Sigma}_\star^{1/L}.
$$

Thus, we have

$$
\mathbf{H}_{m,\ell} = \begin{cases} \mathbf{\Sigma}_\ell^{2(\ell-1)} \otimes \mathbf{\Sigma}_\star^{\frac{2(L-\ell)}{L}} & \text{for } m = \ell, \\ \mathbf{\Sigma}_\ell^{m+\ell-2} \otimes \mathbf{\Sigma}_\star^{2L-m-\ell} & \text{for } m \neq \ell. \end{cases}
$$

Now, we are left with computing the eigenvalues of $\mathbf{H} \in \mathbb{R}^{d^2 L \times d^2 L}$. To do this, let us block diagonalize $\mathbf{H}$ into $\mathbf{H} = \mathbf{P}\mathbf{C}\mathbf{P}^\top$, where $\mathbf{P}$ is a permutation matrix and

$$\mathbf{C} = \begin{bmatrix} \mathbf{C}_1 & & \\ & \ddots & \\ & & \mathbf{C}_{d^2} \end{bmatrix} \in \mathbb{R}^{d^2 L \times d^2 L},$$

where each $(i,j)$-th entry of $\mathbf{C}_k \in \mathbb{R}^{L \times L}$ is the $k$-th diagonal element of $\mathbf{H}_{i,j}$. Since $\mathbf{C}$ and $\mathbf{H}$ are similar matrices, they have the same eigenvalues. Then, since $\mathbf{C}$ is a block diagonal matrix, its eigenvalues (and hence the eigenvalues of $\mathbf{H}$) are the union of each of the eigenvalues of its blocks.

By observing the structure of $\mathbf{H}_{m,\ell}$, notice that each $\mathbf{C}_k$ is a rank-1 matrix. Hence, when considering the top-$r$ diagonal elements of $\mathbf{H}_{m,\ell}$ corresponding to each Kronecker product to construct $\mathbf{C}_k$, each $\mathbf{C}_k$ can be written as an outer product $\mathbf{u}\mathbf{u}^\top$, where $\mathbf{u} \in \mathbb{R}^L$ is

$$\mathbf{u}^\top = \begin{bmatrix} \sigma_{\star,i}^{1-\frac{1}{L}}\sigma_{\star,j}^{0} & \sigma_{\star,i}^{1-\frac{2}{L}}\sigma_{\star,j}^{\frac{1}{L}} & \sigma_{\star,i}^{1-\frac{3}{L}}\sigma_{\star,j}^{\frac{2}{L}} & \dots & \sigma_{\star,i}^{0}\sigma_{\star,j}^{1-\frac{1}{L}} \end{bmatrix}^\top. \tag{58}$$

Then, the non-zero eigenvalue of this rank-1 matrix is simply $\|\mathbf{u}\|_2^2$, which simplifies to

$$\|\mathbf{u}\|_2^2 = \sum_{\ell=0}^{L-1} \left( \sigma_{\star,i}^{1-\frac{1}{L}-\frac{1}{L}\ell} \cdot \sigma_{\star,j}^{\frac{1}{L}\ell} \right)^2.$$

Next, we can consider the remaining $d - r$ components of each Kronecker product of $\mathbf{H}_{m,\ell}$. Notice that for $m = \ell = L$, we have

$$\mathbf{H}_{L,L} = \begin{bmatrix} \sigma_{\star,1}^{\frac{2(L-1)}{L}} \cdot \mathbf{I}_d & & & \\ & \ddots & & \\ & & \sigma_{\star,r}^{\frac{2(L-1)}{L}} \cdot \mathbf{I}_d & \\ & & & \alpha^{2(L-1)}\mathbf{I}_{d-r} \otimes \mathbf{I}_d \end{bmatrix}.$$

This amounts to a matrix $\mathbf{C}_k$ with a single element $\sigma_{\star,i}^{\frac{2(L-1)}{L}}$ and 0 elsewhere. This gives an eigenvalue $\sigma_{\star,i}^{\frac{2(L-1)}{L}}$ for all $i \in [r]$, with multiplicity $d - r$.

Lastly, we can consider the diagonal components of $\mathbf{H}_{m,\ell}$ that is a function of the initialization scale $\alpha$. For this case, each $\mathbf{C}_k$ can be written as an outer product $\mathbf{v}\mathbf{v}^\top$, where

$$\mathbf{v}^\top = \begin{bmatrix} \sigma_{\star,i}^{1-\frac{1}{L}}\alpha^{0} & \sigma_{\star,i}^{1-\frac{2}{L}}\alpha & \sigma_{\star,i}^{1-\frac{3}{L}}\alpha^2 & \dots & \sigma_{\star,i}^{0}\alpha^{L-1} \end{bmatrix}^\top. \tag{59}$$

Similarly, the non-zero eigenvalue is simply $\|\mathbf{v}\|_2^2$, which corresponds to

$$\|\mathbf{v}\|_2^2 = \sum_{\ell=0}^{L-1} \left( \sigma_{\star,k}^{1-\frac{1}{L}-\frac{1}{L}\ell} \cdot \alpha^\ell \right)^2.$$

This completes the proof. $\qquad\square$

### C.3.3 PROOF OF THEOREM 1

*Proof.* To prove the result, we will consider the GD step on the $i$-th singular value and show that the 2-period orbit condition holds given the learning rate $\eta = \frac{2}{K}$. For ease of exposition, let us denote the $i$-th singular value of each $\mathbf{W}_\ell$ as $\sigma_i := \sigma_{\ell,i}$. Under balancing, consider the two-step GD update on the first singular value:

$$\sigma_i(t+1) = \sigma_i(t) + \eta L \cdot \left( \sigma_{\star,i} - \sigma_i^L(t) \right) \cdot \sigma_i^{L-1}(t)$$

$$\sigma_i(t) = \sigma_i(t+2) = \sigma_i(t+1) + \eta L \cdot \left( \sigma_{\star,i} - \sigma_i^L(t+1) \right) \cdot \sigma_i^{L-1}(t+1). \quad \text{(By 2-period orbit)}$$

Define $z := \left(1 + \eta L \cdot \left(\sigma_{\star,i} - \sigma_i^L(t)\right) \cdot \sigma_i^{L-2}(t)\right)$ and by plugging in $\sigma_i(t+1)$ for $\sigma_i(t)$, we have

$$\sigma_i(t) = \sigma_i(t)z + \eta L \cdot \left(\sigma_{\star,i} - \sigma_i^L(t)z^L\right) \cdot \sigma_i^{L-1}(t)z^{L-1}$$

$$\implies 1 = z + \eta L \cdot \left(\sigma_{\star,i} - \sigma_i^L(t)z^L\right) \cdot \sigma_i^{L-2}(t)z^{L-1}$$

$$\implies 1 = \left(1 + \eta L \cdot \left(\sigma_{\star,i} - \sigma_i^L(t)\right) \cdot \sigma_i^{L-2}(t)\right) + \eta L \cdot \left(\sigma_{\star,i} - \sigma_i^L(t)z^L\right) \cdot \sigma_i^{L-2}(t)z^{L-1}$$

$$\implies 0 = \left(\sigma_{\star,i} - \sigma_i^L(t)\right) + \left(\sigma_{\star,i} - \sigma_i^L(t)z^L\right) \cdot z^{L-1}$$

Simplifying this expression further, we have

$$0 = \sigma_{\star,i} - \sigma_i^L(t) + \sigma_{\star,i}z^{L-1} - \sigma_i^L(t)z^{2L-1}$$

$$\implies \sigma_i^L(t) + \sigma_i^L(t)z^{2L-1} = \sigma_{\star,i} + \sigma_{\star,i}z^{L-1}$$

$$\implies \sigma_i^L(t) \cdot \left(1 + z^{2L-1}\right) = \sigma_{\star,i} \cdot \left(1 + z^{L-1}\right)$$

$$\implies \sigma_i^L(t)\frac{\left(1 + z^{2L-1}\right)}{\left(1 + z^{L-1}\right)} = \sigma_{\star,i},$$

and by defining $\rho_i := \sigma_i(t)$, we obtain the polynomial

$$\sigma_{\star,i} = \rho_i^L \frac{1 + z^{2L-1}}{1 + z^{L-1}}, \quad \text{where } z := \left(1 + \eta L(\sigma_{\star,i} - \rho_i^L) \cdot \rho_i^{L-2}\right).$$

Next, we show the existence of (real) roots within the ranges for $\rho_{i,1}$ and $\rho_{i,2}$. We note that these roots only exist within the EOS regime. First, consider $\rho_{i,1} \in \left(0, \sigma_{\star,i}^{1/L}\right)$. We will show that for two values within this range, there is a sign change for all $L \geq 2$. More specifically, we show that there exists $\rho_i \in \left(0, \sigma_{\star,i}^{1/L}\right)$ such that

$$\rho_i^L \frac{1 + z^{2L-1}}{1 + z^{L-1}} - \sigma_{\star,i} > 0 \quad \text{and} \quad \rho_i^L \frac{1 + z^{2L-1}}{1 + z^{L-1}} - \sigma_{\star,i} < 0.$$

For the positive case, consider $\rho_i = \left(\frac{1}{2}\sigma_{\star,i}\right)^{1/L}$. We need to show that

$$\frac{1 + z^{2L-1}}{1 + z^{L-1}} = \frac{1 + \left(1 + \eta L \cdot \left(\frac{\sigma_{\star,i}}{2}\right) \frac{\sigma_{\star,i}^{1-\frac{2}{L}}}{2^{1-\frac{2}{L}}}\right)^{2L-1}}{1 + \left(1 + \eta L \cdot \left(\frac{\sigma_{\star,i}}{2}\right) \frac{\sigma_{\star,i}^{1-\frac{2}{L}}}{2^{1-\frac{2}{L}}}\right)^{L-1}} > 2.$$

To do this, we will plug in the smallest possible value of $\eta = \frac{2}{L\sigma_{\star,i}^{2-\frac{2}{L}}}$ to show that the fraction is still greater than 2, which gives us

$$u(L) := \frac{1 + \left(1 + \frac{1}{2^{1-\frac{2}{L}}}\right)^{2L-1}}{1 + \left(1 + \frac{1}{2^{1-\frac{2}{L}}}\right)^{L-1}}, \tag{60}$$

which is an increasing function of $L$ for all $L \geq 2$. Since $u(2) > 2$, Equation (60) must always be greater than 2. For the negative case, we can simply consider $\rho_i = 0$. Hence, since the polynomial is continuous, by the Intermediate Value Theorem (IVT), there must exist a root within the range $\rho_i \in \left(0, \sigma_{\star,i}^{1/L}\right)$.

Next, consider the range $\rho_{i,2} \in \left(\sigma_{\star,i}^{1/L}, (2\sigma_{\star,i})^{1/L}\right)$. Similarly, we will show sign changes for two values in $\rho_{i,2}$. For the positive case, consider $\rho_i = \left(\frac{3}{2}\sigma_{\star,i}\right)^{1/L}$. For $\eta$, we can plug in the smallest possible value within the range to show that this value of $\rho_i$ provides a positive quantity. Specifically, we need to show that

$$\frac{1 + z^{2L-1}}{1 + z^{L-1}} > \frac{2}{3} \implies \frac{1 + \left(1 + \frac{2}{\sigma_{\star,i}^{2-\frac{2}{L}}} \cdot (\sigma_{\star,i} - \frac{3}{2}\sigma_{\star,i}) \cdot \left(\frac{3}{2}\sigma_{\star,i}\right)^{1-\frac{2}{L}}\right)^{2L-1}}{1 + \left(1 + \frac{2}{\sigma_{\star,i}^{2-\frac{2}{L}}} \cdot (\sigma_{\star,i} - \frac{3}{2}\sigma_{\star,i}) \cdot \left(\frac{3}{2}\sigma_{\star,i}\right)^{1-\frac{2}{L}}\right)^{L-1}} > \frac{2}{3}.$$

We can simplify the fraction as follows:

$$\frac{1 + \left(1 + \frac{2}{\sigma_{\star,1}^{2-\frac{2}{L}}} \cdot (\sigma_{\star,1} - \frac{3}{2}\sigma_{\star,1}) \cdot \left(\frac{3}{2}\sigma_{\star,1}\right)^{1-\frac{2}{L}}\right)^{2L-1}}{1 + \left(1 + \frac{2}{\sigma_{\star,1}^{2-\frac{2}{L}}} \cdot (\sigma_{\star,1} - \frac{3}{2}\sigma_{\star,1}) \cdot \left(\frac{3}{2}\sigma_{\star,1}\right)^{1-\frac{2}{L}}\right)^{L-1}} = \frac{1 + \left(1 - (\frac{3}{2})^{1-\frac{2}{L}}\right)^{2L-1}}{1 + \left(1 - (\frac{3}{2})^{1-\frac{2}{L}}\right)^{L-1}}.$$

Then, since we are subtracting by $(\frac{3}{2})^{1-\frac{2}{L}}$, we can plug in its largest value for $L \geq 2$, which is $3/2$. This gives us

$$\frac{1 + (-0.5)^{2L-1}}{1 + (-0.5)^{L-1}} > \frac{2}{3},$$

as for odd values of $L$, the function increases to 1 starting from $L = 2$, and decreases to 1 for even $L$. To check negativity, let us define

$$h(\rho) := \frac{f(\rho)}{g(\rho)} := \frac{\rho^L \left(1 + z^{2L-1}\right)}{1 + z^{L-1}}.$$

We will show that $h'\left(\sigma_{\star,i}^{1/L}\right) < 0$:

$$h'\left(\sigma_{\star,i}^{1/L}\right) = \frac{f'\left(\sigma_{\star,i}^{1/L}\right) g\left(\sigma_{\star,i}^{1/L}\right) - f\left(\sigma_{\star,i}^{1/L}\right) g'\left(\sigma_{\star,i}^{1/L}\right)}{g^2\left(\sigma_{\star,i}^{1/L}\right)}$$

$$= \frac{f'\left(\sigma_{\star,i}^{1/L}\right) - \sigma_{\star,i} \cdot g'\left(\sigma_{\star,i}^{1/L}\right)}{2}$$

$$= \frac{L\sigma_{\star,i}^{1-\frac{1}{L}} - \sigma_{\star,i}(2L-1)\left(\eta L^2 \sigma_{\star,i}^{2-\frac{3}{L}}\right) - \sigma_{\star,i}(L-1)\left(\eta L^2 \sigma_{\star,i}^{2-\frac{3}{L}}\right)}{2}$$

$$= \frac{L\sigma_{\star,i}^{1-\frac{1}{L}} - (3L-2)\left(\eta L^2 \sigma_{\star,i}^{3-\frac{3}{L}}\right)}{2} < 0,$$

as otherwise we need $\eta \leq \frac{\sigma_{\star,i}^{2/L-2}}{3L^2-2L}$, which is out of the range of interest. Since $h'(\rho) < 0$, it follows that there exists a $\delta > 0$ such that $h(\rho) > h(x)$ for all $x$ such that $\rho < x < \rho + \delta$. Lastly, since $h(\rho) - \sigma_{\star,i} = 0$ for $\rho = \sigma_{\star,i}^{1/L}$, it follows that $h(\rho) - \sigma_{\star,i}$ must be negative at $\rho + \delta$. Similarly, by IVT, there must exist a root within the range $\rho_{i,2} \in \left(\sigma_{\star,i}^{1/L}, (2\sigma_{\star,i})^{1/L}\right)$. This proves that the $i$-th singular value undergoes a two-period orbit with the roots $\rho_{i,1}$ and $\rho_{i,2}$. Then, notice that if the learning rate is large enough to induce oscillations in the $i$-th singular value, then it is also large enough to have oscillations in all singular values from 1 to the $(i-1)$-th singular value (assuming that it is not large enough for divergence). Finally, at the (balanced) minimum, we can express the dynamics as

$$\mathbf{W}_{L:1} = \underbrace{\sum_{i=1}^{p} \rho_{i,j}^L \cdot \mathbf{u}_{\star,i}\mathbf{v}_{\star,i}^\top}_{\text{oscillation subspace}} + \underbrace{\sum_{k=p+1}^{d} \sigma_{\star,k} \cdot \mathbf{u}_{\star,k}\mathbf{v}_{\star,k}^\top}_{\text{stationary subspace}}, \quad j \in \{1, 2\}, \quad \forall \ell \in [L-1]. \qquad (61)$$

This completes the proof.

$\square$

### C.4 Auxiliary Results

**Lemma 8.** *Let $\{\mathbf{R}_\ell\}_{\ell=1}^L \in \mathbb{R}^{n \times n}$ be orthogonal matrices and $\mathbf{H}_{i,j} \in \mathbb{R}^{n^2 \times n^2}$ be diagonal matrices. Consider the two following block matrices:*

$$\mathbf{H} = \begin{bmatrix} \mathbf{H}_{1,1} & \mathbf{H}_{1,2} & \dots & \mathbf{H}_{L,1} \\ \mathbf{H}_{2,1} & \mathbf{H}_{2,2} & \dots & \mathbf{H}_{L,2} \\ \vdots & \vdots & \ddots & \vdots \\ \mathbf{H}_{1,L} & \mathbf{H}_{2,L} & \dots & \mathbf{H}_{L,L} \end{bmatrix}$$

$$\widetilde{\mathbf{H}} = \begin{bmatrix} \mathbf{R}_L \mathbf{H}_{1,1} \mathbf{R}_L^\top & \mathbf{R}_L \mathbf{H}_{1,2} \mathbf{R}_{L-1}^\top & \dots & \mathbf{R}_L \mathbf{H}_{1,L} \mathbf{R}_1^\top \\ \mathbf{R}_{L-1} \mathbf{H}_{2,1} \mathbf{R}_L^\top & \mathbf{R}_{L-1} \mathbf{H}_{2,2} \mathbf{R}_{L-1}^\top & \dots & \mathbf{R}_{L-1} \mathbf{H}_{2,L} \mathbf{R}_1^\top \\ \vdots & \vdots & \ddots & \vdots \\ \mathbf{R}_1 \mathbf{H}_{L,1} \mathbf{R}_L^\top & \mathbf{R}_1 \mathbf{H}_{L,2} \mathbf{R}_{L-1}^\top & \dots & \mathbf{R}_1 \mathbf{H}_{L,L} \mathbf{R}_1^\top \end{bmatrix}.$$

*Then, the two matrices $\mathbf{H}$ and $\widetilde{\mathbf{H}}$ are similar, in the sense that they have the same eigenvalues.*

*Proof.* It suffices to show that $\mathbf{H}$ and $\widetilde{\mathbf{H}}$ have the same characteristic polynomials. Let us define

$$\widetilde{\mathbf{H}} := \begin{bmatrix} \mathbf{A} & \mathbf{B} \\ \mathbf{C} & \mathbf{D} \end{bmatrix},$$

where

$$\mathbf{A} := \mathbf{R}_L \mathbf{H}_{1,1} \mathbf{R}_L^\top \qquad \mathbf{B} := \begin{bmatrix} \mathbf{R}_L \mathbf{H}_{1,2} \mathbf{R}_{L-1}^\top & \dots & \mathbf{R}_L \mathbf{H}_{1,L} \mathbf{R}_1^\top \end{bmatrix} \tag{62}$$

$$\mathbf{C} := \begin{bmatrix} \mathbf{R}_{L-1} \mathbf{H}_{2,1} \mathbf{R}_L^\top \\ \vdots \\ \mathbf{R}_1 \mathbf{H}_{L,1} \mathbf{R}_L^\top \end{bmatrix} \qquad \mathbf{D} := \begin{bmatrix} \mathbf{R}_{L-1} \mathbf{H}_{2,2} \mathbf{R}_{L-1}^\top & \dots & \mathbf{R}_{L-1} \mathbf{H}_{2,L} \mathbf{R}_1^\top \\ \vdots & \ddots & \vdots \\ \mathbf{R}_1 \mathbf{H}_{L,2} \mathbf{R}_{L-1}^\top & \dots & \mathbf{R}_1 \mathbf{H}_{L,L} \mathbf{R}_1^\top \end{bmatrix}. \tag{63}$$

Then, we have

$$\det(\widetilde{\mathbf{H}} - \lambda\mathbf{I}) = \det\left(\begin{bmatrix} \mathbf{A} - \lambda\mathbf{I} & \mathbf{B} \\ \mathbf{C} & \mathbf{D} - \lambda\mathbf{I} \end{bmatrix}\right)$$

$$= \det(\mathbf{A} - \lambda\mathbf{I}) \cdot \det((\mathbf{D} - \lambda\mathbf{I}) - \mathbf{C}(\mathbf{A} - \lambda\mathbf{I})^{-1}\mathbf{B}),$$

where the second equality is by the Schur complement. Notice that

$$(\mathbf{A} - \lambda\mathbf{I})^{-1} = (\mathbf{R}_L \mathbf{H}_{1,1} \mathbf{R}_L^\top - \lambda\mathbf{I})^{-1} = (\mathbf{R}_L \mathbf{H}_{1,1} \mathbf{R}_L^\top - \lambda\mathbf{R_L}\mathbf{R_L}^\top)^{-1}$$

$$= \mathbf{R}_L \cdot (\mathbf{H}_{1,1} - \lambda\mathbf{I})^{-1} \cdot \mathbf{R}_L^\top.$$

Then, we also see that,

$$\mathbf{C}(\mathbf{A} - \lambda\mathbf{I})^{-1}\mathbf{B} = \underbrace{\begin{bmatrix} \mathbf{R}_{L-1} & & \\ & \ddots & \\ & & \mathbf{R}_1 \end{bmatrix}}_{=:\widehat{\mathbf{V}}} \cdot \mathbf{E} \cdot \underbrace{\begin{bmatrix} \mathbf{R}_{L-1}^\top & & \\ & \ddots & \\ & & \mathbf{R}_1^\top \end{bmatrix}}_{=:\widehat{\mathbf{V}}^\top}.$$

where

$$\mathbf{E} := \begin{bmatrix} \mathbf{H}_{2,1} \cdot (\mathbf{H}_{1,1} - \lambda\mathbf{I})^{-1} \cdot \mathbf{H}_{1,2} & \dots & \mathbf{H}_{2,1} \cdot (\mathbf{H}_{1,1} - \lambda\mathbf{I})^{-1} \cdot \mathbf{H}_{1,L} \\ \vdots & \ddots & \vdots \\ \mathbf{H}_{L,1} \cdot (\mathbf{H}_{1,1} - \lambda\mathbf{I})^{-1} \cdot \mathbf{H}_{1,2} & \dots & \mathbf{H}_{L,1} \cdot (\mathbf{H}_{1,1} - \lambda\mathbf{I})^{-1} \cdot \mathbf{H}_{1,L} \end{bmatrix}.$$

Similarly, we can write $\mathbf{D}$ as

$$\mathbf{D} = \widehat{\mathbf{V}} \underbrace{\begin{bmatrix} \mathbf{H}_{2,2} & \dots & \mathbf{H}_{2,L} \\ \vdots & \ddots & \vdots \\ \mathbf{H}_{L,2} & \dots & \mathbf{H}_{L,L} \end{bmatrix}}_{=:\mathbf{F}} \widehat{\mathbf{V}}^\top.$$

Then, we have

$$\det(\widetilde{\mathbf{H}} - \lambda\mathbf{I}) = \det(\mathbf{R}_L \cdot (\mathbf{H}_{1,1} - \lambda\mathbf{I}) \cdot \mathbf{R}_L^\top) \cdot \det\left(\widehat{\mathbf{V}} \cdot (\mathbf{E} - \mathbf{F}) \cdot \widehat{\mathbf{V}}^\top\right)$$
$$= \det(\mathbf{H}_{1,1} - \lambda\mathbf{I}) \cdot \det(\mathbf{E} - \mathbf{F}),$$

which is not a function of $\mathbf{U}, \mathbf{V}, \{\mathbf{R}_\ell\}_{\ell=1}^L$. By doing the same for $\mathbf{H}$, we can show that both $\widetilde{\mathbf{H}}$ and $\mathbf{H}$ have the same characteristic polynomials, and hence the same eigenvalues. This completes the proof.

$\square$

**Lemma 9.** *Let $\mathbf{A}, \mathbf{B} \in \mathbb{R}^{d \times d}$ be two orthogonal matrices. Then, the Kronecker product of $\mathbf{A}$ and $\mathbf{B}$ is also an orthogonal matrix:*

$$(\mathbf{A} \otimes \mathbf{B})^\top (\mathbf{A} \otimes \mathbf{B}) = (\mathbf{A} \otimes \mathbf{B})(\mathbf{A} \otimes \mathbf{B})^\top = \mathbf{I}_{d^2}.$$

*Proof.* We prove this directly by using properties of Kronecker products:

$$(\mathbf{A} \otimes \mathbf{B})^\top (\mathbf{A} \otimes \mathbf{B}) = \mathbf{A}^\top \mathbf{A} \otimes \mathbf{B}^\top \mathbf{B}$$
$$= \mathbf{I}_d \otimes \mathbf{I}_d = \mathbf{I}_{d^2}.$$

Similarly, we have

$$(\mathbf{A} \otimes \mathbf{B})(\mathbf{A} \otimes \mathbf{B})^\top = \mathbf{A}\mathbf{A}^\top \otimes \mathbf{B}\mathbf{B}^\top$$
$$= \mathbf{I}_d \otimes \mathbf{I}_d = \mathbf{I}_{d^2}.$$

This completes the proof.

$\square$

**Lemma 10.** *Let $\{a(t)\}_{t=1}^N$ be a sequence such that $a(t) \geq 0$ for all $t$. If there exists a constant $c \in (0, 1)$ such that $a(t+1) < c \cdot a(t)$ for all $t$, then $\lim_{t \to \infty} a(t) = 0$.*

*Proof.* We prove this by direct reasoning. From the assumption $a(t+1) < c \cdot a(t)$ for some $c \in (0, 1)$, we can iteratively expand this inequality:

$$a(t+1) < c \cdot a(t), \quad a(t+2) < c \cdot a(t+1) < c^2 \cdot a(t),$$

and, more generally, by induction:

$$a(t+k) < c^k \cdot a(t), \quad \text{for all } k \geq 0.$$

Since $c \in (0, 1)$, the sequence $\{c^k\}_{k=0}^\infty$ converges to 0 as $k \to \infty$. Hence:

$$0 \leq \lim_{k \to \infty} a(t+k) \leq \lim_{k \to \infty} c^k \cdot a(t) = 0.$$

Therefore, by the squeeze theorem, the sequence $\{a(t)\}$ converges to 0 as $t \to \infty$.

$\square$

**Lemma 11** (Chen & Bruna (2023)). *Consider any 1-D differentiable function $f(x)$ around a local minima $\bar{x}$, satisfying (i) $f^{(3)}(\bar{x}) \neq 0$, and (ii) $3[f^{(3)}]^2 - f'' f^{(4)} > 0$ at $\bar{x}$. Then, there exists $\epsilon$ with sufficiently small $|\epsilon|$ and $\epsilon \cdot f^{(3)} > 0$ such that: for any point $x_0$ between $\bar{x}$ and $\bar{x} - \epsilon$, there exists a learning rate $\eta$ such that $F_\eta^2(x_0) = x_0$, and*

$$\frac{2}{f''(\bar{x})} < \eta < \frac{2}{f''(\bar{x}) - \epsilon \cdot f^{(3)}(\bar{x})}.$$

