# OpenReview forum: "Learning Dynamics of Deep Matrix Factorization Beyond the Edge of Stability"
_ICLR.cc/2025/Conference — ICLR 2025 Poster_

### Official Review · Reviewer_nnRE · 2024-11-01

**Soundness:** 3
**Presentation:** 2
**Contribution:** 3
**Rating:** 6
**Confidence:** 4

**Summary:**

This paper analyzes the catapult behaviors of DLNs (Deep Linear Networks) and explains many unexplained observations regarding EOS (Edge of Stability) phenomenon: (i) periodic subspace oscillations (especially in the top subspace that corresponds with more prominent features), (ii) mild (or no) sharpening for simple dataset with low complexity (when the eigenvalue for the strongest feature is too small), (iii) difference between DLNs and DiagLNs (i.e., the architecture matters), and (iii) catapult in LoRA.

**Strengths:**

- The paper explains many unexplained observations regarding EOS phenomenon.

**Weaknesses:**

- Lemma 2 does **not** tell us that the singular values become balanced as $t\rightarrow \infty$ (L191-193). The strictly decreasing balancing gap and the difference being lower bounded by zero does **not** lead to the convergence to zero (L307-309). Think about the sequence $0.5+\frac{1}{t}$. Can you provide a proof to show the convergence to zero?

- "When the learning rate is large enough to induce catapults, we observe that the training loss decreases rapidly ..."
The statement does not seem right. Large lr may lead to fast decreasing of the training loss because of its large stepsize not because of the catapults. Same for the statement "when the learning rate is small, convergence takes much longer, as the model seems to bounce around within the same local basin before reaching a low training loss". Can you provide an experiment to show that "the catapult implies faster optimization"?

- It is hard to say "in DLNs, self-stabilization does not occur". Actually, also in the original EOS paper, the sharpness oscillates **above** $2/\eta$ (they say "it hovers above $2/\eta$"). It is true that DLNs do not satisfy the condition of the proposition in the self-stabilization paper, but self-stabilization is a broader concept and the threshold may not be necessarily $2/\eta$. The statement should be written carefully.

**Questions:**

The following questions may be related to the weaknesses as the paper may have some unclear points:
- In Thm 1, what is $\rho_i(t)$ exactly? It only says that $\rho_i(t)\in \\{\rho_1,\rho_2\\}$. Does $i$ depend on $t$? As it says that the matrix oscillates, it is likely that $\rho_i(t)$ changes with $t$. Is it $\rho_i(t)=\rho_1$ if $t$ is odd (even) and $\rho_i(t)=\rho_2$ if $t$ is even (odd)? If the readers have to guess the meaning, then it is not well-written.

- What do you mean by that "To show oscillations in two or more subspace, we can easily extend Theorem 1"? Why do we need to set $\lambda_2\geq K>\lambda_3$? I don't think this is a trivial result from Thm 1. Can you elaborate more on the paragraph L241-248? My understanding is that $\rho_i(t)$ in the first term of $W_\ell(t)$ in Thm 1 corresponds to the oscillation, is it right? I didn't fully understand the $r$-subspace oscillation part.

- For the second term $\sum_j \sigma_{\ast,i}u_{\ast,j}v^\top_{\ast,j}$, isn't it $\sigma_{\ast,j}$ with $j$, not $\sigma_{\ast,i}$ with $i$?

- Is there any reason that we can view the adaptations as individual low-rank matrix factorization problems? Can you elaborate more on this? I hope some "math" may help the reader to understand this.

- "Notably, for ranks $r=4$ and $r=8$, there are catapults ... do not occur for $r=1$ or $r=2$". What catapults are you talking about? In Fig 7, it is hard to see the catapult (or loss oscillation) and compare it with other ranks ($r=1,2$). I don't fully understand Section 4.2.

- Can you somehow compute or estimate the $\sigma_{\ast,1}$ for Fig 10 (a) and for each image in Fig 10 (b)? It would be better to have a quantitative understanding of the "low-complexity learning".

- L 29-30 (spacing): ... 2020).The -> ... 2020). The
- L415: give a ... -> given a ...
-

---

> ### Author Response · Authors · 2024-11-19
> **Author Response (Part 1)**
>
> **Comment:** "Lemma 2 does not tell us that the singular values become balanced"
>
> **Response:** Thank you for highlighting this important aspect. Indeed, even if a sequence $\\{ a(t) \\}\_t $ is lower bounded by $0$ and obeys $a(t+1)<a(t)$, the sequence is not guaranteed to converge to $0$. However, if there exists a $0<c<1$ such that $a(t+1)<c\cdot a(t)$, then the sequence is guaranteed to converge to $0$ as $t \rightarrow \infty$ (see [Lemma 9](https://imgur.com/x51lqO4)). To this end, we made an attempt to modify Lemma 2 to show strict balancing such that $b(t+1) < c\cdot b(t)$. To do this, we analyze 2 cases: (i) $\\pi(t) < \\sigma_*$, and (ii) $\\pi(t)> \\sigma_*$, and showed that there exists a constant $0<c<1$ such that it holds, where $\\pi(t)$ is the product of singular values at time $t$ and $\\sigma_*$ being the target singular value (comparable to the machine precision scale).
> However, when $\\pi(t)> \\sigma_*$, we can only prove for $c=1$, i.e, $b(t+1)< b(t)$. Hence, the balancing gap cannot be proven to go to exactly zero but our experiments suggest that is converges to something very small.
>
> For the final manuscript, we can make these points even more rigorous by performing an error analysis -- we will prove our results with an error term $e(\alpha)$, where $\alpha$ is the initialization scale, which arises when balancing *only approximately* holds. Then, we can re-state our results including the eigenvalue calculation (requires matrix perturbation theory) in the order of $e(\alpha)$. However, due the time limitations, we leave it for the final manuscript but will update the manuscript if completed during the discussion period. Finally, we note that we also added in the balanced initialization in Equation 3, and hence the conclusions made in our paper do not change.
>
> **Comment:** “Large lr may lead to fast decreasing of the training loss because of its large stepsize not because of the catapults.”; “notable for ranks r=4 and r=8, there are catapults…”
>
> **Response:** We agree with this point – our experiments at the current stage do not clearly show that the increase in Pearson correlation is because of simply using a large learning rate or if it is because of catapults. We made this conclusion because if you zoom in to the very early iterations in [Figure 21](https://imgur.com/ThyRT8T) of the revised manuscript, there exists large “jumps” in the training loss for ranks r=4, 8, whereas there are much smaller jumps in ranks r=1, 2. We jumped to the conclusion that this hinted that potential catapults lead to faster optimization. Since we cannot make a stronger claim other than these experiments, we moved this section to the Appendix as suggested by Reviewer n4c2.
>
> **Comment:** Self-stabilization in DLNs
>
> **Response:** We have modified the statement in our revised manuscript to better reflect this point. To summarize, recall that in DLNs, sharpness oscillates periodically about the limit $2/ \\eta$ and this comes from the property that in each eigenvector direction, it satisfies the stable oscillation condition. This is in contrast to the self-stabilization effect explored in Damian et al. where sharpness decreases below a specified threshold and is a result of a closed loop sharpness reduction feedback. The difference between sustained oscillations and damped oscillations/catapults are classically studied in control system by locating the poles of transfer function of a closed loop system. We believe a similar study can be made to study these two cases but it is currently beyond the scope of our work.
>
> **Comment:** "what is $\\rho_i(t)$ exactly?"
>
> **Response:** We have changed the notation in the revised manuscript to remove the dependence on $t$. $\\rho_i$ are the two values that the first singular value takes when it exhibits two-period oscillation.
>
> **Comment:** "to show oscillations in two or more subspace... I don’t think this is a trivial result from Thm 1."
>
> **Response:** We apologize for the confusion -- this is a direct result from Theorem 2 and not Theorem 1. Theorem 2 states that whenever $\\eta$ is set such that it exceeds the stability condition $\eta>2/\\lambda_i$ in each eigenvector direction $\Delta_{i}$, then it stably oscillates about the eigenvector direction $\Delta_{i}$. Hence, by ensuring that $\eta$ lies in $ \lambda_{3}<\frac{2}{\eta}< \lambda_{2} < \lambda_{1} $, oscillations only occur along $\Delta_{1}$ and $\Delta_{2}$ and not along $\Delta_{3}$. We have revised our theorem statement to better reflect this as well. We also included a figure in 2-dimension to visualize the stable subspace oscillation in [Figure 10](https://imgur.com/3UZNmjP).

---

> > ### Comment · Reviewer_nnRE · 2024-11-26
> >
> > Thank you for the answer. I'm happy that the authors updated the Lemma 2.

---

> ### Author Response · Authors · 2024-11-19
> **Author Response (Part 2)**
>
> **Comment:** "...view the adaptations as individual low-rank matrix factorization problems?"
>
> **Response:** For each weight matrix, suppose there existed a “target” weight matrix $\\mathbf{M}$ that serves as the optimal weight matrix for the fine-tuned model. Then, one way of finding the low-rank adaptors to fit $\\mathbf{M}$ can be viewed as a low-rank matrix factorization problem: $$\underset{\\mathbf{A},\ \mathbf{B}}{\\mathrm{min}} \, \\| \\mathbf{W}\_0 + \\mathbf{AB}^\\top -  \\mathbf{M}\\|^2_{\\mathsf{F}},$$ where $\mathbf{W}_0$ is the frozen weight matrix. This is what we envisioned when mentioning viewing the adaptations as individual problems.
>
> **Comment:** "It would be better to have a quantitative understanding of the 'low-complexity learning'"
>
> **Response:**
> Fitting low frequency images with convolutional neural networks can be considered a low complexity task because of spectral bias [1] of such networks. Smooth/low frequency images are much easier to fit for convolutional neural networks due to spectral bias where the networks incrementally learns each Fourier frequency (low to high). Hence, a measure of complexity of each image can be the Fourier bandwidth of the image. Furthermore, while fitting low frequency images or simple datasets like MNIST, intrinsic update dimension in the parameter space of networks tend to be much smaller [2, 3]. However, the exact relation between the intrinsic dimension of parameter update and its sharpness is unknown and calls for further in-depth study. We revised the manuscript stating that we highlight the difference between the low complexity learning in CNNs vs linear networks as in our setting.
>
>
> Thank you for pointing out the typos as well -- they are fixed in the revised manuscript.
>
> References:
>
> [1] Nasim Rahaman and Aristide Baratin and Devansh Arpit and Felix Draxler and Min Lin and Fred A. Hamprecht and Yoshua Bengio and Aaron Courville. "On the Spectral Bias of Neural Networks." https://arxiv.org/abs/1806.08734
>
> [2] Chunyuan Li and Heerad Farkhoor and Rosanne Liu and Jason Yosinski. "Measuring the Intrinsic Dimension of Objective Landscapes". https://arxiv.org/abs/1804.08838
>
> [3] Phillip Pope and Chen Zhu and Ahmed Abdelkader and Micah Goldblum and Tom Goldstein. "The Intrinsic Dimension of Images and Its Impact on Learning". https://arxiv.org/abs/2104.08894

---

### Official Review · Reviewer_n4c2 · 2024-11-03

**Soundness:** 3
**Presentation:** 3
**Contribution:** 3
**Rating:** 8
**Confidence:** 3

**Summary:**

This paper rigorously analyzes gradient descent dynamics of deep linear networks for deep matrix factorization at the Edge of Stability. Assuming initialization within the singular vector stationary set and strictly balanced weights, the authors characterize a 2-period oscillation within top subspaces. The main insight is that, under these assumptions, the loss can be rewritten as a function of singular values, decoupling the effect of singular vectors. The authors also provide empirical support for the strict balancing assumption, showing that singular values become increasingly balanced during gradient descent dynamics, with numerical experiments aligning well with the theoretical findings.

**Strengths:**

This paper makes a solid contribution to the theoretical understanding of the Edge of Stability by analyzing gradient descent dynamics in deep matrix factorization. It extends previous studies on deep scalar linear networks [Kreisler et al., 2023], offering a rigorous proof that oscillations in the EoS regime are confined to top subspaces, as empirically observed in prior work [Zhu et al., 2024]. Additionally, the exact characterization of Hessian eigenvalues at convergence under strict balancing provides a valuable technical contribution to deep matrix factorization problem.

---

**References**

[Kreisler et al., 2023] Gradient Descent Monotonically Decreases the Sharpness of Gradient Flow Solutions in Scalar Networks and Beyond, ICML 2023.

[Zhu et al., 2024] Catapults in SGD: spikes in the training loss and their impact on generalization through feature learning, ICML 2024.

**Weaknesses:**

1. **My main concern is that Lemma 2 does not ensure convergence to strict balancing.** While Lemma 2 shows that the balancing gap decreases with each step, it does not guarantee that this gap converges to zero. The authors suggest that Lemma 2 justifies the strictly balanced assumption, but this needs further clarification. Although Figure 4 empirically shows the gap converging to zero, Lemma 2 alone does not rigorously establish this phenomenon.

2. **The model fails to capture (non-monotonic) decreasing training loss in the Edge of Stability.** A key characteristic of the Edge of Stability is that training loss oscillates but decreases over time, reflecting unstable convergence (e.g., [Cohen et al., 2021], [Ahn et al., 2022]). In this study’s setup, weights oscillate in a 2-period orbit, causing the loss to oscillate indefinitely without decreasing. This limits the model’s applicability to realistic settings.

3. **The claims on LoRA dynamics in Section 4.2 lack sufficient evidence.** The authors claim that using large learning rates in LoRA dynamics leads to oscillations and catapults, enhancing generalization by increasing Pearson correlation. However, Figures 7 and 8 do not clearly show catapult behavior, as oscillations occur even at lower learning rates (e.g., $\eta = 10^{-5}, 10^{-6}$), making it difficult to conclude that catapults are unique to $\eta=10^{-4}$. Additionally, empirical evidence linking catapults to better generalization is insufficient. The claims here seem overstated and less relavant to the main theory. I recommend moving this section to the appendix and avoid making strong claims.


Minor comments
- (Section 3, Page 4, Line 169) Typo: unbold subscript $k$ in $W_k$.
- (Appendix A, Page 14, Line 723) Wrong citation: "Ahn et al. (2022) established the phenomenon in two-layer networks..." is citing the paper [Ahn et al., 2022], but it should be citing another paper [Ahn et al., 2023].
- (Reference, Page 13) Duplicate entries for Zhu et al. (2024) in the reference (Line 692 and Line 696).

---

**References**

[Cohen et al., 2021] Gradient Descent on Neural Networks Typically Occurs at the Edge of Stability, ICLR 2021.

[Ahn et al., 2022] Understanding the Unstable Convergence of Gradient Descent, ICML 2022.

[Ahn et al., 2023] Learning threshold neurons via the “edge of stability”, NeurIPS 2023.

**Questions:**

1. Could the authors please include the following references in the related works section? Prior to [Cohen et al., 2021], [Jastrzebski et al., 2019] and [Jastrzebski et al., 2020] demonstrated that step size influences sharpness along optimization trajectories. Additionally, [Ahn et al., 2023], [Song et al., 2023], and [Karla et al., 2023] provide rigorous analyses of learning dynamics at the Edge of Stability in simplified settings, such as two-layer linear networks. [Zhu et al., 2024] and [Chen et al., 2024] study gradient descent dynamics for quadratic models in large learning rate regimes where catapults occur.

2. When initialization is outside the singular vector invariant set, how do loss and sharpness behave at the Edge of Stability? Specifically, does the weight converge to a period-2 orbit, or does the loss oscillate while decreasing non-monotonically? Similarly, if oscillations begin before strict balance is achieved, how are the loss trajectory and learning dynamics affected? It would be insightful if the authors could provide additional experimental results exploring these scenarios.

---

**References**

[Jastrzebski et al., 2019] On the Relation Between the Sharpest Directions of DNN Loss and the SGD Step Length, ICLR 2019.

[Jastrzebski et al., 2020] The Break-Even Point on Optimization Trajectories of Deep Neural Networks, ICLR 2020.

[Cohen et al., 2021] Gradient Descent on Neural Networks Typically Occurs at the Edge of Stability, ICLR 2021.

[Ahn et al., 2023] Learning threshold neurons via the “edge of stability”, NeurIPS 2023.

[Song et al., 2023] Trajectory Alignment: Understanding the Edge of Stability Phenomenon via Bifurcation Theory, NeurIPS 2023.

[Karla et al., 2023] Universal Sharpness Dynamics in Neural Network Training: Fixed Point Analysis, Edge of Stability, and Route to Chaos, arXiv 2023.

[Zhu et al., 2024] Quadratic models for understanding neural network dynamics, ICLR 2024.

[Chen et al., 2024] From Stability to Chaos: Analyzing Gradient Descent Dynamics in Quadratic Regression, TMLR 2024.

---

> ### Author Response · Authors · 2024-11-19
> **Author Response**
>
> Thank you for your review. Overall, we fixed typos and added in the references as suggested. Below, we provide a few concrete responses regarding weaknesses and concerns.
>
> **Comment:** "main concern is that Lemma 2 does not ensure convergence to strict balancing"
>
> **Response:** Thank you for highlighting this important aspect. Indeed, even if a sequence $\\{ a(t) \\}\_t $ is lower bounded by $0$ and obeys $a(t+1)<a(t)$, the sequence is not guaranteed to converge to $0$. However, if there exists a $0<c<1$ such that $a(t+1)<c\cdot a(t)$, then the sequence is guaranteed to converge to $0$ as $t \rightarrow \infty$ (see [Lemma 9](https://imgur.com/x51lqO4)). To this end, we made an attempt to modify Lemma 2 to show strict balancing such that $b(t+1) < c\cdot b(t)$. To do this, we analyze 2 cases: (i) $\\pi(t) < \\sigma_*$, and (ii) $\\pi(t)> \\sigma_*$, and showed that there exists a constant $0<c<1$ such that it holds, where $\\pi(t)$ is the product of singular values at time $t$ and $\\sigma_*$ being the target singular value.
> However, when $\\pi(t)> \\sigma_*$, we can only prove for $c=1$, i.e, $b(t+1)< b(t)$. Hence, the balancing gap cannot be proven to go to exactly zero but our experiments suggest that is converges to something very small (comparable to the machine precision scale).
>
> For the final manuscript, we can make these points even more rigorous by performing an error analysis -- we aim to prove our results with an error term $e(\alpha)$, where $\alpha$ is the initialization scale, which arises when balancing *only approximately* holds. Then, we can re-state our results including the eigenvalue calculation with an error order of $e(\alpha)$. However, due the time limitations, we leave it for the final manuscript but will update the manuscript if completed during the discussion period. Finally, we note that we also added in the balanced initialization in Equation 3, and hence the conclusions made in our paper do not change.
>
> **Comment:** "the model fails to capture (non-monotonic) decreasing training loss"
>
> **Response:** We point out that the linear model also decreases the loss non-monotonically where oscillations occur while incremental rank learning takes place (e.g., see [Figure 2](https://imgur.com/dMegxPq)). Here, oscillations occur after the first and second saddle jumps take place. However, when the iterates reach the global minimum, the model cannot decrease the training loss further since there are no spurious local minimas and sustained oscillations take place near the global minimum. We discussed this point of difference and included a loss landscape figure comparing linear scalar deep networks with highly non-convex landscapes with local minimas like the Holder table (see Figure [8](https://imgur.com/CsGOdap), [10](https://imgur.com/3UZNmjP), [11](https://imgur.com/7xlMhUA)). Furthermore, DLNs exhibit oscillatory behaviors beyond the stability regime instead of divergence, which Theorem 1 shows. Since, the optimization community has mostly focused on optimization in the stable regime, we believe our work lays the foundation to study deep models that operate beyond stability, which neural networks typically operate on.
>
> **Comment:** "The claims on LoRA dynamics in Section 4.2 lack sufficient evidence."
>
> **Response:** As stated in the global response, we have moved this section to the Appendix, and changed the language such that it warrants further investigation to avoid making strong claims. We hypothesized that the oscillations at lower learning rates occur due to the stochasticity in the updates by using minibatches, and that there exists actual catapults due to the large learning rate. This is because in the very early stages of training, the catapults for larger learning rates happen at a much larger magnitude (see new caption in [Figure 21](https://imgur.com/ThyRT8T)) However, since we cannot precisely delineate these points, we decided to remove it as a contribution and remove it from the main text.

---

> ### Author Response · Authors · 2024-11-19
> **Author Response (Part 2)**
>
> **Comment:** "When initialization is outside the singular vector invariant set, how do loss and sharpness behave..."
>
> **Response:**
> For concreteness, we added Section B.4 in the Appendix to answer this question thoroughly. To summarize, even when initialization is outside the SVS set, the loss still oscillates within a 2-period orbit and the weights indeed converge to period-2 orbit (within the correct learning rate regime). Near the global minimum, the loss cannot decrease non-monotonically since the oscillations are contained in a *single loss basin* and there are no spurious local minima. This implies that our theory still holds, but the evolution of the singular vectors would need to be accounted for if they do not converge to the SVS set.
> Though, we also consider two more initializations outside the SVS set in Section B.4: orthogonal and random initialization (see [Figure 15](https://imgur.com/xON7b4q)). These initialization do not belong to the SVS set but even at EOS, they converge to the SVS set.
>
> As for the balancing, we consider an initialization larger than the one required in Theorem 1 in Section B.4, where strict balancing fails to hold. Here, although oscillations start before strict balancing holds, the loss and singular values of weights exhibits 2-period oscillations. We show this in [Figure 14](https://imgur.com/7RSHpvN).

---

> ### Comment · Reviewer_n4c2 · 2024-11-22
>
> I appreciate the authors for their efforts in revising the paper and conducting additional experiments. The revised version has significantly improved the clarity of the contributions, and I find that most of my initial concerns have been adequately addressed. Thank you for the detailed and thoughtful rebuttal.
>
> That said, I still have a remaining concern regarding **Lemma 2**. While the revised lemma provides a clearer statement ensuring the balancing gap converges to zero under the condition $\pi(t) < \sigma_*$ for all time steps $t$, I find this condition potentially restrictive.
>
> - **Initialization condition:** Could the authors clarify whether there exists a practical initialization condition—particularly when strict balancing does not hold initially—that ensures $\pi(t) < \sigma_*$ for all $t$? It would be helpful to understand if this condition can emerge naturally under typical initialization schemes or during the training dynamics.
>
> - **Case when $\pi(t) > \sigma_*$:** Additionally, I am curious about the implications when $\pi(t) > \sigma_*$ at some time step $t$. Specifically, is the result $b(t+1) < c \cdot b(t)$ with $0 < c < 1$ fundamentally unattainable in such scenarios, thereby necessarily leading to $c = 1$?
>
> Addressing these questions would further strengthen the theoretical contribution of the paper and provide greater clarity on the assumptions underlying the convergence result. I am inclined to increase my score if the authors can address these points.

---

> ### Author Response · Authors · 2024-11-23
> **Author Response (Part 3)**
>
> Thank you for your timely response. Firstly, we would like to clarify when the two cases $\pi(t) < \\sigma\_\star$ and $\\pi(t) > \sigma_\star$ actually occur.
>
> Consider the case in which the learning rate is stable (i.e., oscillations do not occur) and the global minima $\pi(t) = \\sigma\_\star$ is stably achieved. In this case, $\pi(0) = 0$ (due to zero-$\alpha$ initialization) and grows until it hits $\sigma_\star$ stably. Mathematically, this implies that the iterates $\pi(t)$ always stay in the regime of $\pi(t) < \\sigma\_\star$, until it reaches $\pi(t) = \\sigma\_\star$.
>
> Now, let us consider the EOS regime. If the learning rate is unstable, it starts from $\pi(0) =0$ (due to zero-$\alpha$ initialization), where $\pi(t) < \\sigma\_\star$ (Case 1), but overshoots the global minima $\pi(t) = \\sigma\_\star$ to enter Case 2 ($\pi(t) > \\sigma\_\star$). Then due to EOS, it oscillates between the regions Case 1 ($\pi(t) < \\sigma\_\star$) and Case 2 ($\pi(t) > \\sigma\_\star$). In this attached [figure](https://imgur.com/gfYMpq3), we delineate these two regions for clarity. Notice that this is directly implied by Theorem 1, as if you look at $\rho_1$ and $\rho_2$ carefully, the oscillations imply that we alternate between the case $\pi(t) < \\sigma\_\star$ and $\\pi(t) > \sigma_\star$.
>
>  With this in place, we answer your questions, highlighting how our proof explains these two phases:
>
> **Initialization condition (Case 1 $\pi(t) < \\sigma\_\star$):** Consider the initialization in the manuscript, where $\\mathbf{W}\_L = \\mathbf{0}$ and the rest are initialized to $\\mathbf{W}\_l = \alpha \mathbf{I}$. Notice that strict balancing does not hold initially here. In the stable regime, as long as $\alpha$ is sufficiently small, *we will have $\pi(t) < \sigma_\star$ until it hits the global minimum $\pi(t) = \sigma_\star$*.
>
> For the case we analyze, EOS ensures oscillation about the global minima $\pi(t) = \\sigma\_\star$. So for *all* initializations, assuming that $\alpha$ satisfies the conditions in our paper, we will have $\pi(t) < \sigma_\star$ until it starts to alternate between the conditions in Case 1 and Case 2. In the attached [Figure](https://imgur.com/qkzzzTH), notice that the balancing gap does indeed decrease when $\pi(t) < \sigma_\star$, which is when there exists a $0< c < 1$ for which $(1-g(t))<c$ (since $g(t)<1$ in figure). Here $b(t+1)=b(t)(1-g(t))$, $g(t)$ being the quantity we defined in line-1750 in manuscript.
>
> **Oscillation between Case 1 ($\pi(t) < \\sigma\_\star$) and Case 2 ($\pi(t) > \\sigma\_\star$):** Once it overshoots ($t>500$ in figure) $\pi(t) = \\sigma\_\star$, it hits Case 2 ($\pi(t) > \\sigma\_\star$) and oscillates between Case 1 and Case 2 (see [Figure](https://imgur.com/qkzzzTH)). For Case 2 strictly, we could only prove that $(1-g(t))<c$ for $c=1$, but note that in [Figure](https://imgur.com/qkzzzTH), $g(t)<1$ ensuring there exists a $0< c < 1$ even when there is oscillation between these two cases. Although our proof in Case 2 can only ensure $c=1$, the iterates mostly undergoes through Case 1 and then alternates Case 1 and Case 2. The strict balancing effect has undergone in Case 1 (in both the time regions) and is sufficient to reduce the balancing gap to a very small value. This is reflected perfectly in the [figure](https://imgur.com/qkzzzTH), where we see balancing gap decreases monotonically and $g(t)<1$ for all $t$.
>
> We will add a discussion of this in appendix.

---

> > ### Comment · Reviewer_n4c2 · 2024-11-25
> >
> > Thank you for your detailed and thoughtful response. I appreciate the effort you put into addressing my concerns. After reviewing your rebuttal, I am satisfied with the clarifications provided and have increased my score to 8. Please make sure to incorporate the points you mentioned into the final manuscript.

---

### Official Review · Reviewer_GbsE · 2024-11-04

**Soundness:** 2
**Presentation:** 2
**Contribution:** 2
**Rating:** 6
**Confidence:** 3

**Summary:**

The paper analyzes learning dynamics of deep linear networks a.k.a. deep matrix factorization using Gradient Descent (GD) in the Edge-of-Stability (EOS) regime. The paper proves that, under a "strict balanced state" condition of singular values exactly matching across the weight matrices, the top singular values oscillate, leading to oscillations in training loss in the EOS regime. The paper also presents an experiment showing loss oscillations in Low-Rank-Adaptation (LoRA) finetuning on a language modeling task. Results in the paper help explain some observations in existing work.

**Strengths:**

The EOS oscillation theorems for deep linear networks are novel, and they directly map to simulations of training such networks using GD (Figs 2,3,6). The paper discusses oscillations in smaller subspaces first (Theorem 1), then extends the analysis to larger subspaces (Theorem 2). The theorems help theoretically ground observations seen in existing work, on loss oscillations in the EOS regime. It is curious that these oscillations can be somewhat explained by singular value oscillations in the top subspaces for deep linear networks.

**Weaknesses:**

There are 3 main weaknesses:

1. While it is useful to understand learning dynamics of deep linear networks from a theoretical perspective*, I'm not sure the results shed light on dynamics in deep nonlinear networks. In fact, the paper misinterprets the result of [1], which show that only for shallow networks are linear and nonlinear networks (2-layer bias-free ReLU networks) similar---for deep networks there is a strict separation in function approximation. [1] reports similarities between deep linear and nonlinear networks under very strong assumptions on data distribution and structure, which this manuscript does not consider. This submission misses this nuance (in abstract and lines 80-83), and overpromises results for deep nonlinear networks. In fact, the manuscript comments about the differences in loss landscapes in Line 511---nonlinear networks have highly irregular landscapes, and when trained with (minibatch) SGD could lead to catapults across regions not just in the vicinity. This is a major weakness in my view.
2. The $\alpha$-scaled initialization scheme is not used in practice, especially for LoRA training where one of the matrix (A) is initialized at 0 and the other (B) with gaussian entries. The paper argues that the results work when initialization leads to convergence to the "singular vector stationary set" (line 148). Why is this not a strong and impractical assumption on how initialization is done in practice? Does random initialization satisfy this condition?
3. As highlighted in Section 5 by the authors, the assumptions on singular values' "strict balanced state" and initialization's "stationary set condition" are strong. It's not clear if and when they can be met.

*The authors need not motivate deep linear networks by (wrongly) arguing their similarity to deep nonlinear networks; linear networks have been studied in the ML optimization research for a while now.

# Experiments
Section 4.2 LoRA initialization does not match practice. In LoRA, one of the matrices is set to 0 and the other to random, so that $AB^T$ is 0 at finetuning initialization. This is different from the alpha-initialization, which is (1) not random and (2) not starting from 0. I don't think this experiment works well with the theoretical setting studied in this paper---the LoRA finetuning experiment seems ad-hoc when the rest of the paper is about deep linear networks.

Moreover, the "catapult" observation in LoRA experiments is on very shaky ground, since stochasticity affects the training loss and makes it oscillate. The paper mentions this in Line 431, so I'm not convinced that the oscillations are, in fact, "catapults".

A natural experiment to try: In Line 188, step size is suggested to be chosen based on the depth $L$ of the network (as $L$ is larger, smaller values of step size will lead to oscillcations). This is curious to verify.

# Writing
1. In the introduction, 3 observations are described from [2, 3], but it is not clear to me how or which ones the paper addresses. It is also not clear to me how sharpening and catapults from the two papers are related to each other.
2. Definition 2 is introduced abruptly without need or context. There is discussion in Lines 191-195, which can be better placed before/after Def. 2.
3. The paper is missing a related work section. It is included as Appendix A.1, but I'd suggest cutting down on the main sections to include related work within 10 pages.

[1] Zhang, Y., Saxe, A., & Latham, P. E. (2024). When Are Bias-Free ReLU Networks Like Linear Networks?. arXiv preprint arXiv:2406.12615.

[2] Cohen, J. M., Kaur, S., Li, Y., Kolter, J. Z., & Talwalkar, A. (2021). Gradient descent on neural networks typically occurs at the edge of stability. arXiv preprint arXiv:2103.00065.

[3] Zhu, L., Liu, C., Radhakrishnan, A., & Belkin, M. (2023). Catapults in SGD: spikes in the training loss and their impact on generalization through feature learning. arXiv preprint arXiv:2306.04815.

**Questions:**

# High-level questions
1. Theorem 1. Why is this called 1-d subspace oscillation?
2. Figure 2. There are 2 kinks in the EOS regime in the left plot but only 1 is visible in the right plot. Do the authors know why?
3. Section 4.2. The original LoRA paper [1] only applies adaptation to the attention weight matrices. Any reason the authors adapt all weights?
4. Line 450. Pearson correlation between what two quantities? If this is the metric from the STS-B task, then how would this kind of experiment extend to other tasks?
5. Line 474. Why do smooth, low-frequency images correspond to low task complexity? For the CIFAR dataset, number of samples $N$ is used as a proxy task complexity, why? Both datasets are image datasets. It seems to me that I can pick and choose any proxy/hyperparameter from the experiment setup to argue that sharpness does not rise to EOS regime in low complexity tasks.

# Low-level questions
1. Figure 1. What is the pearson correlation in the last plot? A pointer to the relevant section would help.
2. Line 71. What is "sharpening"? First time this term is mentioned.
3. Lines 249-254. Lost here, what are $f_{\Delta_i}$, $x$, and the line? I think more text is needed to explain how the analysis is done for $r$-dimensional oscillations. A figure would really help the reader.

[1] Hu, E. J., Shen, Y., Wallis, P., Allen-Zhu, Z., Li, Y., Wang, S., ... & Chen, W. (2021). Lora: Low-rank adaptation of large language models. arXiv preprint arXiv:2106.09685.


*************
**Update after author response:** I am increasing my review score from 3 to 6 since the authors addressed my [concerns](https://openreview.net/forum?id=J4Dvxv7WnG&noteId=9ZRUV9lccn). The manuscript has been updated significantly, but not to the extent that I do not understand the central claims. Hence, I'm keeping my original confidence score at 3. Thank you to the authors for a good discussion.
*************

---

> ### Author Response · Authors · 2024-11-19
> **Author Response (Part 1)**
>
> We thank the reviewer for their detailed feedback on our work that helped improve our manuscript. Below, we respond to each of the concerns in more detail.
>
> **Comment:** I'm not sure the results shed light on dynamics in deep nonlinear networks.
>
> **Response:** Thank you for highlighting this point. We have revised our manuscript to mainly focus on DLNs, highlighting the similarities and differences between linear and non-linear networks. We removed the claim that DLNs can explain the phenomena that happen in practical non-linear networks at EOS  and added a 2D toy experiment that highlights a significant difference between these two models (see [Figure 8](https://imgur.com/CsGOdap)).
>
> To also better reflect our contributions in DLNs along with its (potential) connections to nonlinear networks, we have revised our manuscript to focus more on DLNs. Primarily, we changed the tone of the paper to focus on DLNs, while still explaining how our results might contribute to some of the unexplained phenomena in deep nonlinear networks. Moreover, to have a fairer comparison between the two networks, we added a figure in Section 4.2 that highlights the differences of the behaviors in EOS. We do this by visualizing the loss landscape using a [2-D depth-4 network](https://imgur.com/3UZNmjP) and a [Holder table function](https://imgur.com/7xlMhUA).  We chose the Holder table loss function as a proxy test for deep non-linear network which is known to have numerous local minimas. Whereas deep linear networks contains only saddle points and global minimas (no spurious local minimas).
>
> Regarding our connections on the similarities between linear and nonlinear networks, we referenced [1] to highlight particular cases in which behaviors at EOS are similar. For example, our experiments with bias-free ReLU networks in [Figure 17](https://imgur.com/ezbbKZV) imply that there are basins that support sustained oscillations over some local neighborhood before catapulting happens.
>
> **Comment:** $\alpha$-scaled initialization scheme is not used in practice.
>
> **Response:** While the $\alpha$-scaled initialization may not be used for nonlinear networks, it is a very common setting to study for DLNs, as it corresponds to the “rich” training regime. The rich regime promotes "incremental learning", where training performs greedily to fit the target singular values [2]. In fact, Figure 4 of [2] shows that test accuracy improves with a smaller alpha for image classification datasets. To the best of our knowledge, a small initialization scale is used in most (if not all) works that study DLNs. Furthermore, to show that the convergence to the singular vector stationary set is not an impractical assumption, we added additional experiments in the Appendix to show that many other initializations converge to such a set. This includes identity (which is from the original manuscript), orthogonal, and random initialization (see [Figure 15](https://imgur.com/xON7b4q)).
>
> **Comment:** the assumptions on singular values' "strict balanced state" and initialization's "stationary set condition" are strong
>
> **Response:** Although the strict balance state may seem strong, we only consider it for unbalanced initializations to cover a wide range of initialization schemes. However, for balanced initialization, the strict balanced state is not an assumption, which we added to the revised manuscript.
> Moreover, all of our experiments show that these two conditions are generally met, provided that they satisfy the conditions in our theorem statements (e.g. the initialization scale needs to satisfy the threshold). We provided additional experiments for these assumptions, which are available in Appendix B.4 (see [Figure 15](https://imgur.com/xON7b4q) and [Figure 14](https://imgur.com/7RSHpvN)). Furthermore, we would like to point out that almost all the literature on deep matrix factorization assumes that these two conditions hold exactly. In our work, we go one step further to show increasingly balancedness.
>
> Finally, we note that these assumptions are mainly used such that we can focus on analyzing the periodicity that arises in the singular values – they are tools to make the analysis a little bit “simpler”.
>
>
> [1] Yedi Zhang, Andrew Saxe, Peter E. Latham. "When Are Bias-Free ReLU Networks Like Linear Networks?". https://arxiv.org/abs/2406.12615
>
> [2] Blake Woodworth and Suriya Gunasekar and Jason D. Lee and Edward Moroshko and Pedro Savarese and Itay Golan and Daniel Soudry and Nathan Srebro. "Kernel and Rich Regimes in Overparametrized Models". https://arxiv.org/abs/2002.09277

---

> ### Author Response · Authors · 2024-11-19
> **Author Response (Part 2)**
>
> **Comment:** Experiments: LoRA initialization does not match practice.
>
> **Response:**  As suggested by another reviewer, we moved the section LoRA to the Appendix, as we do not state it as a contribution but rather pose it as a future work. Generally, we use the same setup as described by [7] where $\alpha$-scaled random initialization was used along with adapting the attention and MLP weights. We did not clarify that we used random weights in the original manuscript, which we have now included in appendix. The Pearson correlation corresponds to measuring the correlation of the predicted labels to the true labels in the test set, and can be viewed as testing the performance of the fine-tuned model. If the Pearson correlation is higher than another, then we can conclude that one fine-tuned model is better than the other.
>
> **Comment:**  A natural experiment to try: In Line 188, step size is suggested to be chosen based on the depth
>
> **Response:** We performed this experiment in Section B.4, we do indeed verify this claim (see [Figure 16](https://imgur.com/rzEvH2Y)).
>
> **Comment:**  In the introduction, 3 observations are described from [2, 3], but it is not clear to me how or which ones the paper addresses. It is also not clear to me how sharpening and catapults from the two papers are related to each other.
>
> **Response:**  This confusion probably arose since [3] examines catapults in the top eigenvector space of the NTK matrix, whereas in our paper we study the catapult in the top eigenvector space of the Hessian. The NTK and the Hessian behaves almost similarly near the end of training since the residual goes to zero for MSE loss, which the authors in [3] address in Appendix A.1. In our paper, we study sharpening (or progressive sharpening) in DLNs for the max eigenvalue of the Hessian like in Cohen et al. and then prove the oscillatory phenomenon in the top eigenvector subspace of the Hessian based on the learning rate. The latter observation validates experiments in [3] since they observe catapults in the top eigenvector space of the NTK (or equivalently the Hessian). We make this clear in the introduction.
>
> **Comment:**  Definition 2 is introduced abruptly without need or context.
>
> **Response:**  Our revised manuscript removes this abruptness and addresses the importance of strict balancing before defining it.
>
> **Comment:**  "Theorem 1. Why is this called 1-d subspace oscillation?"
>
> **Response:**  In Theorem 1, the oscillations in subspace is a rank-1 outer-product. Hence, we changed the definition to a *rank-1 oscillation* instead of 1-d subspace oscillation. Initially, we stated it as 1-D oscillations since oscillations took place along the top eigenvector of the flattened Hessian of the DLN loss.
>
> **Comment:**  "Figure 2. There are 2 kinks in the EOS regime in the left plot but only 1 is visible in the right plot. Do the authors know why?"
>
> **Response:** Thanks for pointing this out. This is an artifact of plotting in the logarithm scale. For DLNs with small initialization, the sharpness is negative in the earlier learning stages, since this region is a saddle point with a negative curvature. Hence, the logarithm (along with some irregular quantization) made the negative sharpness look like a "kink". We updated the figure.
>
> **Comment:** Why do smooth, low-frequency images correspond to low task complexity? For the CIFAR dataset, number of samples is used as a proxy task complexity, why? Both datasets are image datasets.
>
> **Response:**  Smooth/low frequency images are much easier to fit for convolutional neural networks due to *spectral bias*
>  [4], where the networks incrementally learns each Fourier Frequency (low to high). Similarly, fitting simple datasets such as MNIST or even datasets with less images are low complexity tasks since the intrinsic update dimension in parameter space is much smaller [5, 6]. However, the exact relation between the intrinsic dimension of parameter update and it's sharpness is unknown, it is a common observation that sharpness do not rise much when the network easily/quickly fits the underlying function (appendix in Cohen et al). Although currently the relation between complexity of the tasks in linear and non-linear network are far from perfect, the top singular value of the target matrix **$M$** is a proxy for complexity. We updated the manuscript to reflect this.

---

> ### Author Response · Authors · 2024-11-19
> **Author Response (Part 3)**
>
> **Comment:** Line 71. What is "sharpening"? First time this term is mentioned.
>
> **Response:** We have defined "progressive sharpening" at the start of Section 3 initially. Now, we included it where we defined EOS so the reader can understand what "sharpening" is referred to as.
>
> **Comment:** I think more text is needed to explain how the analysis is done for r-dimensional oscillations. A figure would really help the reader.
>
> **Response:**  We added figure which highlights the statement of this theorem in the Appendix ([Figure 10](https://imgur.com/3UZNmjP)). For example the X-axis and Y-axis here refers to the two perpendicular lines in which oscillation occurs. In higher dimensions, these orthogonal directions are denoted by $\Delta_{i}$, (which are the eigenvector directions of the Flattened Hessian) which passes through the global minima (we referred to as $x$ in paper). Hence, this line is denoted as $y=x+t\Delta_{i}$.
>
>
> References:
>
> [2] Jeremy M. Cohen, Simran Kaur, Yuanzhi Li, J. Zico Kolter, Ameet Talwalkar "Gradient descent on neural networks typically occurs at the edge of stability". arXiv preprint arXiv:2103.00065.
>
> [3] Libin Zhu, Chaoyue Liu, Adityanarayanan Radhakrishnan, Mikhail Belkin. "Catapults in SGD: spikes in the training loss and their impact on generalization through feature learning". arXiv preprint arXiv:2306.04815.
>
> [4] Nasim Rahaman and Aristide Baratin and Devansh Arpit and Felix Draxler and Min Lin and Fred A. Hamprecht and Yoshua Bengio and Aaron Courville. "On the Spectral Bias of Neural Networks." https://arxiv.org/abs/1806.08734
>
> [5] Chunyuan Li and Heerad Farkhoor and Rosanne Liu and Jason Yosinski. "Measuring the Intrinsic Dimension of Objective Landscapes". https://arxiv.org/abs/1804.08838
>
> [6] Phillip Pope and Chen Zhu and Ahmed Abdelkader and Micah Goldblum and Tom Goldstein. "The Intrinsic Dimension of Images and Its Impact on Learning". https://arxiv.org/abs/2104.08894
>
> [7] Yaras, Can, et al. "Compressible Dynamics in Deep Overparameterized Low-Rank Learning & Adaptation." arXiv preprint arXiv:2406.04112 (2024).

---

> > ### Comment · Reviewer_GbsE · 2024-11-22
> > **Reply to Rebuttal**
> >
> > Thank you for the revision. I think the authors put in a lot of effort during the rebuttal period to address my concerns. Notably, they have
> > - Toned down connections to deep nonlinear networks.
> > - Moved LoRA experiments to the appendix, and leave it to future work to study.
> > - Added experiments to demonstrate that various initialization schemes, including random, lead to the 2 assumptions on relevant singular values.
> > - Added an experiment to test choice of step size based on the depth.
> >
> > The updates directly address my points, and I'd be open to increasing my score to see this paper at the conference. I must say that the manuscript has changed significantly during this discussion phase. Unfortunately, I have not been able to read the revised manuscript with the same care as I did earlier. So I am less confident about my understanding.
> >
> > I appreciate the authors work on adding these new experiments. They are quite convincing about EOS phenomena in deep linear networks.

---

> ### Author Response · Authors · 2024-11-23
> **Author response (Part 4)**
>
> We sincerely thank the reviewer for their thoughtful analysis of our work and for providing constructive feedback, which has greatly helped us improve the manuscript. We would like to clarify that while the revisions may appear substantial, the major changes are primarily related to the connections with deep nonlinear networks and the removal of the LoRA section.
>
> If there are any additional clarifications or questions that could help address your concerns further, we would be happy to provide them. We kindly request you to consider these improvements and reassess the score for our submission.

---

### Official Review · Reviewer_CF1F · 2024-11-09

**Soundness:** 3
**Presentation:** 3
**Contribution:** 3
**Rating:** 8
**Confidence:** 3

**Summary:**

This paper studies the learning dynamics of deep linear networks trained using gradient descent at a fixed learning rate, especially beyond the edge of stability (EOS) regime.

Using the singular value stationary set and the strictly balancing assumptions (both can be validated in examples), the paper argues that within the EOS regime, periodic 2-period fixed orbit oscillations occur in a low-dimensional subspace, with the subspace dimension determined by the learning rate. This phenomenon explains the presence of oscillations within top features. Additionally, the paper compares the difference between deep linear networks and diagonal linear networks in the EOS regime, offering insights into how network structure impacts training dynamics. Experiments further reveal that fine-tuning with large learning rates in LoRA for low-rank adaptation of large language models induces "catapult dynamics," which can potentially improve generalization.

**Strengths:**

This work is both theoretically rigorous and experimentally thorough. The theoretical results are new to me, especially Theorem 1, and the proofs are rigorous and not difficult to follow. Additionally, the experiments are extensive and effectively validate the theoretical insights. Overall, the study contributes valuable advancements in understanding and optimizing the training dynamics of deep networks.

**Weaknesses:**

Although deep linear network shares some similarities with deep nonlinear networks, the claim that ``Our analysis explains two key phenomena in deep nonlinear networks" in abstract seems overstated and exaggerated. In the analysis, the initialization (eqn (3)) is very specific, and it is just a member in the singular vector stationary set. It is not clear if all statements work only based on the initialization (eqn (3)) or for any initialization in the singular vector stationary set. For example, will Lemma 1 and Lemma 2 work with general singular vector stationary initial data?

**Questions:**

1. Why is the period of the fixed orbit is always 2? It seems that it comes from the two-step GD update. How about other periodicities?

2. (15)-(16) on page 19-20 seem to have typos.  $\dot{W}_l(t)$ on the left side should be $\dot{U}_l(t)$?

---

> ### Author Response · Authors · 2024-11-19
> **Author Response**
>
> We appreciate the reviewers insightful comments. We have also fixed the typos. Below, we provide concrete responses regarding the concerns:
>
> **Comment:** "the claim in the abstract seems overstated and exaggerated."
>
> **Response:** As stated in the global response, we have updated our manuscript by toning down their similarities and highlighting the similarities and differences between the two networks. For example, in [Figure 8](https://imgur.com/CsGOdap), we plot the landscape of a two variable depth-4 linear scalar network and GD dynamics at EOS. Along with it, we also plot GD optimization trajectory for a Holder table landscape (which is highly non-convex with local minimas) for increasing learning rates [Figure 11](https://imgur.com/7xlMhUA). Here, the sharpness closely follows the limit $2/\\eta$ supporting the EOS observations made in Cohen et al. This helps visualize the differences (along with similarities) between the two networks. Overall, we had adapted our paper to focus more on DLNs.
>
> **Comment:**  "It is not clear if all statements work only based on the initialization (eqn (3)) or for any initialization in the singular vector stationary set."
>
> **Response:** The theoretical results hold for *any* initialization that converges to the SVS set in Proposition 1. We choose the initializations in Equation (3) for concreteness, as we prove in Propositions 2 and 3 that these initializations belong to the SVS set. Hence, our analysis can be extended to many different forms of initializations, as long as they can be shown to satisfy Proposition 1.
>
> **Comment:** "Why is the period of the fixed orbit is always 2? It seems that it comes from the two-step GD update. How about other periodicities?".
>
> **Response:** This is an excellent question. In [Figure 1](https://imgur.com/RD2ZWGq) of the revised manuscript, we provide a bifurcation plot showing that other periodicities also exist for a properly chosen learning rate. This shows a period-doubling route to chaos (and then to divergence). Our paper focuses on the period-2 regime, as it still shows a sufficient amount of information to understand what happens in the EOS regime for DLNs. We leave the other cases (i.e., analyzing what learning rates are required for more periodicities) for future work.
>
> **Comment:** "on page 19-20 seem to have typos".
>
> **Response:** Thanks for pointing it out. We have fixed it.

---

> > ### Comment · Reviewer_CF1F · 2024-11-26
> > **Reply to your revision**
> >
> > Thanks a lot for your tremendous efforts in revising and improving the paper.
> >
> > All my concerns have been clearly addressed. Overall, I think this paper presents an interesting topic and is theoretically novel. Thus I decide to increase my score to recommend acceptance.

---

### Author Response · Authors · 2024-11-19
**Global Response to All Reviewers**

Firstly, we would like to thank the reviewers for taking the time to provide insightful comments. We are encouraged that the reviewers find our work “theoretically rigorous and experimentally thorough” (Reviewer CF1F) and “a solid contribution to the theoretical understanding of the Edge of Stability” (Reviewer n4c2). Based on the comments, we have uploaded a revised manuscript, where the revisions can be summarized as follows:
- Toned down our claims on deep nonlinear networks to say that our results shed light on certain phenomena in nonlinear networks rather than directly explaining them.
- Moved LoRA to the Appendix as suggested by Reviewer n4c2 and edited the manuscript to have a stronger focus on DLNs.
- Clarificaiton regarding the use of singular vector stationary set.


In the following, we provide a few more thorough responses regarding shared concerns amongst the reviewers.

**Comment:** Overclaiming results on deep nonlinear networks using our results in deep linear networks.

**Response:** While we believe that our results still shed some light on phenomena that occur in deep nonlinear networks within the EOS regime, we agree that our claims may have been slightly overexaggerated. As the reviewers point out (and stated in our experiments section), DLNs fail to capture catapults that occur in nonlinear networks due to the difference in the loss landscape. To address this point,  we have revised our manuscript in two ways: (i) toned down the contributions such that we say that our results on DLNs contributes to an understanding of phenomena in EOS for nonlinear networks rather than directly explaining them; (ii) provide a fair comparison on the similarities and differences between the two networks in Section 4.2. See [Figure 1](https://imgur.com/3UZNmjP) and [Figure 2](https://imgur.com/7xlMhUA) for landscape visualization for DLNs and complex non-convex function such as Holder table loss at EOS. We chose the Holder table loss function as a proxy for deep non-linear network which is known to have numerous local minimas. Whereas deep linear networks contains only saddle points and global minimas (no spurious local minimas).


**Comment:** The use of the singular vector stationary set is a strong assumption.

**Response:** While this may seem like a strong assumption, we do observe that many of the commonly used initialization schemes ultimately converge to the singular vector stationary set for DLNs. In [Figure 15](https://imgur.com/xON7b4q), we show that the identity, orthogonal, and random initializations all converge to this set empirically. Recall that we do prove that the identity initialization we considered in the original manuscript converged to this set. Since it is exhaustive to prove that all initializations converge to such a set, we used Proposition 1 to cover an extensive base called the SVS set. Theorem-1 for rank-1 subspace oscillation can be easily modified when the initialization is made inside any general point in the SVS set.


**Comment:** Concerns about LoRA

**Response:** Based on most reviewers' comments, we moved this section to the Appendix and left it as future work for more investigation.

---

### Meta-Review · Area_Chair_kuVj · 2024-12-17

**Metareview:**

This paper studies the edge of stability (EoS) phenomenon in deep matrix factorization, also known as deep linear neural networks. The paper analyzes the 2-period oscillation in GD dynamics when the learning rate is larger than the stability threshold, under two assumptions: singular vector stationary set, and strict balanced state.

The reviewers unanimously found the paper to be an interesting submission. To me as well, it is deemed that the paper delivers useful insights into the low-dimensional oscillations in the EoS regime and offers technical tools for analyzing deep matrix factorization. A downside of the paper is that the main results hold under two assumptions: singular vector stationary set, and strict balanced state, as the authors also note in the conclusion.

Although I recommend acceptance of this paper, I am concerned about a slight lack of precision in the authors’ claims regarding their assumptions. It seems to me that the authors assume that some properties that only hold for GF (the continuous time dynamics) also hold for GD with large step sizes. The preservation of balancedness is a good example: For GF, once $W_{j+1}^T(0) W_{j+1}(0) = W_j(0) W_j(0)^T$, the property continues to hold forever. However, for GD the balancedness property slightly breaks every iteration due to the “discretization error”. For example, consider the problem $L(A,B) = 1/2*  \\lVert A B - W \\rVert_F^2$. The GD update for $A$ and $B$ look like $A_+ = A - \eta (A B - W) B^T$ and $B_+ = B - \eta A^T (A B-W)$. Even if $A$ and $B$ satisfy $A^T A = B B^T$, we can realize that $A_+^T A_+ = A^T A - \eta B (A B-W)^T A - \eta A^T (A B - W) B^T + \eta^2 B (A B-W)^T (A B-W) B^T$ and $B_+ B_+^T = B B^T - \eta B (A B-W)^T A - \eta A^T (A B - W) B^T + \eta^2 A^T (A B-W) (A B-W)^T A$, and hence the $O(\eta^2)$ terms do not match in general.

Nevertheless, I believe this issue may not be grounds for rejection because 1) the preservation of balancedness may be true under the singular vector stationary set assumption, 2) the authors may be able to extend Lemma 2 and show that the balance is (approximately) preserved in the balanced initialization case, 3) the empirical observations show balancedness should hold anyway. In any case, I encourage the authors to take a serious look into this matter and make proper adjustments.

**Additional Comments On Reviewer Discussion:**

The summary includes a summary on the discussion of strengths/weaknesses raised by the reviewers. Other issues raised by the reviewers raised by the reviewers and addressed by the authors include:
- Refinement of Lemma 2: Some reviewers initially claimed that the initial version of Lemma 2 does not necessarily imply that the singular values eventually become balanced. The authors made refinements to the lemma.
- Claims on LoRA: The initial version included some experiments related to LoRA finetuning. Some reviewers did not found it very convincing; the authors moved them to appendix.

During meta review, I (AC) think I found some other minor typos.
1. In Theorem 1, shouldn’t $\sigma_{*,j}$ have exponents $1/L$?
2. In line 242, “$K$ less than the second largest eigenvalue” -> greater than?
3. Theorem 2, “$\lambda_i \leq K < \lambda_{i+1}$” -> ineqs reversed?

---

> ### Public Comment · ~Soo_Min_Kwon1 · 2025-02-05
> **On the Balancedness of the Singular Values**
>
> Thank you for pointing out the typos—those will be fixed in the final manuscript. We also appreciate your efforts in overseeing the discussion during the rebuttal phase.
>
> Regarding the balancedness, while your argument is correct, the singular vector stationary set assumption does make the higher order terms match and so the preservation does hold for the balanced initialization case. For the unbalanced case, please note that we never assumed that the conserved flow in GF (i.e., the balancedness of the weights) holds for finite step sizes in GD. We apologize for any confusion in the manuscript that may have led to this conclusion. In fact, as you pointed out, with finite step sizes, the layerwise balancing breaks; however, **at EoS, this balancing gap converges to zero**. This is the central argument of our proof detailed in Lemma 2, and has been discussed extensively with Reviewer nnRE and n4c2.
>
> To clarify this point, we plan to add a figure in the camera-ready version that clearly delineates the differences between GF, GD with step sizes below the EoS regime, and GD in the EoS regime. Our theorem and lemma are consistent with our empirical observations: balancedness breaks for finite step-size GD, and at EoS, the balanced quantity monotonically converges to zero.

---

### Decision · Program_Chairs · 2025-01-22

Accept (Poster)